# BigDocs: An Open Dataset for Training Multimodal Models on Document and Code Tasks

Juan Rodriguez[1,2,3,a]   Xiangru Jian[1,4,a]   Siba Smarak Panigrahi[1,2,a]   Tianyu Zhang[1,2,5,a]

Aarash Feizi[1,2,6,a]   Abhay Puri[1,a]   Akshay Kalkunte[1,a]   François Savard[1,a]

Ahmed Masry[1,7,b]   Shravan Nayak[1,2,5,b]   Rabiul Awal[1,2,5,b]   Mahsa Massoud[1,2,6,b]

Amirhossein Abaskohi[1,8,b]   Zichao Li[1,2,6,b]   Suyuchen Wang[2,5,b]   Pierre-André Noël[1,b]

Mats Leon Richter[1,b]   Saverio Vadacchino[1]   Shubham Agarwal[1,2]   Sanket Biswas[9]

Sara Shanian[1]   Ying Zhang[1]   Noah Bolger[1]   Kurt MacDonald[1]   Simon Fauvel[1]

Sathwik Tejaswi[1]   Srinivas Sunkara[1]   Joao Monteiro[1]   Krishnamurthy DJ Dvijotham[1]

Torsten Scholak[1]   Nicolas Chapados[1]   Sepideh Kharagani[1]   Sean Hughes[1]   M. Özsu[4]

Siva Reddy[1,2,6,10]   Marco Pedersoli[1,3]   Yoshua Bengio[2,5,10]   Christopher Pal[1,2,10,11]

Issam Laradji[1,8]   Spandana Gella[1]   Perouz Taslakian[1]   David Vazquez[1]   Sai Rajeswar[1,2]

[1]ServiceNow   [2]Mila   [3]École de Technologie Supérieure   [4]University of Waterloo   [5]Université de Montréal   [6]McGill University
[7]York University   [8]University of British Columbia   [9]Universitat Autònoma de Barcelona   [10]CIFAR AI Chair   [11]Polytechnique Montréal

[a]First Author Equal contribution.   [b]Second Author Equal contribution.

## Abstract

Multimodal AI has the potential to significantly enhance document-understanding tasks, such as processing receipts, understanding workflows, extracting data from documents, and summarizing reports. Code generation tasks that require long-structured outputs can also be enhanced by multimodality. Despite this, their use in commercial applications is often limited due to limited access to training data and restrictive licensing, which hinders open access. To address these limitations, we introduce BigDocs-7.5M, a high-quality, open-access dataset comprising 7.5 million multimodal documents across 30 tasks. We use an efficient data curation process to ensure our data is high-quality and license-permissive. Our process emphasizes accountability, responsibility, and transparency through filtering rules, traceable metadata, and careful content analysis. Additionally, we introduce BigDocs-Bench, a benchmark suite with 10 novel tasks where we create datasets that reflect real-world use cases involving reasoning over Graphical User Interfaces (GUI) and code generation from images. Our experiments show that training with BigDocs-Bench improves average performance up to 25.8% over closed-source GPT-4o in document reasoning and structured output tasks such as Screenshot2HTML or Image2Latex generation. Finally, human evaluations showed a preference for outputs from models trained on BigDocs over GPT-4o. This suggests that BigDocs can help both academics and the open-source community utilize and improve AI tools to enhance multimodal capabilities and document reasoning. The project is hosted at **https://bigdocs.github.io**.

## 1 Introduction

*Visually-Rich Document* data containing text and structured elements (such as charts, infographics, diagrams, sketches, tables, etc.) are essential cues for users to efficiently understand complex information holistically. To facilitate this understanding, foundation models must process these structured documents and subsequently extract key insights, identify patterns, and generate concise summaries and responses from user requests with reasoning (Landeghem et al., 2023; Zhu et al., 2022). Recent advances in multimodal AI (Yang et al., 2023; Team et al., 2024b) have demonstrated impressive capabilities, including generating functional web pages (Zheng et al., 2024), automating document understanding workflows (Wang et al., 2023b), and extracting detailed information from

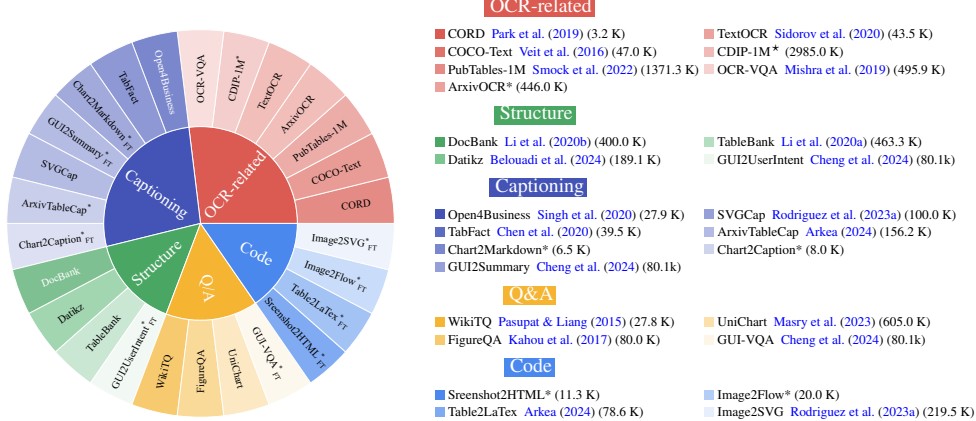

**Figure 1:** *BigDocs: A Large-Scale Structured Continual Pretraining and Finetuning Dataset.* The inner circle represents the distribution of BigDocs, detailing the categories. The outer circle displays the specific datasets compiled to form 7 million image-text pairs. Datasets with * denotes our contribution.

documents to produce comprehensive reports (Borchmann, 2024). However, the datasets used to train these models remain closed-source, with undisclosed details and restrictive licensing, which hampers their broader adoption in research and the advancement of open-source model development. In contrast, current open-source models and datasets (Chen et al., 2023; Liu et al., 2023b; 2024) primarily focus on basic document understanding tasks, such as optical character recognition (OCR), e.g., DocStruct4M (Hu et al., 2024), or basic question-answering and mathematical problems, e.g., Cambrian-7M (Tong et al., 2024). These efforts do not sufficiently address the complexity of processing intricate visual documents or generating long-structured outputs, such as JSON and HTML, which are valuable in real-world applications.

In this work, we present BigDocs (https://bigdocs.github.io), a large-scale and open *dataset*, *benchmark suite*, and *models* specifically designed for user-facing document-related tasks. BigDocs bridges existing gaps by enabling open-source models to meet the rising demand for sophisticated document understanding technologies. Comprising 7.5 million image-text pairs, BigDocs is carefully curated to support three core areas: (1) **Document Information Extraction**, which includes enhanced OCR for diverse document types, named entity recognition, layout analysis, and table detection; (2) **Document Understanding**, covering semantic comprehension tasks such as document classification, question answering, and analysis of diagrams; and (3) **Document Creation and Manipulation**, converting visual data into structured formats like HTML, LaTeX, and JSON.

Our survey of 133 existing datasets revealed that 80% of them (i.e., around 100 datasets) have either non-permissive licenses (Jaume et al., 2019; Štěpán Šimsa et al., 2023) or no clear licensing information (Chaudhry et al., 2019; kleister Charity, 2021), creating barriers to reuse and transparency. In response, BigDocs prioritize datasets with permissive licenses (e.g., CC-BY-4.0, MIT) and document-related information, ultimately retaining 16 fully accessible datasets. To further support accessibility, we developed the BigDocs Toolkit, which offers modular tools for data preprocessing, filtering, and consolidation. Additionally, we introduced a unified metadata framework to enhance dataset traceability (e.g., properties, sources, licenses), including detailed documentation of transformations applied by us to the original data. We also conduct a data contamination analysis on downstream tasks data, showing that BigDocs-7.5M has lower contamination rates than previous datasets.

To further advance document intelligence, BigDocs-Bench offers 10 novel downstream tasks, each with four splits: train, validation, test, and hidden test (with 329k training samples, 11k validation samples, 10k testing samples). These tasks focus on structured output generation, including code formats such as HTML, LaTeX, SVG, and Markdown. Our experimental results demonstrate that models trained on the BigDocs suite outperform those trained on existing datasets like DocStruct4M (Hu et al., 2024) on standard document benchmarks. Additionally, automatic and human evaluations on the novel tasks introduced in BigDocs-Bench highlight the advanced capabilities of these models in generating long-format, structured outputs. User evaluations reveal a preference for our models' outputs 88% of the time over Phi3.5 Instruct and 63% over GPT-4.

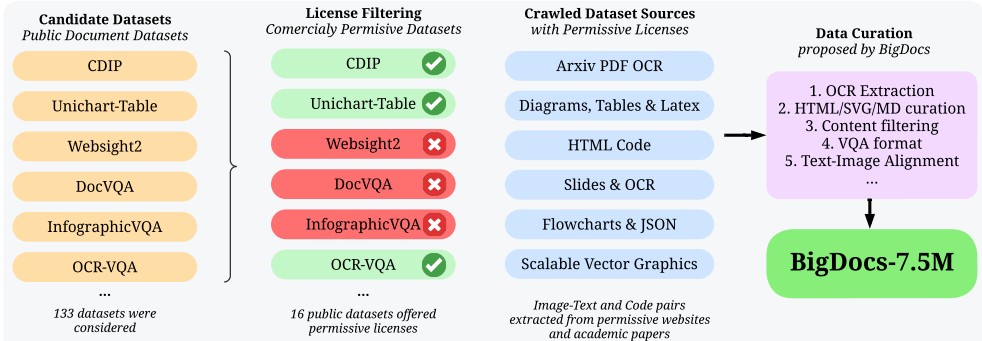

**Figure 2:** *BigDocs-7.5M Dataset Curation. The figure illustrates the extraction, filtering, and curation process of BigDocs-7.5M, which emphasizes maintaining permissive licensing. To build BigDocs-7.5M, we first gather publicly-available vision-language datasets, particularly those centered on document analysis, and apply a rigorous filtering process. We then augment these datasets with our own crawled data. Finally, we standardize all samples and tasks into a unified format to produce BigDocs-7.5M.*

Built with a commitment to accountability, responsibility, and transparency (ART) (Bommasani et al., 2023; Vogus & Llansóe, 2021), BigDocs is open-sourced, including datasets, models, and documentation to foster responsible AI development. In summary, BigDocs contributions include:

1. **BigDocs-7.5M**, a large-scale, license-permissive dataset designed for continual pretraining (further training from a pretrained foundation model checkpoint) and downstream finetuning (e.g., to follow instructions or task formats) of multimodal models on document-related tasks. It includes traceable *metadata* and curated licensing drawing from document-rich *multiple* data sources, ensuring full public accessibility.
2. **BigDocs-Bench**, a set of 10 new benchmarks, including test datasets as well as corresponding innovative evaluation metrics for multimodal models to generate long-structured code outputs from images, including formats such as HTML, LaTeX, Markdown, and SVG.
3. **BigDocs Toolkit**, unified tools supporting open-source efforts. These tools allow efficient data curation, filtering, formatting, and preparation for training models generating structured outputs.
4. **BigDocs Models**: We conduct extensive experiments using four state-of-the-art public models, demonstrating the advantages of training with BigDocs over alternative datasets and enabling the models to learn novel tasks through our dataset suite.

## 2 RELATED WORK

**Multimodal Datasets.** General-purpose vision-language datasets like COCO Caption (Chen et al., 2015) and SBUCaption (Ordonez et al., 2011) primarily feature photographic content, lacking visually-rich document (VRD) data (Sharma et al., 2018; Changpinyo et al., 2021). In contrast, our focus is on text-heavy datasets including PDFs, tables, and invoices (Veit et al., 2016; Mishra et al., 2019; Singh et al., 2021; Li et al., 2020b;a; Soboro, 2022), which are crucial for tasks like information extraction and parsing (Masry et al., 2022; Rodriguez et al., 2023c;b). While datasets like DocStruct4M (Hu et al., 2024) and Cambrian7M (Tong et al., 2024) partially address these needs, they often lack permissive licenses or are not open-source. Kosmos-2.5 (Lv et al., 2023) focuses on building a large document dataset; however, the authors did not make it public. BigDocs fills these gaps by providing 7.5M permissively licensed image-text pairs from 16 academic datasets and other open platforms, supporting diverse document understanding tasks.

**Responsible Data and Licensing.** Enterprise models like GPT-4 (OpenAI, 2023) and Claude (Anthropic, 2024) are often closed-source, lacking transparency in training data (Bommasani et al., 2024). Foundational works (Gebru et al., 2021; Bender & Friedman, 2018) emphasize the importance of open access and transparency in dataset documentation. Previous works promoting open-access such as StarCoder (Li et al., 2023), The Stack (Kocetkov et al., 2022), FineWeb (Penedo et al., 2024), and LLaMA (Dubey et al., 2024), have addressed this by releasing data or models for language models pretraining. Our work builds on these efforts by creating a curated, well-documented resource for

open access (Laurençon et al., 2023), addressing licensing complexities ranging from permissive licenses (e.g., Apache 2.0, MIT (Hu et al., 2024; Gadre et al., 2023)) to restrictive ones like CC BY-SA (Foundation, 2024; Zhang et al., 2024) and CC BY-NC-SA (Wang et al., 2024b). Compound datasets, like Cambrian-7M (Tong et al., 2024), face mixed licensing issues, while some datasets, such as DocVQA (Mathew et al., 2021b), have different licenses for images and annotations. In developing BigDocs, we prioritized permissive licensing in data curation and contributed to new datasets that adhere to open-access principles, ensuring transparency at all stages of development.

**Multimodal Document Understanding Models.** Recent advancements have introduced several general-purpose multimodal models (Liu et al., 2023b;a; 2024; Bai et al., 2023b; Laurençon et al., 2024; Tong et al., 2024). For example, LLaVA (Liu et al., 2023b) integrates a vision encoder with a language model, with later versions enhancing reasoning and OCR capabilities (Liu et al., 2023a; 2024). Qwen2-VL (Bai et al., 2023b) processes images at native resolutions, while Phi 3.5 Vision (Abdin et al., 2024) offers a lightweight model for reasoning tasks. Specialized models for visually-rich documents, like DocOwl1.5 (Hu et al., 2024) and DocLLM (Wang et al., 2023a), have also gained traction, particularly for commercial applications. However, the datasets for training these models are often not publicly available or have restrictive licenses (Hu et al., 2024). Our work, BigDocs, addresses this by consolidating existing permissive datasets to support the development of reproducible, commercially viable document understanding models.

## 3 BigDocs-7.5M

BigDocs-7.5M is a large-scale, license-permissive, and carefully curated dataset for visual document understanding designed to train foundational models across various document types and tasks. It consolidates public datasets and newly crawled data with permissive licenses by preprocessing, cleaning, and filtering them into a unified collection of 7.5 million image-text pairs. All curated datasets and related artifacts will be openly released to foster community collaboration (Bender & Friedman, 2018). The curation process is illustrated in Figure 2 and detailed below.

### 3.1 Dataset Curation Process

**Existing Dataset Acquisition.** The authors, along with domain experts and researchers, guided the collection strategy, assessing dataset relevance, quality, and diversity. We gathered 133 public vision-language datasets by searching academic repositories, open data platforms, and research papers. The collection focused on tasks like image captioning (Chen et al., 2015; Sidorov et al., 2020), OCR (Park et al., 2019; Smock et al., 2022), visual question answering (Mishra et al., 2019; Mathew et al., 2021b), scene-text recognition (Veit et al., 2016; Singh et al., 2021), and document layout analysis (Li et al., 2020a;b), resulting in a diverse multimodal dataset repository.

**Datasheets for Datasets.** During data acquisition, we compiled detailed datasheets for each dataset, capturing metadata such as ownership, status, size, references, source type, annotations, licensing, and specific observations. We filtered datasets based on licensing compatibility and relevance to our document-related tasks, then extended the datasheets of selected datasets for better categorization. This extension organized datasets by attributes such as medium type (e.g., digital, scanned), document type (e.g., articles, infographics), sourcing method, text type (e.g., computer-generated, handwritten), structure, language, timeframe, and licensing. For licensing, we documented both the image licenses and annotation licenses separately, as these often differed and impacted the overall permissiveness and usability of each dataset. This structured approach aligned datasets with specific use cases (e.g., OCR, structured parsing) and grouped them for pretraining, finetuning, and evaluation, ensuring effective integration into our visual document understanding pipeline.

**License Filtering.** A key criterion for dataset selection was ensuring permissive licenses (e.g., CC-BY, MIT, Apache 2, CC0) for both images and annotations, suitable for open access and commercial use (more details on various of licenses in Appendix A.7). Datasets with non-permissive licenses, like DVQA (Kafle et al., 2018) (CC-BY-NC 4.0) or DocILE (Štěpán Šimsa et al., 2023) (non-commercial use), or with no license information, like DeepForm (Svetlichnaya, 2020), were excluded. Ultimately, we prioritized permissive licenses for both text and images, resulting in 20% being kept, while 7.5% moderately restrictive and 72.5% non-permissive were discarded. Some included datasets still have images under less clear terms, such as "Fair Use" (e.g., OCR-VQA), documented in the metadata.

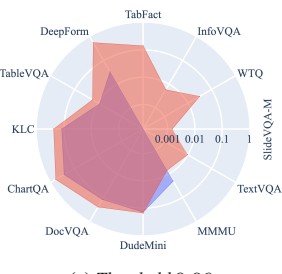 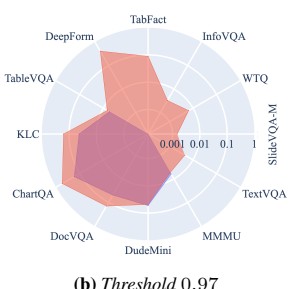 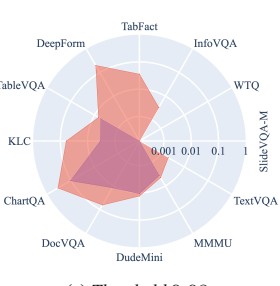

**(a)** *Threshold* 0.96      **(b)** *Threshold* 0.97      **(c)** *Threshold* 0.98

**Figure 3:** *Assessing data contamination (smaller is better). The radial axis (log scale) indicates the proportion of images from the evaluation dataset that exhibit similarity to a training sample beyond a given threshold according to CLIP. Human evaluations indicate that most instances captured at a threshold of* 0.98 *are problematic, and most problematic samples are identified at a threshold of* 0.96. *Except for MMMU and DudeMini, BigDocs-7.5M (blue/darkred) is less contaminated compared to DocStruct4M (red).*

### 3.2 BigDocs Toolkit: Data Preprocessing, Filtering, and Consolidation

The BigDocs Toolkit provides modular tools for preprocessing, filtering, contamination management, metadata management, and dataset loading. These components work in unison to streamline the integration of large-scale document datasets, ensuring quality and ease for efficient model training.

**✎ Datamaker Module.** The BigDocs Toolkit offers a modular framework for dataset curation, focusing on standardization, quality control, and metadata management. Its core *DataMaker* class acts as a template for handlers that extract annotations and convert raw data into a standardized format. A universal function processes tasks like OCR, VQA, and code generation, ensuring consistency. Bounding boxes are standardized, and corrupted samples are filtered. The Toolkit also generates metadata to enhance transparency, covering licensing and processing details (see Appendix A.9).

**🔗 Unified Metadata Framework.** We propose a unified metadata framework for BigDocs to ensure transparency and traceability. This framework thoroughly examines each raw data source, extracts fine-grained license information, and documents transformations applied to the data (e.g., different sources may have distinct licenses). Each data sample includes a metadata attribute detailing its properties, licenses, sources, and transformations (see Appendix A.9 for an example in Figure 13 and structure details). To our knowledge, this is the first systematic approach to track metadata for visually rich documents, advancing transparency in multimodal dataset curation.

**Assessing Contamination.** The presence of downstream evaluation data in training datasets can significantly affect the accurate measurement of a model's effectiveness (Magar & Schwartz, 2022). BigDocs-7.5M contains samples from the training split of TabFact, WTQ, and TextVQA. However, any evaluation dataset may *a priori* overlap with training datasets through less direct dependencies. We favor a transparency strategy: search for overlaps and report what was found. Figure 3 gives an overview of the main observations on this front; see Appendix A.3 for additional details.

#### 3.2.1 The resulting BigDocs-7.5M

BigDocs-7.5M consists of 7.5M image-text pairs for training (4M unique images), 500k for validation (234k unique images), and 470k for testing (261k unique images). These data points aggregate four document-related tasks – OCR, structured parsing, captioning, and question-answering – enabling models to handle diverse document use cases (see Figure 1 and Appendix A and A.1 for more details). Low-quality data is filtered out, and metadata details our best-effort licensing compliance, making the dataset suitable for training foundational models, even in commercial applications.

## 4 Building BigDocs-Bench

In the previous section, we introduced BigDocs-7.5M, a unified and permissive dataset for training models on document understanding tasks. Building on this, we present BigDocs-Bench, a benchmark suite for evaluating downstream tasks that transform visual inputs into structured outputs, such as

**Table 1:** *Statistics of the ten downstream tasks in BigDocs-Bench.* GPT2 tokenizer is used to produce token numbers of both queries and annotations (if any), where the format is (avg $\pm$ std).

| Downstream Task | | # Training | # Validation | # Public Test | # Hidden Test | # Text Tokens |
|---|---|---|---|---|---|---|
| </> | Screenshot2HTML | 9,338 | 1,000 | 500 | 500 | $32,700 \pm 53,105$ |
| X¹ | Table2LaTeX | 77,669 | 1,000 | 500 | 500 | $438 \pm 540$ |
|  | Image2SVG | 198,000 | 2,000 | 748 | 748 | $2,871 \pm 1,728$ |
|  | Image2Flow(GraphViz) | 8,000 | 1,000 | 500 | 500 | $418 \pm 124$ |
|  | Image2Flow (JSON) | 8,000 | 1,000 | 500 | 500 | $1,771 \pm 601$ |
| M↓ | Chart2Markdown | 4,516 | 1,000 | 500 | 500 | $1,559 \pm 4,442$ |
|  | Chart2Caption | 5,412 | 1,300 | 650 | 650 | $94 \pm 49$ |
|  | GUI2UserIntent | 79,000 | 1,000 | 500 | 500 | $28 \pm 4$ |
|  | GUI2Summary | 79,000 | 1,000 | 500 | 500 | $132 \pm 25$ |
|  | GUI-VQA | 78,991 | 1,000 | 500 | 500 | $35 \pm 24$ |

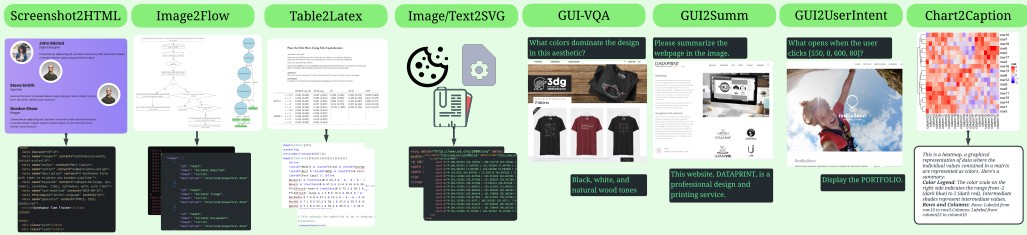

**Figure 4:** *8 of the new tasks introduced in BigDocs-Bench. These tasks share a focus on understanding the underlying structure of visually rich documents, with many also requiring generating lengthy outputs, such as SVG and HTML code. More tasks are shown in Figure 9 and 10.*

GUI2UserIntent (fine-grained reasoning), Image2Flow (structured output), Chart2Caption (understanding), and Screenshot2HTML (creative generation). BigDocs-Bench includes ten specialized tasks, with 329k training samples, 11k validation samples, 10k testing samples, and an additional hidden test set. Refer to Table 1 for task details and Figure 4 for examples.

## 4.1 BIGDOCS-BENCH TASKS SUITE

**</> Screenshot2HTML:** We introduce a benchmark for Screenshot2HTML conversion, with 10,838 real-world website screenshots paired with HTML code (see A.4.2). Curated from diverse, text-heavy websites in the FineWeb corpus (Penedo et al., 2024), it contrasts with synthetic sites from prior work (Laurençon et al., 2024b). Using `Playwright`, we retrieved, rendered, and filtered sites for accessibility, content, and licensing. External assets (e.g., CSS, fonts) were inlined, JavaScript removed, and images replaced to focus on structure. Screenshot2HTML evaluates the accuracy of HTML generated from webpage screenshots, emphasizing layout fidelity and semantic correctness.

**X¹ Table2LaTeX:** We propose a benchmark for Table2LaTeX conversion, consisting of 79,669 table images paired with original LaTeX code and captions (details in Appendix A.4.3). The dataset was curated by crawling arXiv papers with permissible licenses and extracting tables from PDFs and TeX source files(`https://arxiv.org/`). Instead of relying on imperfect PDF detection, tables were rendered from the LaTeX `.tex` code to ensure accurate visuals. For instance, an image of a table is paired with its corresponding LaTeX snippet code and caption. This benchmark supports precise table extraction and LaTeX generation evaluation from academic documents.

**Image2SVG:** We present a benchmark for Image2SVG conversion, curated from the existing SVG-Stack collection by StarVector (Rodriguez et al., 2023a) (details in Appendix A.4.4). The dataset includes 200k raster images paired with SVG code (e.g., flowchart image paired with SVG code replicating it) or descriptive text. We filtered for complex designs, using image entropy to exclude simple graphics, and ranked image-text pairs by CLIP Score (Hessel et al., 2022; Radford et al., 2019). This benchmark evaluates models on precise vector image reconstruction and scalability.

**Image2Flow:** We introduce two benchmarks, Image2Flow(GraphViz) and Image2Flow(JSON), mapping flowchart images to JSON or GraphViz code (Appendix A.4.1). The dataset includes 10,000 flowchart samples with GraphViz files generated using LLaMA 3.1 (Dubey et al., 2024) and JSON files detailing nodes and connections. Random colors and styles were applied for diversity.

Unlike FlowchartQA (Tannert et al., 2023), which focuses on QA over preprocessed flowcharts, this benchmark tests models' ability to extract structure from raw images.

**Chart2Markdown:** This dataset assesses the models' capabilities in extracting data values from chart images: given an image, produce a data table of the underlying data table in markdown format. To create this dataset, we crawled recent chart images from the Statista website (`https://www.statista.com/`), focusing on charts from 2023 and 2024 that were not used in prior datasets like UniChart (Masry et al., 2023) and ChartQA (Masry et al., 2022). Our dataset reflects the most up-to-date facts and trends and overlaps less with existing datasets and benchmarks. We collected 6,516 chart images with their data tables and human-written summaries.

**Chart2Caption:** We introduce a benchmark for Chart2Caption conversion, aiming to generate textual summaries from chart images. The dataset includes 6,516 samples, with charts sourced from Kaggle by running public analytics notebooks and extracting charts and code. Summaries were generated using multimodal InterVL2-26B model (Chen et al., 2023; 2024) based on a custom prompt (see Appendix A.4.5) and augmented with human-written summaries from the Chart2Markdown task. This benchmark evaluates models' abilities to interpret and summarize visual data representations.

**GUI2UserIntent:** This benchmark interprets user intent from GUI interactions, identifying elements linked to clicked bounding boxes (details in Appendix A.4.7). The dataset, repurposed from SeeClick (Cheng et al., 2024), includes 80,000 website screenshots with bounding box coordinates and corresponding user intents sourced from Common Crawl to capture user interactions effectively.

**GUI2Summary:** The GUI2Summary task generates descriptions of website screenshots, focusing on web layouts. We synthesized 80,000 summaries (under 100 words each) using InternVL2-8B (Chen et al., 2023) in a zero-shot setting (details in Appendix A.4.8). Each summary provides an overview of the main content, referencing key visual elements, layout, and color schemes.

**GUI-VQA:** The GUI-VQA task answers questions about website screenshots, focusing on content and elements. We generated 80k QA pairs using sentences from GUI2Summary and prompting LLaMA 3.1-8b (Dubey et al., 2024) in a zero-shot setting (details in Appendix A.4.9).

## 4.2 Filtering and Quality

**Filtering and Verification with BigDocs Toolkit.** To ensure a high-quality, open-access dataset, we employed an NSFW detector (e.g., Llama-Guard-3) and filtering tools to eliminate harmful content, corrupted images, misaligned annotations, and personally identifiable information (PII). The BigDocs Toolkit, adhering to ART principles, streamlined web crawling and filtering processes. This multi-layered approach, iteratively refined using typical errors identified during human verification, ensured a clean and reliable dataset. The test set underwent manual human verification, with at least two annotators per sample to ensure quality and accuracy. Sampling-based checks on the training split further validated the robustness of the filtering process, achieving a 99% pass rate (see A.13).

## 5 Training Multimodal Models on BigDocs

We trained several state-of-the-art multimodal models of varying sizes and architectures on BigDocs to assess its effectiveness for document-based continual pretraining and downstream finetuning. For comparison, we also trained the models on DocStruct4M, the closest alternative in terms of scale and document-oriented tasks. Additionally, we experiment on the training set from BigDocs-Bench that requires generating long structure outputs, e.g., valid code outputs.

We follow two stages of training: continual pretraining (CPT) and downstream finetuning (FT). The CPT stage involves training on large domain-specific datasets, such as BigDocs and DocStruct4M, learning general tasks like OCR, layout understanding, and captioning. For FT smaller datasets, such as DocDownStream and BigDocs-Bench, to focus on specific tasks like question answering or generating HTML from images. A previous stage of pretraining (PT) is typically performed for general multimodal alignment. In our framework, we do not perform this stage and rely on publicly available checkpoints. While pretraining (PT) is typically performed for general multimodal alignment, we rely on public checkpoints instead. In CPT, we train the image encoder and connector to align image features with the LLM. For FT, both the connector and LLM remain unfrozen. See Figure 11 in Appendix A.5 for more details.

**Table 2:** ***General Document Benchmarks***. *Models trained on {BigDocs-7.5M+DocDownstream} perform competitively across multimodal document benchmarks. We compare them to base checkpoints and those trained on {DocStruct4M+DocDownstream}. Results for instruction-tuned models (first row of each block; italic) are provided for reference only (unfair comparison). BigDocs models show consistent performance.*

| Model | DocVQA VAL | InfoVQA VAL | DeepForm TEST | KLC TEST | WTQ TEST | TabFact TEST | ChartQA TEST | TextVQA VAL | MMMU VAL | DudeMini TEST | SlideVQA-M TEST | TableVQA TEST | Avg. Score |
|---|---|---|---|---|---|---|---|---|---|---|---|---|---|
| *DocOwl1.5-8B (instruct)* | *80.73* | *49.94* | *68.84* | *37.99* | *38.87* | *79.67* | *68.56* | *68.91* | *33.67* | *34.64* | *31.62* | *52.60* | *53.84* |
| DocOwl1.5-8B (base) | 2.07 | 1.84 | 0.00 | 0.00 | 0.00 | 0.00 | 0.00 | 0.00 | 24.44 | 19.07 | 3.30 | 13.63 | 5.36 |
| DocOwl1.5-8B (base) + DocStruct4M | 75.99 | 46.88 | 62.77 | 35.21 | 32.86 | 71.56 | **68.36** | 65.08 | **33.67** | 29.00 | 27.03 | 46.27 | 49.56 |
| DocOwl1.5-8B (base) + BigDocs (Ours) | **78.70** | **47.62** | 64.39 | 36.93 | 35.69 | 72.65 | 65.80 | 67.30 | 32.33 | 32.55 | 29.60 | 49.03 | 51.05 |
| *Qwen2-VL-2B (instruct)* | *89.16* | *64.11* | *32.38* | *25.18* | *38.20* | *57.21* | *73.40* | *79.90* | *42.00* | *45.23* | *46.50* | *43.07* | *53.03* |
| Qwen2-VL-2B (base) | 7.26 | 0.78 | 0.00 | 0.00 | 0.00 | 0.00 | 0.00 | 1.14 | 34.89 | 28.43 | 14.55 | 0.00 | 7.25 |
| Qwen2-VL-2B (base) + DocStruct4M | 59.53 | **32.00** | **53.98** | **36.38** | 28.48 | 64.24 | 54.44 | 55.89 | 34.89 | **28.78** | **22.68** | 46.53 | 43.15 |
| Qwen2-VL-2B (base) + BigDocs (Ours) | 57.23 | 31.88 | 49.31 | 34.39 | **31.61** | **64.75** | **68.60** | **61.01** | **35.67** | 27.19 | 17.46 | **47.53** | **43.89** |
| *Phi3.5-Vision-4B (instruct)* | *86.00* | *56.20* | *10.47* | *7.49* | *17.18* | *30.43* | *82.16* | *73.12* | *46.00* | *37.20* | *30.93* | *70.70* | *45.66* |
| Phi3.5-Vision-4B + DocStruct4M | 86.76 | 68.90 | 70.12 | **37.83** | **51.30** | **82.12** | 79.76 | 68.60 | 44.11 | 35.52 | 31.90 | **69.17** | 60.51 |
| Phi3.5-Vision-4B + BigDocs (Ours) | **87.05** | **70.05** | **70.97** | 37.45 | 51.21 | 81.24 | **81.56** | **68.72** | 45.00 | 36.15 | 32.47 | 67.77 | **60.80** |
| *LLaVA-NeXT-7B (instruct)* | *63.51* | *30.90* | *1.30* | *5.35* | *20.06* | *52.83* | *52.12* | *65.10* | *38.89* | *17.94* | *7.46* | *32.87* | *32.36* |
| LLaVA-NeXT-7B + DocStruct4M | **60.95** | **26.14** | 39.78 | 28.34 | 25.90 | 67.72 | **61.20** | **52.25** | **25.78** | 21.70 | 15.33 | 27.03 | 37.68 |
| LLaVA-NeXT-7B + BigDocs (Ours) | 57.13 | 24.47 | **46.38** | **31.09** | **27.06** | **72.58** | 54.72 | 49.06 | 17.78 | **22.88** | **16.07** | **33.13** | **37.70** |

## 5.1 EXPERIMENTAL SETUP

**Baseline Models.** We selected DocOwl1.5-8B (Hu et al., 2024), Qwen2VL-2B (Bai et al., 2023a), Phi-3.5-Vision-4B (Abdin et al., 2024), and LLaVa-NeXT-7B (Li et al., 2024) for our training experiments. These models were chosen due to their focus on document-related tasks (DocOwl1.5), their openness regarding checkpoints (Qwen2VL, DocOwl1.5), and their state-of-the-art performance and task generalization capabilities (LLaVa-NeXT, Phi-3.5).

**Training Details.** We conduct all experiments using 8 nodes of 8 H100 GPUs, using Fully Sharded Data Parallel (FSDP) for distributed training. All experiments use a batch size of 256 and a learning rate of 2e-5, with AdamW as the optimizer. More training details are provided in Appendix A.5.

**Evaluation Benchmarks & Metrics.** In addition to the newly introduced BigDocs-Bench, we assess performance on well-known document-oriented benchmarks, termed as **General Document Benchmarks**. We select these benchmarks for their relevance and diverse range of document tasks. DocVQA (Mathew et al., 2021b), InfoVQA (Mathew et al., 2021a), DeepForm (Svetlichnaya, 2020), KLC (Stanisławek et al., 2021) (Kleister Benchmark for Key information extraction), WTQ (Pasupat & Liang, 2015) (Wikipedia Tables), TabFact (Chen et al., 2020), ChartQA (Masry et al., 2022), TextVQA (Singh et al., 2019), MMMU (Yue et al., 2024), DUDE (Landeghem et al., 2023), Slide-VQA (Tanaka et al., 2023), and TableVQA (Kim et al., 2024). We utilize (and extended) VLM Eval Kit (Duan et al., 2024).

In **BigDocs-Bench**, we employ the following evaluation methods based on each task's characteristics. For Screenshot2HTML, inspired by related works in this domain (Reis et al., 2004), we compute Tree Edit Distance (TE Dist.) between the Document Object Model (DOM) of ground truth and generation. For Table2LaTeX, we report TeXBLEU (Jung et al., 2024). For Image2SVG, we compare the cosine similarity between the DINOv2 (Oquab et al., 2023) representations of the ground-truth and generated SVG images (DINOScore). For the Image2Flow tasks, we propose the Length-Shape Triplet F1 (**LST F1**) score between ground-truth and generated flowcharts' edge sets. More specifically, each edge is represented as a $(s, e, d)$ triplet, where $e$ is the edge label, and $s$ and $t$ are the source and destination nodes' labels concatenated with their shapes, respectively. For Chart2Markdown, we adopt the RMSF1 metric for markdown tables (Liu et al., 2022; Masry et al., 2023; 2024). For summarization and VQA tasks, we report Rouge-L F1 score (Lin, 2004). For more details about the evaluation process, please refer to Appendix A.6.

**Setup.** For our CPT on DocOwl1.5-8B and Qwen2VL-2B, we used base weights, while on Phi-3.5-Vision-4B and Llava-NeXT-7B, we initialized from their instruction-tuned versions, since their base weights are not publicly available. We first trained each model on a CPT corpus, either DocStruct4M (Hu et al., 2024) or BigDocs-7.5M, for one epoch. Following this, we performed further alignment (finetuning) using DocDownStream to enhance the model's ability to follow instructions. For each selected model, we also evaluated the author-provided base model and the instruction-tuned version (separate from the base checkpoint, if available) as baselines and reported the performance on general document benchmarks.

**Table 3:** *Comparison of model performance in BigDocs-Bench. BigDocs models trained on {BigDocs-7.5M+DocDownstream+BigDocs-Bench train-split}, which combine CPT and FT, outperform all baselines in tasks requiring long-format code generation, particularly in flow generation, GUI reasoning, and image-to-LaTeX generation, surpassing even state-of-the-art closed models. The symbol † denotes that the model required 1-shot prompts as opposed to the default 0-shot prompts.*

| Model | Chart2MD RMSFI | Chart2Cap. ROUGE-L-F1 | Image2Flow (GraphViz) LST-F1 | Image2Flow (JSON) LST-F1 | GUI2Sum. ROUGE-L-F1 | GUI2Intent ROUGE-L-F1 | Image2SVG DINO Score | Screenshot2HTML DOM/TEDist. | Table2Latex TeXBleu | GUI-VQA ROUGE-L-F1 | Avg. Score |
|---|---|---|---|---|---|---|---|---|---|---|---|
| *Open Models* | | | | | | | | | | | |
| DocOwl-1.5-8B | 0.08 | 18.69 | 0.00† | 0.00† | 11.22 | 13.88 | 3.58 | 3.50 | 75.07 | 27.22 | 15.32 |
| Qwen2-VL-2B | 41.17 | 22.88 | 0.00† | 0.00† | 23.98 | 17.70 | 23.18 | 6.46 | 74.83 | 26.40 | 23.66 |
| Phi3.5-V-4B | 60.64 | 21.88 | 1.61† | 0.65† | 27.80 | 10.81 | 34.57 | 4.25 | 74.14 | 34.96 | 27.13 |
| LLaVA-NeXT-7B | 22.00 | 20.67 | 1.58† | 0.46† | 21.99 | 12.38 | 20.53 | 5.00 | 73.81 | 27.54 | 20.60 |
| Idefics2-8B | 25.34 | 20.95 | 1.17† | 0.00† | 8.75 | 5.06 | 37.73 | 3.56 | 74.50 | 27.76 | 20.48 |
| Llama-3.2.90B | 45.21 | 20.60 | 0.73† | 0.52† | 22.16 | 12.04 | 45.97 | 7.32 | 74.79 | 27.28 | 25.66 |
| Qwen2-VL-72B | 70.47 | 19.42 | 1.07† | 0.23† | 18.80 | 33.94 | 54.43 | 10.03 | 74.51 | 30.67 | 31.36 |
| *Closed Models* | | | | | | | | | | | |
| GPT-4o 20240806 | 66.70 | 25.23 | 22.66† | 27.28† | 27.12 | 17.57 | 60.34 | 10.33 | 74.65 | 36.58 | 36.84 |
| Claude-3.5 Sonnet | 54.81 | 23.59 | 13.92† | 37.46† | 26.45 | 13.12 | 25.46 | 9.70 | 74.44 | 26.58 | 30.55 |
| GeminiPro-1.5 | 76.63 | 25.90 | 11.51† | 33.59† | 25.54 | 16.79 | 15.21 | 7.43 | 75.22 | 35.35 | 32.32 |
| *BigDocs Models (ours)* | | | | | | | | | | | |
| DocOwl-1.5-8B + BigDocs | 74.43 | 33.38 | 42.16 | 48.54 | 45.55 | **89.15** | 33.66 | 3.64 | 81.28 | 43.46 | 49.52 |
| Qwen2-VL-2B + BigDocs | 72.25 | 33.74 | 41.61 | 52.11 | 42.59 | 71.65 | 33.51 | 9.20 | 78.54 | 33.97 | 46.92 |
| LLaVA-NeXT-7B+ BigDocs | 72.78 | 32.88 | 59.66 | 71.49 | 46.14 | 79.55 | **60.63** | 10.40 | 80.79 | 40.67 | 55.50 |
| Phi3.5-v-4B + BigDocs | **84.01** | **36.78** | **63.07** | **71.86** | **47.32** | 86.91 | 34.65 | **12.05** | **81.94** | **44.81** | **56.34** |

We also provide a comprehensive evaluation on the proposed BigDocs-Bench. We conducted an off-the-shelf performance analysis on BigDocs-7.5M using models such as GPT4 (Achiam et al., 2023), Claude (Anthropic, 2024), Gemini Pro (Team et al., 2024a) and Qwen2VL-72B (Wang et al., 2024a) and Idefics2 (Laurençon et al., 2024a). In addition, we also evaluated the previously selected models on their instruction versions, including DocOwl1.5-8B, Qwen2VL-2B, Phi-3.5-Vision-4B, and Llava-NeXT-7B. To incorporate the new capabilities introduced in BigDocs-Bench, we further finetuned these models after BigDocs CPT using the training set from BigDocs-Bench (see Table 3). For each model, we only evaluate the instruction-tuned version (where available) as baselines, reporting their respective performance.

## 5.2 QUANTITATIVE RESULTS

**Results on Existing Document Downstream Tasks.** Table 2 presents the models performance across general document benchmarks. Base models perform poorly, mainly due to their inability to follow user instructions, like answering questions. Phi3.5 Vision, originally optimized for reasoning tasks, achieves an average score of 60.80% when finetuned on BigDocs, enhancing its ability to handle complex document-based tasks. For Qwen2-VL, an interesting pattern emerges: while the instruction-tuned version excels on tasks reported in its technical report (e.g., DocVQA, InfoVQA, ChartQA, MMMU), the BigDocs-trained model surpasses it on new tasks like TabFact, DeepForm, KLC, and TableVQA, suggesting that Qwen2-VL's instruction-tuning may rely on complex prompt engineering and task-specific optimizations. *We find that performing additional continual pretraining and finetuning on instruction-tuned models does not significantly degrade performance.* Moreover, **we observe substantial improvements on previously underperforming tasks**. As shown in Table 2, Phi3.5 and LLaVA-Next show marked gains on DeepForm, KLC, WTQ, and SlideVQA. However, LLaVA's performance declined on MMMU, indicating the need for more multiple-choice question data. This reinforces our argument that transparency in training datasets is essential for proper evaluation. Finally, across all models, BigDocs training yields higher average scores compared to training on DocStruct4M, even with lower contamination rates on most benchmarks, as we highlighted in Section 3. These findings indicate that **BigDocs supports better generalization and robustness without complex task-specific optimizations while also being license-permissive.**

**Results on BigDocs-Bench Tasks.** Table 3 shows results for our proposed downstream tasks, evaluating models' ability to generate lengthy structured and valid code outputs, and reasoning from GUIs. Overall, BigDocs models consistently outperform both open and closed models on most tasks, particularly in Flow and GUI tasks such as GUI2UserIntent, GUI2Summary, GUI-VQA, and Image2Flow, revealing areas where existing models fall short. The performance gap is narrower on tasks like Screenshot2HTML, Image2SVG and Chart2Caption, suggesting these tasks have been explored in the literature but not extensively enough. Phi3.5-V4B + BigDocs stands out as the

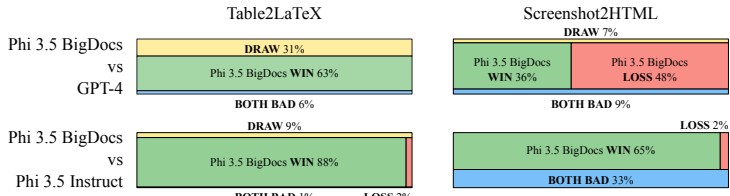

**Figure 5:** *Human evaluation results comparing Phi3.5 BigDocs-Bench against Phi3.5 Instruct and GPT-4o on two tasks: Table2LaTex (Left) and Screenshot2HTML (Right).*

top performer in 8 tasks out of 10, with an average score of 50.46. However, its performance on Image2SVG (25.98 points behind LLaVA-NeXT-7B + BigDocs) indicates less exposure to SVG data in its instruction tuning. We find that the model can generate valid code outputs in different formats, including SVG, HTML, JSON, or Latex, when conditioned on images.

### 5.3 Human Evaluations and Qualitative Results

We conducted a human evaluation comparing the performance of **Phi-BigDocs**, **Phi-Instruct**, and **GPT-4o** on **Screenshot2HTML** and **Table2LaTeX** tasks. Twenty-eight evaluators participated, providing 1,900 annotations. For Table2LaTeX, evaluators assessed if the LaTeX table matched the input table. For Screenshot2HTML, they evaluated the visual similarity between the rendered HTML and the screenshot. Evaluators chose between four options: one of the two model outputs is superior to the alternative (recorded as "WIN" if it is **Phi-BigDocs** and "LOSS" otherwise), both outputs are acceptable but of very similar quality ("DRAW") and neither output is acceptable ("BOTH BAD"). See Appendix A.8 for more details on the evaluation platform.

From human evaluation results in Figure 5, for the Table2LaTeX task, Phi3.5 BigDocs wins 88% of the time against Phi3.5 Instruct and achieves a 63% win rate, with a 31% draw rate, against GPT-4o. These results highlight our model's ability to accurately preserve the table's structure, including lines, borders, and margins, whereas GPT-4o often struggles to maintain consistent formatting despite capturing content accurately. In the Screenshot2HTML task, Phi3.5 BigDocs achieves a 65% win rate against Phi3.5 Instruct and performs competitively against GPT-4o, with a 36% win rate and 7% draw rate, demonstrating its strong capability to reproduce visual elements faithfully.

**Qualitative Results.** We provide qualitative results in Appendix A.10. Figure 8 presents outputs from experiments with the Phi-3.5-Vision-4B model on BigDocs-Bench for tasks like Chart2Markdown, Table2LaTeX, and Image2SVG. The model delivers visually consistent outputs and generates valid code across formats, adhering to task instructions. Table 6 compares sample outputs between Phi-3.5-Vision models and GPT-4o, highlighting the strong performance of our BigDocs trained version in captioning and VQA tasks.

## 6 Conclusion

We introduce BigDocs-7.5M, a large-scale, license-permissive dataset for training multimodal models on document and code-related tasks. Along with a comprehensive suite of tools and data analysis, we present BigDocs-Bench, featuring 10 downstream tasks that assess a model's ability to generate long-format code outputs from images. These tasks serve as practical benchmarks for real-world applications. Our experiments show that models trained on BigDocs outperform those trained on existing datasets. Furthermore, training on the BigDocs-Bench train split endows the resulting models with new capabilities and significantly enhances their ability to generate long, structured outputs. All BigDocs artifacts will be freely available under permissive licenses.

**Limitations** Our work presents a pioneering license-permissive dataset for multimodal document understanding, achieving strong performance across tasks. However, there are limitations: (1) Suboptimal performance on some public benchmarks, indicating a need to refine the data mixture and explore additional sources. (2) Limited context length, as models are trained with a maximum of 8192 tokens, restricting performance on tasks with long, structured outputs like HTML and SVG. (3) Uncertainty in the commercial viability of base models, as their pretraining data lacks transparency.

## ETHICS AND REPRODUCIBILITY STATEMENT

**Ethics Statement** Our work is centered around responsible and transparent curation of datasets for multimodal document understanding models. While we have made extensive efforts to filter harmful content from our dataset, we cannot fully guarantee that the models will not generate offensive language. Additionally, we have taken significant steps to remove personally identifiable information (PII) from the compiled datasets to protect user privacy. However, we cannot ensure that the generated code will be free of malicious snippets, and developers are encouraged to implement protection protocols to safeguard against potential risks. Finally, all human evaluation studies were conducted by collaborator researchers, and no PII was collected during this process.

**Reproducibility Statement** We are committed to ensuring the reproducibility of our work by providing all necessary details and resources. All artifacts, including code, datasets, model weights, data sheets, and metadata, will be publicly released. Furthermore, we have fully documented all hyperparameters, experimental setups, and evaluation metrics to allow for accurate replication of our results. For human evaluation, we provide clear instructions and describe the environment used for comparison to ensure transparency and consistency.

## ACKNOWLEDGMENTS

We thank Aishwarya Agarwal, Ghazwa Darwiche, Christian Hudon, Tom Murray, and Chao Wang for their precious administrative and/or technical assistance. Yoshua Bengio thanks Microsoft and Samsung for funding.

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

# A APPENDIX

## A.1 BIGDOCS-7.5M TASKS

1. **OCR-Related Tasks**: These tasks involve converting images of text (e.g., scanned documents) into machine-readable formats, including transformations such as bounding-box-to-text and text-to-bounding-box (text localization as described in Hu et al. (2024)). Models learn to recognize and map textual information within images.

2. **Structured Parsing and Extraction**: This task focuses on extracting and transforming structured data from documents, like parsing tables, forms, and charts into formats such as JSON or Markdown. It includes handling documents with complex visual layouts and sometimes incorporates bounding box information for individual elements.

3. **Captioning and Summarization**: This task requires models to generate captions or summaries for visual or textual content, such as figures, charts, or document sections. The models provide concise descriptions or overviews, enhancing document comprehension.

4. **Question-Answering (QA)**: QA tasks involve responding to questions posed over structured or visual data (e.g., tables, figures). This includes multi-turn QA, where models address a series of related questions to improve their reasoning and comprehension.

## A.2 DATASETS INCLUDED IN BIGDOCS-7.5M

The following datasets are utilized in our work, supporting the four core tasks mentioned in Section 3.2.1. We reference the task supported by each dataset via the index in Section A.1. Note that datasets with * are those fully or partially curated by us (details explained in the description of each of them).

1. **TabFact** (Chen et al., 2020): [Task included: (2), (4)] TabFact is used for question-answering tasks, where models check whether a given statement is supported or refuted by a table. It also involves structured parsing, helping models extract and process structured table data into formats like Markdown.

2. **Open4Business (O4B)** (Singh et al., 2020): [Task included: (2), (3)] O4B is a dataset focused on business-related documents and is processed for structured parsing and extraction, as well as captioning and summarization, allowing models to retrieve key insights from documents and generate summaries or descriptions.

3. **WikiTQ** (Pasupat & Liang, 2015): [Task included: (2), (4)] WikiTableQuestions is used for question-answering tasks over tables, where models answer questions based on table data, and structured parsing for converting table data into Markdown.

4. **CORD** (Park et al., 2019): [Task included: (1), (2)] CORD is a dataset for parsing receipts, used for both OCR-related tasks and structured parsing and extraction, helping models interpret structured financial data from document layouts. This latter task requires extracting entities and providing their text, category, and location as a JSON output.

5. **UniChart** (Masry et al., 2023): [Task included: (2), (4)] UniChart is used for structured parsing and extraction to extract structured information from chart-like tables, converting complex visual layouts into formats like JSON or Markdown. And also for question-answering tasks, where models answer questions related to the chart content.

6. **TextOCR** (Sidorov et al., 2020): [Task included: (1)] TextOCR is processed for OCR-related tasks, enabling models to perform bounding-box-to-text transformations on scene text from images.

7. **COCO-Text** (Veit et al., 2016): [Task included: (1)] COCO-Text is used for OCR-related tasks, helping models extract and recognize text from real-world images with natural scene text.

8. **CDIP-1M**[*]: [Task included: (1), (2)] To create this dataset, we sourced from the original IIT-CDIP (Soboro, 2022) dataset and curated it into CDIP-1M. CDIP-1M is processed for OCR-related tasks and structured parsing, focusing on extracting text and structure from large-scale scanned document collections. We used an in-house OCR engine to get the text (i.e. annotations) from its 11M documents from the source. Like the text localization task in DocStruct4M Hu et al. (2024), we generate word-, line- and block-level bounding-box-to-text and text-to-bounding-box QA pairs. We subsample zones randomly, and based on OCR confidence, a large fraction of the images are pretty noisy. In addition, for the block level, we generate structured parsing QA pairs where text lines and their location need to be given as JSON by the model.

9. **PubTables-1M** (Smock et al., 2022): [Task included: (1), (2)] PubTables-1M is a large dataset of tables from scientific papers. It is used for **OCR-related tasks** to extract information in tables. It is also processed for **structured parsing and extraction**, allowing models to handle scientific tables and convert them into markdown.

10. **FigureQA** (Kahou et al., 2017): [Task included: (4)] FigureQA is focused on question-answering tasks, where models answer questions based on charts and figures, improving reasoning over visual and tabular data.

11. **DocBank** (Li et al., 2020b): [Task included: (2), (4)] DocBank is utilized for structured parsing and extraction and question-answering tasks, enabling models to interact with scholarly documents and extract structured data from layouts.

12. **TableBank** (Li et al., 2020a): [Task included: (2)] TableBank is a large dataset used for structured parsing and extraction, helping models parse table structures from both Word and LaTeX documents into structured formats.

13. **OCRVQA** (Mishra et al., 2019): [Task included: (1),(4)] OCRVQA focuses on OCR-related tasks and question-answering tasks over OCR-extracted text, where models answer questions based on text and visual data from real-world scenes.

14. **Datikz** (Belouadi et al., 2023): [Task included: (2), (3)] Datikz is processed for captioning and summarization as well as structured parsing, enabling models to describe and interpret complex diagrams and generate structured data from them.

15. **ArxivOCR**[*]: [Task included: (1)] The ArxivOCR dataset contains OCR-scanned academic papers and is used for OCR-related tasks, where models perform bounding-box-to-text transformations, improving accessibility and structure for scholarly articles. This dataset is curated by us. We filter out the papers from Arxiv that have permissive licenses, i.e. CC-BY 4.0 and CC0 in this case. Then we use in-house OCR engines to produce OCR results on the pages from the papers collected.

16. **ArxivTableCap**[*]: [Task included: (3)] The ArxivTable dataset focuses on generating captions for the tables and figures extracted from Arxiv papers, helping models describe the content and context of tables in academic papers. Among the 156.2k samples, 70k of them are from AFTdb Arkea (2024). We perform the filtering to make sure all the selected ones are in papers with permissive licenses, i.e. CC-BY 4.0 and CC0 in this case.

17. **SVGCap Dataset** (Rodriguez et al., 2023a): [Task included: (3)] The SVG Dataset supports captioning and summarization tasks, where models generate descriptive captions for SVG content, summarizing the structure and elements of vector graphics.

## A.3 Assessing Contamination

BigDocs-7.5M's direct dependency on TabFact, WTQ, and TextVQA is restricted to their training splits, and DocStruct4M reports a similar dependency on DocVQA, InfoVQA, DeepForm, KLC, ChartQA, TabFact, WTQ, and TextVQA. However, overlaps between either of these training sets and evaluation datasets may emerge through indirect means (e.g., the same source material was involved).

Our primary "automatic" approach to assess contamination consists of embedding all images from the reference dataset (i.e., BigDocs-7.5M or DocStruct4M) using a pretrained CLIP (Radford et al., 2021), namely `clip-ViT-B-32` from `sentence-transformers` (Reimers, 2019). We similarly embed each image from an evaluation dataset and retrieve the reference image with the highest cosine similarity: the closer this measure is to $1.0$, the more likely it is that the evaluation image is part of the reference dataset. Figures 3 and 6, as well as table 4 all report these values.

Except when the cosine similarity is exactly $1.0$ – indicating an exact match – interpretations of these scores must be grounded in human evaluations. However, assessing whether two images are "the same" can be a non-trivial task, even for human eyes.

Consider the following edge cases:

- receipts for recurring orders emitted on different days;
- the same form filed by different people;
- different versions of the same form at the same company; and
- different full-page table from the same appendix of the same report.

Technically, all these cases involve pairs of different documents. However, one could argue that training on one such samples confers an unfair advantage to a model evaluated on the other sample. For this reason, we annotate such instances as *same template* whenever we encounter them.

Conversely, the "actual" same document can appear as quite diversified images. Indeed, stamps, watermarks, censorship, and/or annotations may be apposed *a posteriori* to a document, in addition to scaling, crops, rotations, and fax-induced artifacts. In all these cases, the original intent to communicate the same information matters: different copies of the same memorandum, scanned separately, are here treated as the *same document*.

With these definitions in hand, a human is tasked with labeling the samples that are most likely to be problematic in the DocVQA evaluation dataset. Figure 7 reports the results, calibrating how we interpret the cosine similarities for other evaluation datasets.

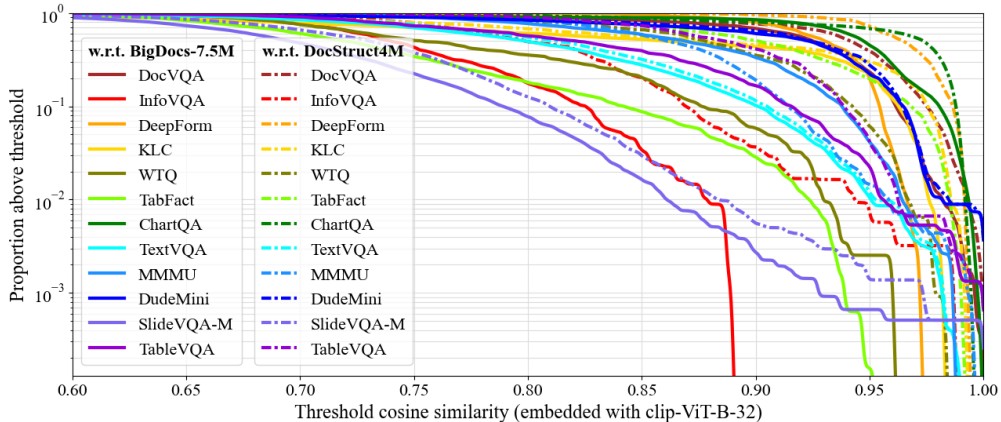

**(a)** *InfoVQA, SlideVQA-M, and TabFact overlap very little with BigDocs-7.5M (train).*

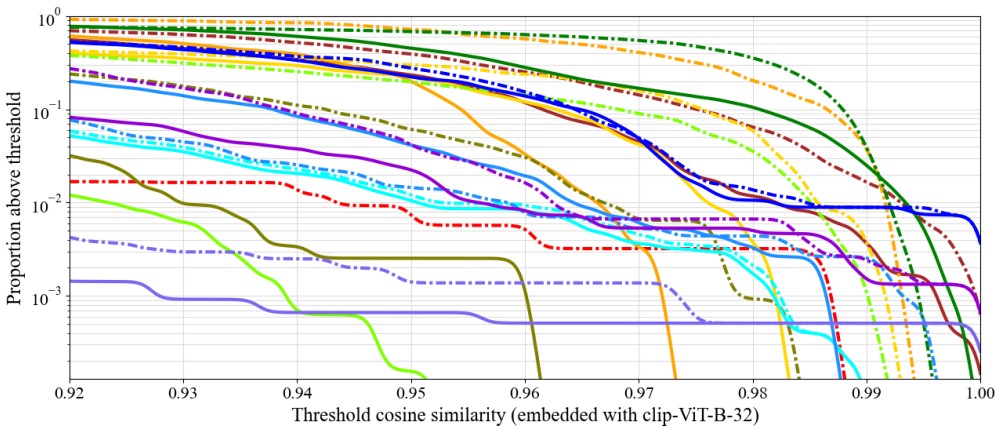

**(b)** *Zoom on the above (same legend).*

**Figure 6:** *Cumulative distribution by cosine similarity. Each curve shows the proportion of samples in an evaluation dataset for which there exists at least one sample in the reference dataset with cosine similarity higher than the specified threshold. Lower values are better; higher cosine similarity thresholds are more pertinent. Based on human evaluations, samples with cosine similarity < 0.96 are unlikely to be problematic, whereas those > 0.98 are most likely problematic. Except for MMMU, and DudeMini, BigDocs-7.5M appears less contaminated than DocStruct4M.*

**Table 4:** *Detailed Results on Contamination Experiments, related to Figure 6. Lower values are better; higher cosine similarity thresholds are more pertinent. Except for MMMU and DudeMini, BigDocs-7.5M appears to be less contaminated by these metrics than DocStruct4M. Figure 3's data comes from this table.*

| Cosine Similarity Threshold | Reference Dataset | DocVQA | InfoVQA | DeepForm | KLC | WTQ | TabFact | ChartQA | TextVQA | MMMU | DudeMini | SlideVQA-M | TableVQA |
|---|---|---|---|---|---|---|---|---|---|---|---|---|
| 0.99 | DocStruct4M | 0.016 | 0.0 | 0.037 | 0.0033 | 0.0 | 0.00039 | 0.034 | 0.0 | 0.0026 | 0.0089 | 0.0 | 0.0027 |
| | BigDocs-7.5M | **0.0036** | 0.0 | **0.0** | **0.0** | 0.0 | **0.0** | **0.025** | 0.0 | **0.0** | 0.0089 | 0.0 | **0.0** |
| 0.98 | DocStruct4M | 0.065 | 0.0032 | 0.20 | 0.061 | 0.0 | 0.036 | 0.36 | 0.0020 | 0.0044 | 0.013 | 0.0 | 0.0067 |
| | BigDocs-7.5M | **0.012** | **0.0** | **0.0** | **0.0033** | 0.0 | **0.0** | **0.10** | **0.0** | **0.0035** | **0.011** | 0.0 | **0.0053** |
| 0.97 | DocStruct4M | 0.14 | 0.0032 | 0.41 | 0.16 | 0.0064 | 0.089 | 0.55 | 0.0044 | **0.0053** | **0.048** | 0.0014 | 0.0067 |
| | BigDocs-7.5M | **0.049** | **0.0** | **0.0** | **0.043** | 0.0 | **0.0** | **0.17** | **0.0** | 0.0061 | 0.053 | **0.0** | **0.0053** |
| 0.96 | DocStruct4M | 0.26 | 0.0057 | 0.57 | 0.24 | 0.031 | 0.14 | 0.63 | 0.0096 | **0.0070** | 0.15 | 0.0014 | 0.017 |
| | BigDocs-7.5M | **0.12** | **0.0** | **0.033** | **0.12** | 0.0 | **0.0** | **0.28** | **0.0** | 0.021 | **0.14** | 0.0 | **0.0087** |
| 0.95 | DocStruct4M | 0.39 | 0.0057 | 0.72 | 0.30 | 0.062 | 0.20 | 0.68 | 0.012 | 0.015 | 0.28 | 0.0014 | 0.04 |
| | BigDocs-7.5M | **0.24** | **0.0** | **0.21** | **0.22** | 0.0 | **0.0** | **0.45** | **0.0** | 0.045 | **0.22** | **0.0012** | **0.023** |
| Number of samples | | 5349 | 2801 | 1500 | 4872 | 4343 | 12722 | 2500 | 5000 | 1140 | 5275 | 19600 | 1500 |
| Number of unique images | | 1284 | 500 | 300 | 608 | 421 | 1693 | 1509 | 3166 | 1100 | 609 | 3596 | 751 |

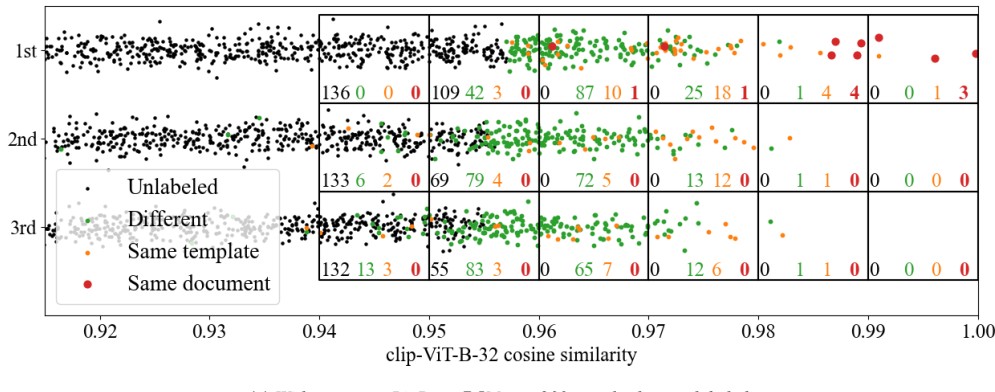

(a) *With respect to BigDocs-7.5M; top 200 samples human-labeled.*

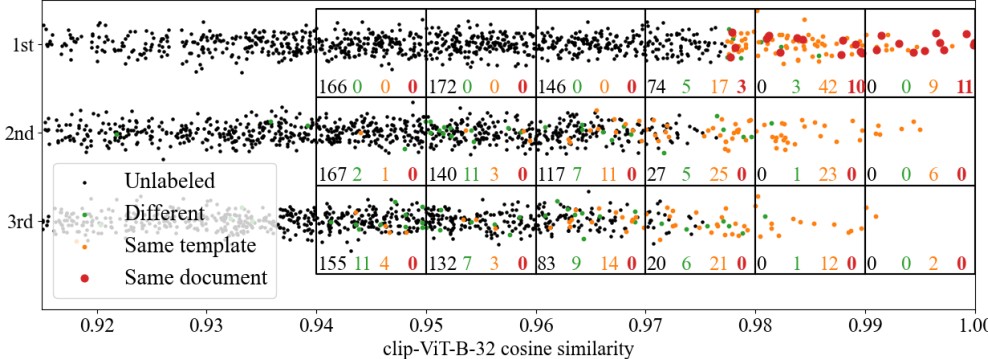

(b) *With respect to DocStruct4M; top 100 samples human-labeled*

**Figure 7:** *Human evaluation of DocVQA's overlap. Among the 1284 unique images in DocVQA's samples, we use the same cosine similarity method as in figure 6 to identify the samples that are the most similar to samples in the corresponding training set. Although we prioritize using only the closest match, we also retrieve the second and third closest matches after deduplication (i.e., if the next closest match is identical to the previous match, skip it). A human is then tasked to label the top matches as either "different", "same template" or "same document" (see text for definitions). Counts are provided for the most relevant 0.01-wide cosine similarity intervals. Most samples below 0.96 are "different", and most samples above 0.98 are not. From these numbers, we expect less than 10% of the samples with cosine similarity below 0.97 to be "same template", and less than 1% of the samples with cosine similarity below the same threshold to be "same document". All identified "same document" are found at the first rank. There is only one instance where the first rank is "different" but a "same template" is identified at higher rank.*

Note that this labeling procedure does not alter the composition of BigDocs-7.5M: we leave all samples, problematic or not, in the dataset. Instead, we release the annotations, which may help the community develop an intuition of the overlaps that may not have been identified yet, or even enable better detection strategies in the future.

## A.4    DESCRIPTION OF DOWNSTREAM TASKS PROPOSED IN BIGDOCS-BENCH

The following is a formal description of the downstream tasks we aim to solve using the proposed BigDocs-Bench dataset.

### A.4.1    IMAGE2FLOW

The task at hand is an image-to-flow conversion, where the input is an image of a flowchart, and the output is the corresponding information in JSON format. This JSON includes the nodes and edges that represent the flowchart's structure.

Formally, given an image $I$ representing a flowchart, the goal is to extract a graph $G = (V, E)$ where $V$ is the set of nodes, and $E$ is the set of directed edges between these nodes. The output $JSON(G)$ contains two main components: (1) a list of nodes $V$ with their corresponding attributes such as node

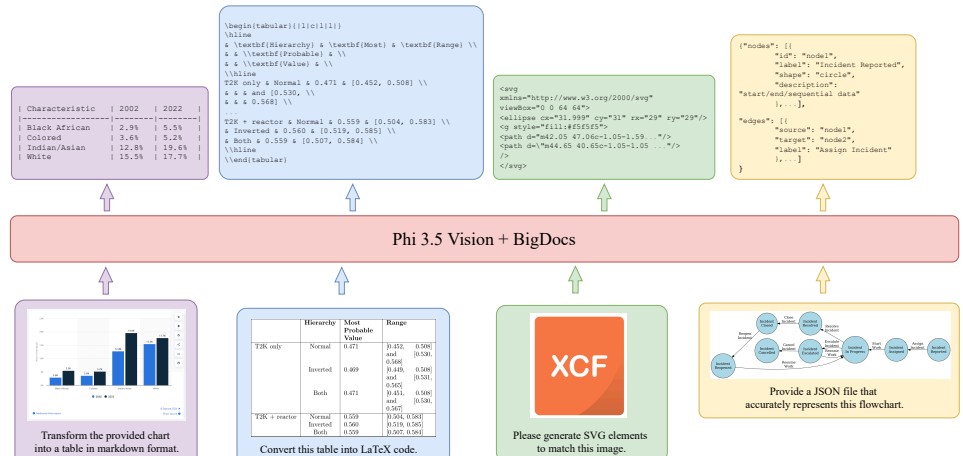

**Figure 8:** *Qualitative Results. Generations of our Phi3.5-Vision model on the presented document downstream tasks, as part of the test set of BigDocs-Bench. We show samples of Chart2Markdown, Table2LaTex, Image2SVG, and Image2Flow (JSON). A single model trained on BigDocs datasets can generate image-conditioned code in different coding languages while providing valid outputs.*

ID, label, shape, and description, and (2) a list of edges $E$ with their source node ID, destination node ID, and label (if any).

The dataset for this task consists of 10,000 samples. Each sample includes an image of a flowchart, its corresponding Graphviz file, and a JSON representation.

The dataset was generated using the following pipeline: First, a random number between 5 and 15 was generated to determine the number of nodes in each flowchart. Based on the total number of nodes, a distribution was assigned to different types of nodes, such as conditions (diamond shape), statements (rectangular nodes), and others (parallelogram, circle). A random flow direction was then selected from the options: BT (bottom to top), TB (top to bottom), LR (left to right), and RL (right to left). Using the LLaMA 3.1 model, we used the prompt in Table A.4.1 to generate a Graphviz file.

> **Prompt used with Llama 3.1 to generate Graphviz data**
>
> ```
> Create a directed flowchart graph in DOT format with the
> following specifications:
> - The direction of the graph should be {direction}.
> - Total number of nodes:  {total_nodes}.
> - Nodes distribution:  {nodes}.
> - The edges should connect the nodes in a way that makes sense:
> * Each node should have at least one outgoing edge.
> * For 'diamond' nodes, there should be at least two outgoing
> edges.  The graph can include any colors or additional styling.
> Generate a valid and coherent graph.  The graph should
> represent an enterprise-level workflow like the steps of an
> incident resolution workflow.  The enterprise-level workflow
> labels are so important.  Just give me the graph in Graphviz
> format no need for any python code or any description about the
> graph.
> ```

Random colors and styles were applied to remove model biases toward specific stylings. The generated Graphviz files (.gv) were then converted to PNG images and JSON files. The JSON files contain two main components: Each node has an ID, label, shape, and a description of the shape (e.g., diamonds represent conditions). Each edge contains the source and destination node IDs and a label if present (the label represents the text on the edge).

### A.4.2  SCREENSHOT2HTML

The Screenshot-to-HTML conversion task involves transforming an image of a website's layout, such as a screenshot, into the corresponding HTML code that accurately reconstructs the structure and content of the original website. This process enables the generation of a functional website solely from its visual representation, facilitating applications in web design automation, accessibility enhancements, and rapid prototyping.

Formally, given an input image $I$ depicting the visual layout of a website, the objective is to generate the HTML structure $H$ that includes essential web elements such as headers, paragraphs, images, links, forms, and navigation bars. The resulting HTML code $\text{HTML}(H)$ should not only replicate the visual appearance of the original website but also ensure semantic correctness and structural integrity, enabling the recreated website to be interactive and accessible.

**Data Collection and Filtering Process.** The Image-to-HTML dataset was curated through an automated pipeline designed to ensure the quality and diversity of website layouts. Utilizing the `Playwright` library, the system asynchronously retrieves and renders websites from a comprehensive list of URLs (Penedo et al., 2024). Each website undergoes a series of checks to verify its accessibility, compliance with `robots.txt` directives, and predominance of English content. Additionally, content appropriateness is assessed by filtering out websites containing NSFW language.

External CSS and JavaScript resources are inlined to maintain consistency and reduce dependencies, and unnecessary or oversized scripts are removed. Images are replaced with placeholders, and background images are eliminated to focus on structural elements. The viewport is adjusted to capture only the visible portion of each page, enhancing the clarity of the layout representations.

Websites are further evaluated based on performance and structural metrics, including load time, page size, number of JavaScript and CSS files, DOM depth, and total number of HTML elements. Technologies and frameworks used by each website are identified to exclude those utilizing disallowed technologies, e.g., allowing a website to render without JavaScript. We also removed comments and prettified the HTML for standardization. Only websites that meet all predefined criteria are included in the final dataset.

**Dataset Statistics.** The resulting dataset comprises 11,000 website samples, each with a high-resolution screenshot and its corresponding HTML representation. On average, each layout contains 20.3 HTML elements, reflecting a diverse range of website designs. This diversity provides a robust foundation for training and evaluating image-to-HTML conversion models, ensuring the dataset is both comprehensive and representative of various web structures.

### A.4.3  TABLE2LATEX

The task involves identifying and associating tables in academic PDFs with their corresponding LaTeX code and captions. The aim is to precisely match each table image with the LaTeX source used to generate it and the relevant caption, ensuring accurate alignment between the visual content and its textual description.

More specifically, given a table image $I$, the LaTeX code $C$ used to render the table, and the caption $T_{\text{caption}}$, the goal is to create a dataset $\text{Dataset}(I, C, T_{\text{caption}})$ that establishes a reliable association between the tables, their LaTeX source, and their descriptive captions.

We crawled publicly available, license-compliant arXiv papers to create this dataset, collecting both their PDFs and associated TeX source files. We began by parsing the TeX files to extract the LaTeX code for tables and their captions. Next, we used the PDF parsing library `PyMuPDF` to locate tables within the PDFs. However, this table detection process proved to be imperfect, as the algorithm frequently misidentified content with parallel lines as tables, leading to false positives.

To address this challenge, we adopted an alternative approach. Instead of cropping tables directly from the PDFs—where false positives were common—we chose to render the parsed LaTeX table code to generate accurate table images. This method ensured that the images faithfully represented the original table formatting, reducing detection errors and improving the reliability of the dataset. As a result, we created a high-quality dataset comprising over 95,000 table images, each paired with

its corresponding LaTeX code and caption, providing a valuable resource for further research into table structures in academic papers.

### A.4.4 IMAGE2SVG & TEXT2SVG

Scalable Vector Graphics (SVG) provide a precise alternative to pixel-based images, capable of representing diagrams, icons, plots, and graphic designs with superior detail and scalability. Unlike raster images, SVGs can be scaled to any resolution without losing quality. In this work, we introduce the task of image-to-SVG generation, where the goal is to process an input image and generate SVG code that closely resembles the image upon rendering. This task requires advanced parsing capabilities for textual and numerical data and an understanding of various shapes, such as squares and arrows, commonly found in diagrams. Given an input image $I$, the model outputs SVG code $C$ that visually replicates the image. Additionally, the task can extend to scenarios where a textual description $T$ serves as input, yielding SVG code that aligns with the described content.

We curate a dataset of images, SVG codes, and texts sourced from the SVG-Stack dataset introduced by StarVector (Rodriguez et al., 2023a). The curation process involves filtering to ensure high-quality samples. First, we filter based on image entropy to select images with complex designs and intricate details, excluding simpler icons or shapes. Second, we compute the CLIP Score (Hessel et al., 2022; Radford et al., 2019) for image-text pairs and retain the top 100k examples to build our curated SVG dataset. The SVG-Stack dataset adheres to permissive licensing standards, originating from TheStack (Kocetkov et al., 2022), carefully designed for open and permissive use.

### A.4.5 CHART2MARKDOWN

This dataset is a novel contribution of our work, designed to assess the models' capabilities in extracting data values from chart images. In this task, the model is given a chart image $I$ and asked to produce a data table $T$ of the underlying data table in markdown format.

To create this dataset, we crawled recent chart images from the Statista website [c], focusing on charts from 2023 and 2024 that were not used in prior datasets like UniChart Masry et al. (2023) and ChartQA Masry et al. (2022). This ensures that the dataset reflects the most up-to-date facts and trends and overlaps less with existing datasets and benchmarks. We collected 6,516 chart images, their corresponding data tables, and human-written summaries.

### A.4.6 CHART2CAPTION

The task at hand is a chart-to-caption conversion. The input consists of an image of a chart along with the code used to generate it and the dataset's name and description. The output is a textual caption of the important insights and information conveyed by the chart.

Formally, given a chart image $I$, the code $C$ used to generate the chart, and the dataset information $D$, the goal is to produce a caption $\text{Caption}(I, C, D)$ that highlights key insights and information represented in the chart.

The dataset for this task consists of 1,496 pairs of chart images and the corresponding code that generated these charts. The data was collected by selecting various Kaggle public datasets and their associated data analytics notebooks. We executed these notebooks locally and parsed their outputs to generate the chart-image and code pairs.

To generate the captions, we used the prompt below with the provided chart image, code, dataset name, and description. The caption was generated by using InterVL2-26B (Chen et al., 2023; 2024). This process allows us to leverage the model's capabilities to generate meaningful captions based on the provided context of the chart, code, and dataset description.

> **⛁ Prompt used with InternVL2-26B to generate chart summaries**
>
> ```
> You are a powerful data analyst.  This is a notebook with name
> "{dataset_name}".  In the description of this dataset, it's
> ```

---
[c] https://www.statista.com/

```
told that:  "{dataset_description}".  You are seeing a plot
from this notebook.  Here is the code that was used to generate
this plot:
{code}
Now as a data analyst, summarize the important insights and
information about the chart.
```

### A.4.7 GUI2USERINTENT

The GUI2UserIntent task tests the abilities of grounding on GUI. Concretely, given the bounding box coordinate clicked by the user, the goal is to identify the text element the user intends to interact with. While this is similar to the GUI grounding task introduced in Cheng et al. (2024), which predicts the bounding box based on a user query, it differs in its focus on interpreting user clicks. The datasets are curated by repurposing the pretraining dataset for SeeClick (Cheng et al., 2024). The original dataset includes clickable text elements and their bounding box coordinates in website screenshots. We directly extracted them to curate the GUI2UserIntent dataset.

### A.4.8 GUI2SUMMARY

The GUI2Summary task is similar to the Chart-to-Summary task in NovelCharts; however, instead of charts, the input images $I$ are website screenshots. Unlike existing UI summary datasets like Screen2Words (Wang et al., 2021) that focus on short, phrase-level summaries, our GUI-to-Summary dataset provides paragraph-level summaries. These summaries are comprehensive, providing a brief overview of the main content, referencing key visual and textual elements in the screenshots, and additional aspects like layout and color schemes. To synthesize data for the GUI-to-Summary task, we used InternVL2-8B (Chen et al., 2023), prompting it with website screenshots to create concise descriptions of the main content and layout. Appendix A.4.8 shows the specific prompt used for data generation.

**Prompt used with InternVL2-8B to generate website screenshot summaries**

```
Summarize this website in less than 100 words.  Cover the main
content, layout, color, and other style elements.
```

### A.4.9 GUI-VQA

In the GUI-VQA task, the model answers questions about website screenshots, covering the overall content and specific elements like buttons, text boxes, and menu items. This task requires recognizing key components within the GUI and understanding their functionalities and interdependencies. We synthesized data for GUI-VQA using the GUI2Summary dataset. Specifically, we sampled a sentence from each summary and prompted LLaMA 3.1-8b (Dubey et al., 2024) to convert it into a QA pair. The prompt is shown in Appendix A.4.9.

**Prompt used with InternVL2-8B to convert a summary to a QA pair.**

```
You will be provided with a sentence.  Your task is to convert
it into a question-answer pair.  The question should focus
on factual information and avoid subjective inquiries.  Do
not generate Yes/No questions.  Structure the response in the
format:
Q: {question}
A: {answer}
```

### A.5 DETAILS ON TRAINING VISION-LANGUAGE MODELS ON BIGDOCS

Figure 11 illustrates our training pipeline for evaluating the quality of BigDocs and its impact compared to other datasets. The process has three stages: (1) Pretraining, which focuses on general

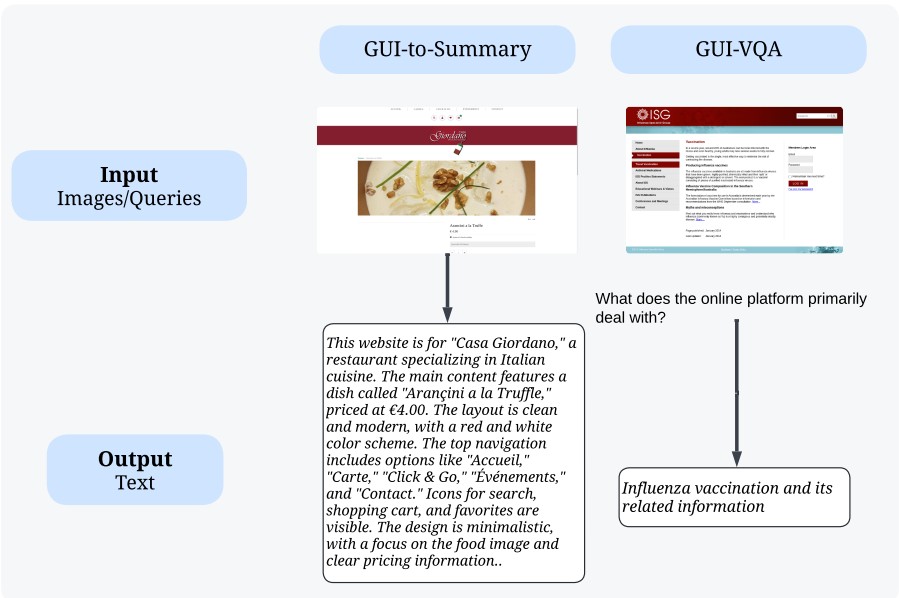

**Figure 9:** *GUI-to-Summary and GUI-VQA introduced in BigDocs-Bench.*

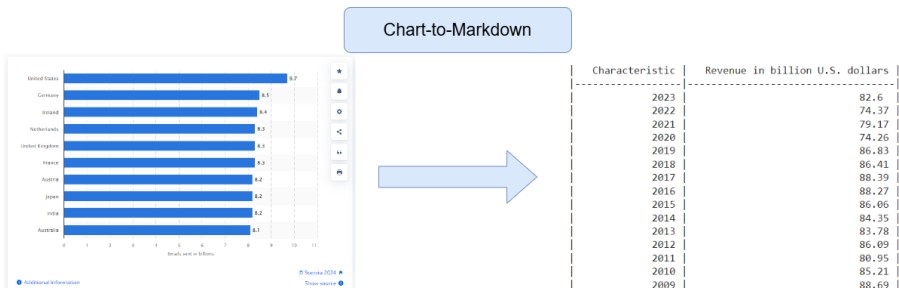

**Figure 10:** *Chart-to-Markdown task introduced in BigDocs-Bench.*

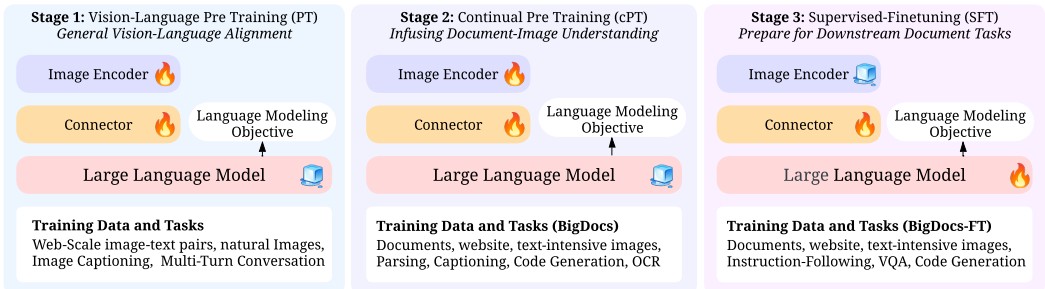

**Figure 11:** *Training Stages for BigDocs. Three-Stage Training Pipeline for multimodal Document Understanding with BigDocs. Our approach consists of: (1) General Vision-Language Pretraining, (2) Document-Specific Continual Pretraining, and (3) Supervised Finetuning for Document Tasks. We evaluate the impact of BigDocs by using checkpoints of models after Stage 1, as provided by their original authors, and comparing them with models that undergo all three stages. Performance is assessed based on standard document tasks and novel tasks, including HTML/SVG code generation, flowchart generation/parsing, and LaTeX interpretation.*

modality alignment; (2) Continual pretraining, where models are further aligned to a specific domain, such as documents, using general tasks like OCR and captioning; and (3) Finetuning, which uses smaller datasets to prepare models for specific downstream tasks, like document processing.

We did not perform stage 1 in this paper, instead relying on checkpoints after pretraining. Additionally, we conducted some experiments using instruction-tuned checkpoints. Throughout all three stages, we employ a generative loss objective, specifically a categorical cross-entropy loss, to predict the next token given the context. The goal of BigDocs-7.5M is to enhance the model's understanding and alignment with document-specific structures. At the same time, BigDocs-Bench focuses on training models to handle tasks that require processing images and converting them into structured code representations. These outputs often follow strict validity constraints to ensure they are syntactically correct. Additionally, the BigDocs-Bench test sets introduce novel benchmarks comprising five company-related downstream tasks, further validating the model's ability to generalize to real-world document-based applications.

**Training Details.** We conduct all our experiments on a cluster of 64 H100 80GB GPUs leveraging the Transformers library (Wolf et al., 2019) for model development. Using Accelerate (Gugger et al., 2022) and Fully Sharded Data Parallel (FSDP) (Zhao et al., 2023), we distribute training over all GPUs in our cluster. We wrap the transformer blocks in both the encoder and the decoder into separate FSDP units with activation checkpointing to achieve a data parallel of 64 without running out of memory. FlashAttention-2 (Dao, 2024) gives a significant speedup on our training runs. For reproducibility, we provide the hyperparameters and training details of our experiments. All experiments maintain a constant total batch size of 256 and a learning rate of 2e-5. We utilize the AdamW optimizer with the following parameters: a cosine learning rate scheduler, 60 warm-up steps, a beta1 coefficient of 0.95, a beta2 coefficient of 0.999, a weight decay of 1e-6, and an epsilon value of 1e-8. For the CPT experiments, models are trained for 1 epoch, while the finetuning experiments on the DocDownstream dataset and BigDocs-Bench are conducted over 3 epochs.

## A.6 DETAILS ON BIGDOCS-BENCH'S EVALUATION

In this section, we dive into more details of the evaluation process for BigDocs-Bench, including the preprocessing procedure before the evaluation and the implementation details of the metrics.

### A.6.1 PREPROCESSING

Before conducting the evaluation, we perform the following preprocessing step for both the ground truth and generation texts:

1. **Image2Flow (GraphViz)**: If the text contains markdown-style fenced code blocks, we use a regular expression to extract the contents within the first code block; otherwise, we treat

the entire text as GraphViz code and remove anything before `digraph` or `graph` if either exists. The flowchart triplets are extracted with the `pydot` library.

2. **Image2Flow (JSON)**: If the text contains markdown-style fenced code blocks, we use a regular expression to extract the contents within the first code block; otherwise, we remove all contents outside the first "{" and last "}" in the text.

3. **Screenshot2HTML**: If the text contains markdown-style fenced code blocks, we use a regular expression to extract the contents within the first code block; otherwise, we treat the entire text as the output HTML code. We first normalize the HTML code with `htmlmin.minify()`, which removes the comments in HTML and condenses the attributes to their most miniature possible representations. Then, we create the DOM tree with `BeautifulSoup4`.

4. **Table2LaTeX**: If the text contains markdown-style fenced code blocks, we use a regular expression to extract the contents within the first code block; otherwise, we treat the entire text as the output LaTeX code. We use a series of regular expressions to normalize the contents. This procedure removes the comments, excessive whitespaces, and commands such as `\label`, `\cite`, `\citep`, `\citet`, `\ref`, `\eqref`, and `\pageref`.

5. **Image2SVG & Text2SVG**: If the text contains markdown-style fenced code blocks, we use a regular expression to extract the contents within the first code block; otherwise, we treat the entire text as the output SVG code. We parse the SVG code and generate a PNG image with the `cairosvg` library.

6. **Chart2Markdown**: If the text contains markdown-style fenced code blocks, we use a regular expression to extract the contents within the first code block; otherwise, we treat the entire text as the output Markdown code. We use regular expressions to remove the code's comments, links, and excessive whitespaces.

7. **Chart2Caption, GUI2UserIntent, GUI2Summary, & GUI-VQA**: We directly evaluate the generated texts against the ground truth.

### A.6.2 METRICS

Here, we provide brief explanations of BigDocs-Bench's evaluation metrics.

1. **Flowchart Triplet F1**: Designed for Image2Flow tasks, this metric evaluates the accuracy of node relationships in flowchart codes, focusing on the correctness of edge triplets $(s, e, d)$ extracted from GraphViz or JSON representations. Here, $e$ denotes the edge label (set to `None` if unlabeled), while $s$ and $d$ represent the source and destination nodes, formatted as "label#shape". The Triplet F1 score is calculated by comparing the generated triplet list against the ground truth, disregarding the ordering of nodes and edges to emphasize relational accuracy.

2. **HTML DOM Tree Edit Distance**: Applicable to Screenshot2HTML tasks, this metric measures the similarity between generated and ground-truth DOM trees using the Tree Edit Distance. Leveraging `BeautifulSoup4` with `lxml` parsing and the `zss` library, the distance is normalized by the larger node count of the compared trees. Invalid GraphViz or JSON generations receive a Triplet F1 score of 0.

3. **TeXBLEU** (Jung et al., 2024)[d]: Utilized for the Table2LaTeX task, TeXBLEU employs a LaTeX-trained tokenizer and a finetuned embedding model with positional encoding. Unlike traditional BLEU, TeXBLEU assesses similarity based on n-gram token precision, demonstrating a higher correlation with human evaluations of LaTeX math expressions compared to BLEU, SacreBLEU, and Rouge (Jung et al., 2024).

4. **RMSF1** (Liu et al., 2022)[e]: Used for the Chart2Markdown task, RMS F1 treats tables as mappings from headers to values, measuring textual and numeric similarity through normalized Levenshtein distance and relative distance. This approach ensures robustness against row and column permutations and transpositions, making it well-suited for evaluating flexible Markdown table structures.

---

[d]Available at `https://github.com/KyuDan1/TeXBLEU`.
[e]Available at `https://github.com/google-research/google-research/tree/master/deplot`.

5. **DINOv2Score**: Employed in the Image2SVG task, this metric calculates the cosine similarity between representations of the ground-truth and generated images using DINOv2 (Oquab et al., 2023), which better captures image similarity than comparable models [f]. Invalid SVG generations are assigned a DINOv2Score of 0.

6. **Rouge-L F1** (Lin, 2004): Applied to summarization and VQA tasks, Rouge-L F1 measures the longest common subsequence between the generated and reference texts. We compute this score using the implementation provided by `torchmetrics`.

## A.7 MAKING BIGDOCS LICENSE-PERMISSIBLE

We dedicated significant effort to acquiring a license-permissible dataset suitable for training models for commercial purposes in document images. We aim to create a large-scale dataset that supports various tasks relevant to companies while adhering to accountability, responsibility, and transparency principles. To achieve this, we thoroughly investigated all public datasets concerning their licenses, evaluating both the sources of the images and their annotations. Our complete analysis, summarized in Table 5, enabled us to identify and retain only the sources that meet permissive licensing criteria.

Dataset licensing frameworks are crucial in determining how data can be used, shared, and modified, generally falling into two categories: *permissive* and *restrictive* licenses. *Permissive licenses* offer the most freedom, allowing for broad usage and modification of the data. For instance, the **CC0** license places the data in the public domain, enabling unrestricted use, modification, and distribution. The **MIT License** permits both commercial and non-commercial applications, provided the license terms are retained in any distribution. In contrast, the **Apache 2.0** License extends these freedoms with an additional grant of patent rights.

In contrast, *restrictive licenses* imposes certain limitations on data usage. The CC BY license requires users to provide attribution to the original creator. In contrast, the **CC BY-SA** license demands that any derivative works also carry the same licensing terms. More restrictive options, like the **CC BY-NC** and **CC BY-ND** licenses, prohibit commercial use and modifications, respectively. In cases where the licensing terms are *unclear*, it is prudent to exercise caution in using the data, as misinterpretation can lead to legal risks.

Additionally, the concept of **Fair Use** allows for limited use of copyrighted material without explicit permission, particularly for purposes such as research, criticism, or commentary. However, Fair Use does not equate to unrestricted permission, and its applicability can vary, necessitating careful consideration when applied to dataset usage in research contexts.

## A.8 HUMAN EVALUATION

To ensure fairness and minimize bias, we invited evaluators from diverse backgrounds, including PhD researchers, ML practitioners, multimodal AI experts, and individuals from both technical and non-technical domains. Key authors were excluded from participating to avoid conflicts of interest. Some non-key authors, such as advisors and authors who performed smaller tasks on the data creation effort, participated. These authors did not know the performance of models on the proposed tasks. For the evaluation, we used our own web interface, to make sure that all participants were anonymous and model outputs were randomized to prevent pattern recognition. We are open to releasing the detailed human evaluation results, as a supplementary document.

We developed a web application using Flask, Javascript, and HTML/CSS where users are presented with pairs of outputs from two models and are asked to judge which model has better output for a given input image. See Figure 12 for a snapshot of the platform.

## A.9 DETAILS OF UNIFIED METADATA FRAMEWORK

We propose a unified metadata framework for the BigDocs dataset to ensure transparency and traceability. This framework is organized into three primary keys: `license`, `origin`, and an optional `features` section. By adopting this structure, we provide a standardized and flexible system that enhances the dataset's usability, ensuring clarity for research and commercial applications.

---

[f] https://medium.com/aimonks/clip-vs-dinov2-in-image-similarity-6fa5aa7ed8c6

**Table 5: *Datasheet of candidate datasets.*** *For transparency purposes, we provide a detailed description of over 100 datasets considered in curating our BigDocs dataset. This table presents a systematic analysis of public datasets, including information on medium types, source documents, text structures, languages, years, annotation types and methods, and licensing for both data and annotations. We also share sample counts across different modalities and splits. Some fields may be blank due to unavailable information. This comprehensive overview enables assessment of each dataset's characteristics and potential contributions to BigDocs.*

| Dataset Name | Data | | | | | | Annotations | | Licenses | | | Total Size | | Units | |
| Acronym | Medium | Source Document Type | Data Sourcing | Text Type | Text Structure | Text Languages | Annotations Type | Annotation method | Images | Annotation | Permissive | Documents | Annotations | Documents | Annotations |
|---|---|---|---|---|---|---|---|---|---|---|---|---|---|---|---|
| unichart-table | Photo | Article | Dataset | Computer Generated | Structures - Charts | English | Layout | Automated | MIT | MIT | Good to use | 304997 | 304997 | Image - Figure | Full page annotation |
| unichart-qa | Photo | Article | Dataset | Computer Generated | Structures - Charts | English | Q&A (Question and Answer) | Synthetic | MIT | MIT | Good to use | 170639 | 300000 | Image - Figure | Element - Q&A pairs |
| DocBank | Digital, Word, Latex | Article - Scientific paper | Repository - ArXiV | Computer Generated | Text with Structures | English | OCR, Layout | Weak Supervision | Apache 2 | Apache 2 | Good to use | 500000 | 500000 | Page | Full page annotation |
| TableBank | Digital, Word, Latex | Article - Scientific paper | Crawling, Repository - ArXiV | Computer Generated | Text with Structures | English, Chinese, Japanese, Arabic, Other | OCR, Table Detection/Extraction | Weak Supervision | Unknown license | Apache 2 | Borderline | 424045 | 562697 | Multi-page | Element - Table structure |
| DocVQA | Scanned, Digital | Legal, Bussines | Repository - Government | Handwritten, Typewritten, Computer Generated | Structures, Text with Structures | English | OCR, Q&A (Question and Answer) | Annotators - Crowdsourced | Fair Use | MIT | Borderline | 12767 | 50000 | Multi-page | Element - Q&A pairs |
| InfographicVQA | Digital | Infographics | Crawling, Repository | Computer Generated | Infographics | English | Q&A (Question and Answer) | Annotators - In house | Unknown license | CC BY | Good to use | 5485 | 30035 | Image | Element - Q&A pairs |
| TQA | Digital | Book - Textbook, Infographics | Repository | Computer Generated | Text with Structures | English | Q&A (Question and Answer) | Weak Supervision | | Not good to use | CC BY-NC 3.0 | 1076 | 26260 | Multi-page | Element - Q&A pairs |
| HierText | Photo | Natural Scene | Dataset | Computer Generated | Natural Image | English | OCR, Layout | Weak Supervision | CC BY 2.0 | CC BY-SA | Good to use | 11639 | 1208128 | Image | Word |
| RecipeQA | Digital - Website | Book - Manuals | Repository | Computer Generated | Text with Images | English | Q&A (Question and Answer) | Annotators - Crowdsourced | Various Licenses | Various Licenses | Not sure | 19779 | 36786 | Multi-page | Element - Q&A pairs |
| ST-VQA | Photo | Natural Scene | Dataset | Computer Generated | Natural Image | English | Q&A (Question and Answer) | Annotators - Crowdsourced | Unknown license | Unknown license | Not sure | 22020 | 30471 | Image | Element - Q&A pairs |
| FigureQA | Digital | Synthetic | Synthetic | Computer Generated | Structures - Figures | English | Q&A (Question and Answer) | Synthetic | MIT | MIT | Good to use | 140000 | 1800000 | Image | Element - Q&A pairs |
| OCR-VQA | Scanned | Infographics | Dataset | Computer Generated | Infographics - Covers | English | Q&A (Question and Answer) | Weak Supervision | Fair Use | Apache 2 | Borderline | 207572 | 1000000 | Image | Element - Q&A pairs |
| DVQA | Digital | Synthetic | Synthetic | Computer Generated | Structures - Charts | English | Q&A (Question and Answer) | Weak Supervision | CC BY-NC 4.0 | CC BY-NC 4.0 | Not good to use | 300k | 3487194 | Image | Element - Q&A pairs |
| Bunny-v1.0-data | Photo | Natural Scene | Dataset | Computer Generated | Natural Image | | Q&A - Dialogs | | | | | 2000000 | | | |
| EXAMS-V | Digital | Academic | Repository - Government | Computer Generated | Text with Structures | English, Chinese, French, German, Italian, Arabic, Polish, Hungarian, Bulgarian, Croatian, Serbian | Q&A (Question and Answer) | Weak Supervision | CC BY-SA 4.0 | Apache 2 | Good to use | 20932 | 20932 | Multi-page | Element - Answers |
| DocILE | Digital, Scanned | Bussines | Repository - Government | Computer Generated | Structures | English | KIE (Key Information Extraction) | Annotators - Crowdsourced | Custom | Custom | Not good to use | 106608 | 106608 | Multi-page | Full page annotation |
| LEAF-QA | Digital | Synthetic | Repository - Government | Computer Generated | Structures - Charts | English | Q&A (Question and Answer) | Weak Supervision | Unknown license | Unknown license | Not sure | 240k | 1.8M | Image - Figure | Element - Q&A pairs |
| WikiTQ | Digital | Article - Wikipedia | Repository - Wikipedia | Computer Generated | Structures - Tables | English | Q&A (Question and Answer) | Annotators - Crowdsourced | CC BY-SA 4.0 | CC BY-SA 4.0 | Good to use | 2108 | 22033 | Image - Table | Element - Q&A pairs |
| VisualMRC | Digital - Website | Article | Crawling | Computer Generated | Text with Structures | English | Q&A - Refered to a ROI | Annotators - Crowdsourced | Unknown license | Custom | Not sure | 10000 | 30562 | Image | Element - Q&A pairs |
| Open4Business | Digital | Report - financial | Repository - Government | Computer Generated | Text with Structures | English | Summary | Weak Supervision | CC BY 3.0, CC BY 4.0 | CC0 | Good to use | 17458 | 17458 | Multi-page | Element - Summary |
| TabFact | Digital | Article - Wikipedia | Repository - Wikipedia | Computer Generated | Structures - Tables | English | Q&A - True/False sentences | Annotators - Crowdsourced | CC BY-SA 4.0 | CC BY-SA 4.0 | Good to use | 16575 | 117854 | Image - Table | Element - Sentence |
| FinTabNet.c | Digital | Report - Financial | Repository - Government | Computer Generated | Structures - Tables | English | OCR, TD, TSE | Weak Supervision | CDLA-Permissive | CDLA-Permissive | Good to use | 112887 | 112887 | Image - Table | Element - Table structure |
| SVIT | Photo | Natural Scene | Dataset | Computer Generated | Natural Image | English | Q&A (Question and Answer) | LLM generated - No permissive like GPT | Various Licenses | CC BY 4.0 | Not sure | 108077 | 4240311 | Image | Element - Q&A pairs |
| Kleister Charity (KLC) | Digital | Report | Repository - Government | Computer Generated | Text with Structures | English | NER (Name-Entity Recognition) | | Unknown license | Unknown license | Not sure | 2778 | 2778 | Multi-page | Full page annotation |
| Kleister-NDA | Digital - PDF | | | | | | | | Unknown license | Unknown license | | 2000000 | | | |
| CORD | Scanned | Bussines - Receipts | Crowdsourced | Computer Generated | Structures - Forms | English | OCR, Layout, KIE | Annotators - Crowdsourced | CC BY 4.0 | CC BY 4.0 | Good to use | 1000 | 1000 | Image | Full page annotation |
| FUNSD | Scanned | Bussines | Dataset | Typewritten, Computer Generated | Structures - Forms | English | OCR, Layout, KIE | Annotators - Crowdsourced | | Custom | Not good to use | 199 | 199 | Image | Full page annotation |
| DeepForm | Digital - PDF | Administrative | Repository - Government - FCC-PIF | Computer Generated | Text with Structures | English | KIE (Key Information Extraction), OCR | Annotators - Volunteers | | MIT | Not sure | 20000 | 20000 | Multi-page | Full page annotation |
| PWC | | | | | | | | | | | Not sure | 2000 | | | |
| COCOText | Photo | Natural Scene | Dataset | Computer Generated | Natural Image | English | Caption - Bounding box | Annotators - Crowdsourced | Unknown license | CC BY 4.0 | Borderline | 63686 | 173589 | Image | Word |
| TextOCR | Photo | Natural Scene | Dataset | Computer Generated | Natural Image | English | | Annotators - In house | CC BY 2.0 | CC BY 4.0 | | 28134 | 903069 | Image | Word |
| PubTables-1M-TSR-TFA | Digital - PDF | Article - Scientific paper | Repository - Pubmed | Computer Generated | Structures - Tables | English | OCR, TD, TSE | Automated | MIT | MIT | Good to use | 948000 | - | Multi-page | |
| PubTables-1M-TD | Digital - PDF | Article - Scientific paper | Repository - Pubmed | Computer Generated | Structures - Tables | English | OCR, TD, TSE | Automated | MIT | MIT | Good to use | 948000 | - | Multi-page | |
| SROIE | Scanned | Bussines - Receipts | Dataset | Computer Generated | Structures - Forms | English | OCR, Layout | Annotators | Unknown license | MIT | Not sure | 1000 | - | Image | Full page annotation |
| VRDU-Registration-Form | Scanned | Administrative | Repository - Government | Computer Generated | Structures - Forms | English | OCR, Layout | Annotators - In house | Unknown license | Unknown license | Not sure | 1915 | - | Multi-page | |
| VRDU ad-buy | Digital - PDF | Administrative | Repository - Government | Computer Generated | Structures - Forms | English | OCR, Layout | Annotators - In house | Unknown license | Unknown license | Not sure | 641 | - | Multi-page | |
| Websight1 | Digital - Website | Synthetic | Synthetic | Computer Generated | Text with Images | English | | LLM generated - Permissive like Llama | CC BY 4.0 | CC BY 4.0 | Good to use | 823000 | - | Image - Screenshot | Code |
| Websight2 | Digital - Website | Synthetic | Synthetic | Computer Generated | Text with Images | English | | LLM generated - Permissive like Llama | CC BY 4.0 | CC BY 4.0 | Good to use | 2000000 | - | Image - Screenshot | Code |
| TextVQA | Photo | Natural Scene | Dataset | Computer Generated | Natural Image | English | OCR, Q&A (Question and Answer) | Annotators - In house | CC BY 2.0 | CC BY 4.0 | Good to use | 28408 | 45336 | Image | Element - Q&A pairs |
| Paper2Fig100k | Digital | Article - Scientific paper | Repository - ArXiV | Computer Generated | Structures - Figures | English | Caption | Weak Supervision | Various Licenses | Various Licenses | Borderline | 102453 | 102453 | Image | Element - Caption |
| CommonCatalog | Photo | Natural Scene | Repository | No Text | | | Caption | LLM generated - Permissive like Llama | Various Licenses | Various Licenses | Good to use | 26232417 | 34.3M | Image | Element - Caption |
| RVL-CDIP | Scanned | Article, Report, Legal documents... | Repository - Government | Handwritten, Typewritten, Computer Generated | Text with Structures | English | OCR, Class | Automated | | | Good to use | 399999 | 399999 | Image | Full page annotation |
| IIT CDIP Colection | Scanned | Article, Report, Legal documents... | Repository - Government | Handwritten, Typewritten, Computer Generated | Text with Structures | English | OCR, Class | Automated | | | Good to use | 11000000 | 11000000 | Image | Full page annotation |
| OCR-IDL | Scanned | Article, Report, Legal documents... | Repository - Government | Handwritten, Typewritten, Computer Generated | Text with Structures | English | OCR, Class | Automated | WTFPL | | Good to use | 26621635 | 26621635 | Page | Full page annotation |
| PDFA | Digital - PDF | Various | Crawling | Various | Text with Structures | English | OCR | Automated | CommonCrawl | CommonCrawl | Not good to use | | | | |
| ShareGPT4V | Photo | Natural Scene | Dataset | No Text | | | Caption | LLM generated - No permissive like GPT | Various Licenses | CC BY-NC 4.0 | Not sure | 100000 | 100000 | Image | Element - Caption |
| ShareGPT4V-PT | Photo | Natural Scene | Dataset | No Text | | | Caption | LLM generated - No permissive like GPT | Various Licenses | CC BY-NC 4.0 | Not sure | 1200000 | 1200000 | Image | Element - Caption |
| LLaVA-Instruct-150K | Photo | Natural Scene | Dataset | No Text | | | Q&A - Dialogs | LLM generated - No permissive like GPT | | CC BY 4.0 | Not good to use | 158000 | 158000 | Image | Element - Q&A pairs |
| LLaVA-CC3M-Pretrain-595K | Photo | Natural Scene | Dataset | No Text | | | Q&A - Dialogs | LLM generated - No permissive like GPT | | CC BY 4.0 | Not good to use | 595000 | 595000 | Image | Element - Q&A pairs |
| LAION-400M | Photo | Natural Scene | Crawling | No Text | | English | Caption | Automated | CommonCrawl | CC BY 4.0 | Various | 413000000 | 413000000 | Image | Element - Caption |
| LAION-5B | Photo | Natural Scene | Crawling | No Text | | Multilingual | Caption | Automated | CC BY 4.0 | CC BY 4.0 | Various | 5850000000 | 5850000000 | Image | Element - Caption |
| LAION-COCO | Photo | Natural Scene | Dataset | No Text | | English | Caption | LLM generated - Permissive like Llama | CommonCrawl | CC BY 4.0 | Various | 600000000 | 600000000 | Image | Element - Caption |
| LAION-Aesthetic | Photo | Natural Scene | Dataset | No Text | | English | Caption | LLM generated - Permissive like Llama | CommonCrawl | CC BY 4.0 | Various | 1200000000 | 1200000000 | Image | Element - Caption |
| LVIS Instruct 4V | Photo | Natural Scene | Dataset | No Text | | English | Q&A - Dialogs | LLM generated - No permissive like GPT | MIT | Borderline | | 100000 | 200000 | Image | Element - Q&A pairs |
| Visual Genome | Photo | Natural Scene | Dataset | No Text | | English | Q&A - Refered to a ROI | Annotators - Crowdsourced | Various Licenses | CC BY 4.0 | Good to use | 108077 | 1773258 | Image | Element - Q&A pairs |
| OK-VQA | Photo | Natural Scene | Dataset | No Text | | English | Q&A (Question and Answer) | Annotators - Crowdsourced | Various Licenses | Unknown license | Not sure | 14031 | 14055 | Image | Element - Q&A pairs |
| A-OKVQA | Photo | Natural Scene | Dataset | No Text | | English | Q&A (Question and Answer) | Annotators - Crowdsourced | Various Licenses | Unknown license | Not sure | 23700 | 24900 | Image | Element - Q&A pairs |
| VisDial | Photo | Natural Scene | Dataset | No Text | | English | Q&A - Dialogs | Annotators - Crowdsourced | Various Licenses | Unknown license | Not sure | 133060 | 1330600 | Image | Element - Q&A pairs |
| Hateful Memes | Digital | Infographics | Crowdsourced | Computer Generated | Text with Images | English | Class | Annotators - Crowdsourced | Custom | Custom | Not sure | 10000 | 10000 | Image | Element - Class |
| Table-LLaVA | Digital | Article | Dataset | Computer Generated | Structures - Tables | English | Q&A (Question and Answer) | LLM generated - No permissive like GPT | Various Licenses | Unknown license | Not sure | 97000 | 150000 | Image - Table | Element - Q&A pairs |

Along with the dataset, we include two separate files named `tracking_instructions.json` and `tracking_transformations.json`, which are dictionaries that store information standards across large subsets of samples, reducing redundancy and promoting consistency in metadata.

**License Metadata** The `license` key is mandatory and provides detailed information regarding the licensing terms of both the images and annotations. It is structured as a dictionary with three sub-keys:

- `image_license` (mandatory): Specifies the license under which the image is provided.

- `annotation_license` (mandatory): Indicates the license for the annotations or labels associated with the image (e.g., text, bounding boxes, tables).

- `license_note` (optional): Includes additional licensing information or clarifications, such as specific usage conditions or exceptions. This allows for clear communication of any nuances in licensing terms.

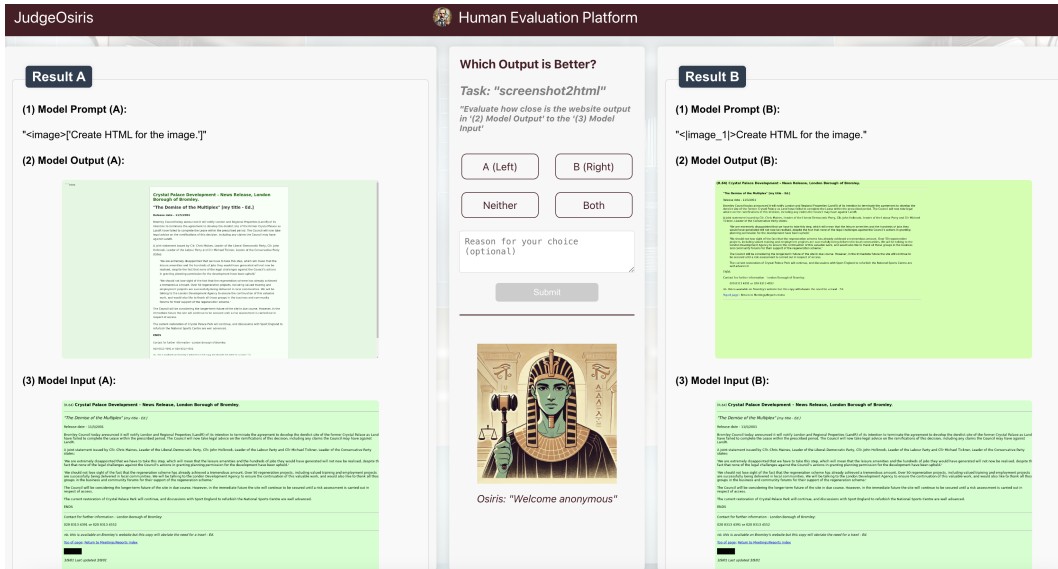

**Figure 12:** *Human Evaluation Platform. A user is presented with two model outputs and asked to select the one that is more accurate in relation to the original model input.*

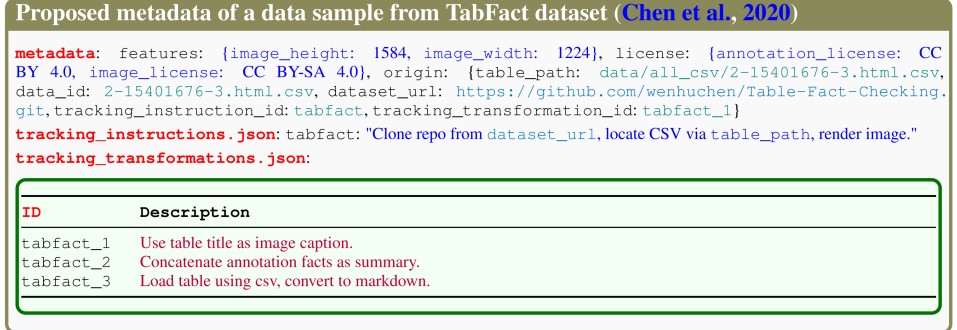

**Figure 13:** *Metadata framework from our BigDocs Toolkit for a sample in the BigDocs 7.5M dataset. It includes details about image properties, licenses, data sources, and transformation steps, such as using the table title as an image caption.*

**Origin Metadata**    The `origin` key is mandatory and serves as a traceability mechanism, allowing users to track each data sample back to its source. This key is also structured as a dictionary and includes the following sub-keys:

- `data_id` (mandatory): A unique identifier for each data sample (e.g., file name, index, or ID in the source dataset).
- `tracking_instruction_id` (mandatory): A reference to an entry in the `tracking_instructions.json` file, specifying how to use the metadata fields to trace the data sample back to its source.
- `tracking_transformation_id` (optional): A reference to an entry in the `tracking_transformations.json` file, providing a high-level description of any transformations or modifications applied to the original sample.
- `dataset_url` (optional): The URL of the original dataset from which the sample was obtained, if applicable.
- `image_path` (optional): The local path to the image within the raw dataset files, if available. This key is helpful when the dataset is stored in raw form.
- `image_url` (optional): The URL from which the image was initially downloaded.
- `Misc.` (optional): Other information related to the origin of the data entry.

**Features Metadata**    The `features` key provides detailed characteristics of the data sample. This can be useful for downstream tasks that require specific information about the sample, like the image size and type of annotations.

- `image_height`: The height of the image, in pixels.
- `image_width`: The width of the image, in pixels.

**Common    Metadata    Files.**        The    `tracking_instructions.json`    and `tracking_transformations.json` files are designed to store information common to many samples, minimizing redundancy and ensuring consistency. Each file is structured as a dictionary:

- `tracking_instructions.json`: Contains instructions on how to use the metadata fields (such as `data_id`, `dataset_url`, etc.) to trace the data sample back to its source. Each entry is uniquely identified and can be referenced in the individual sample metadata by `tracking_instruction_id`.
- `tracking_transformations.json`: Contains descriptions of any transformations, processing steps, or additions to the original samples. This includes synthetic generation processes, data augmentation steps, or other modifications. Each transformation is uniquely identified and can be referenced in the individual sample metadata by `tracking_transformation_id`.

### A.10    QUALITATIVE RESULTS

We provide a detailed qualitative analysis to highlight the performance of different models on tasks within BigDocs-Bench. Table 6 showcases the outputs of our best-performing model, Phi-3.5-Vision trained on BigDocs, alongside a publicly available instruction-tuned model and GPT4o. The analysis covers tasks such as GUI-VQA, Chart2Caption, Image2Latex, and Screenshot2HTML. Notably, the BigDocs-trained model demonstrates more concise and accurate free-form outputs for VQA tasks, generates perfect LaTeX tables from table images, and produces precise HTML code from web screenshots.

Additionally, we present a series of qualitative examples across BigDocs-Bench tasks, detailed in Tables 9-28. These examples provide direct comparisons of outputs from all models discussed in the paper. The examples are accompanied by a thorough error analysis that highlights each model's strengths, limitations, and both success and failure cases.

### A.11    GLOSSARY & TASK DEFINITIONS

We facilitate a list of concepts introduced and utilized throughout our research. Below is a list of these concepts and tasks, along with their definitions, including descriptions, inputs, and outputs.

1. **Text with Structures**
    - **Description**: Analyzing documents combining text with elements like headers, lists, and paragraphs to understand the overall structure and organization.
    - **Input**: Documents containing structured text elements such as headers, lists, and paragraphs.
    - **Output**: A structured representation of the document's content.
2. **Structures - Figures**
    - **Description**: Identifying and interpreting figures, diagrams, and other visual elements in documents to extract the information they represent.
    - **Input**: Documents with embedded figures, diagrams, and visual elements.
    - **Output**: A detailed understanding or extraction of the visual content.
3. **Infographics - Covers**
    - **Description**: Examining infographics and cover pages that use a mix of text and graphics to communicate information visually.
    - **Input**: Infographics and cover pages containing text and visual elements.

**Table 6:** *Task Structure and Samples with Model Outputs for Phi Models and GPT-4o*

| Task | Query | Phi-3.5-Vision Instruct | Phi-3.5-Vision Instruct-BigDocs | GPT-4o |
|------|-------|-------------------------|--------------------------------|--------|
| **GUI-VQA** | [What opens when the user clicks [550, 0, 600, 80]?] | [The image does not provide information on what opens when the user clicks the specified coordinates.] | [View the news] | [By clicking on the coordinates [620, 50, 650, 70], the user will navigate to the "news" section of the website.] |
| **Chart2Caption** | [I need a caption for the chart in the image, please] | [The image displays a pie chart with various segments, each representing a different range of values and their corresponding percentages. The segments are color-coded and labeled with both the value range and the percentage they represent. The chart is sourced from Statista 2024] | [In 2022, the most common salary range for employees in the United Kingdom was between 100,000 and 149,999 British pounds, with 19.3 percent of employees falling into this bracket. Employees earning between 200,000 and 249,999 British pounds made up the second most common salary range, at 15.8 percent.] | [Distribution of Income Levels: A Pie Chart Showing the Percentage of Households by Income Bracket in 2024] |
| **Image2Latex** | [Create LaTeX code to produce this table.] | | | |
| **Screenshot2HTML** | [Could you create Hypertext Markup Language code from the screenshot?] | | | |

- **Output**: A comprehensive analysis of the combined visual and textual information.

4. **Structures - Tables**

    - **Description**: Finding and understanding tables in documents, focusing on accurately capturing the data and relationships they present.
    - **Input**: Documents containing tabular data.
    - **Output**: A structured dataset or representation of the tabular information.

5. **Structures - Forms**

    - **Description**: Recognizing and processing forms within documents, extracting data from fields and labels typically found in structured forms.
    - **Input**: Documents with structured forms, including fields and labels.
    - **Output**: An organized collection of the extracted form data.

6. **Natural Image**

    - **Description**: Analyzing photographs or natural images in documents, interpreting their content in the context of the surrounding text.
    - **Input**: Documents containing photographs or natural images.
    - **Output**: An interpreted or categorized representation of the image content.

## A.12 ABLATION EXPERIMENTS

### A.12.1 ABLATIONS ON DATA CURATION

We ablated the main data curation procedures on BigDocs, namely the use of 1) VQA format during CPT, i.e. using (question, image, answer) tuples with *assistant* and *user* conversation formats, 2) the use of heavy OCR tasks during CPT (converting most of our document images into OCR tasks) and our 3) text-image alignment tasks, mainly captioning of many datasets. **Experiment Setup.** Using

**Table 7:** *Ablation Experiment on the effect of VQA formatting, presence of OCR data, and presence of captioning data in BigDocs-7.5M. Results show the importance of including these curation steps and tasks. See discussion in Section A.12.*

| Model | DocVQA VAL | InfoVQA VAL | DeepForm TEST | KLC TEST | WTQ TEST | TabFact TEST | ChartQA TEST | TextVQA VAL | MMMU VAL | DudeMini TEST | SlideVQA-M TEST | TableVQA TEST | Avg. Score |
|---|---|---|---|---|---|---|---|---|---|---|---|---|---|
| **Original** | 87.05 | 70.05 | 70.97 | 37.45 | 51.21 | 81.24 | 81.56 | 68.72 | 45.00 | 36.15 | 32.47 | 67.77 | **60.80** |
| **No VQA** | 82.65 (-4.4) | 54.92 (-15.13) | 40.64 (-30.33) | 28.8 (-8.65) | 36.93 (-14.28) | 65.3 (-15.94) | 78.4 (-3.16) | 65.24 (-3.48) | 42.33 (-2.67) | 34.71 (-1.44) | 8.37 2(-4.1) | 61.13 (-6.64) | **51.62 (-9.19)** |
| **30% OCR** | 85.79 (-1.26) | **60.03** (-10.02) | **54.19** (-16.78) | 34.9 (-2.55) | 52.68 (1.47) | 73.76 (-7.48) | 81.56 (0) | 72.07 (3.35) | 42.56 (-2.44) | 36.72 (0.57) | 32.07 (-0.4) | 70.83 (3.06) | **58.1 (-2.71)** |
| **30% caption** | 85.36 (-1.69) | **57.98** (-12.07) | **53.36** (-17.61) | 35.09 (-2.36) | 50.91 (-0.3) | 72.5 (-8.74) | 82.4 (0.84) | 71.92 (3.2) | 40.89 (-4.11) | 36.76 (0.61) | 31.97 (-0.5) | 69.37 (1.6) | **57.38 (-3.43)** |

the Phi3.5-Vision-Instruct model, we created three modified versions of BigDocs-7.5M: one without the VQA format, one with OCR tasks removed, and one with reduced captioning tasks (text-image alignment). Each dataset variant was used to train the model, followed by fine-tuning the model on our instruction-tuning datasets. We evaluated these models on the benchmarks listed in Table 2 (General Document Benchmarks). The results are presented below. In parentheses, we show the delta with respect to the original result.

**Main Findings.**

1. **VQA Format:** VQA formatting proved crucial, as its removal led to a 9.19% average performance decrease. This was especially apparent in InfoVQA (-15.13), DeepForm (-30.33), WTQ (-14.28), and TabFact (-15.94). A common pattern observed here is a lack of adherence to the requested output format as demonstrated in the example in Table X. We hypothesize that the VQA format used in the continual pre-training stage (CPT) enables better instruction tuning and supports multitask generalization.

2. **OCR Impact:** Removing OCR tasks led to an average performance drop of 2.71%, with varying impacts across benchmarks. Notably, we observed significant declines on InfoVQA (-10.02), DeepForm (-16.78), and TabFact (-7.48), while results on DUDE, SlideVQA, and ChartQA remained largely unaffected. The reduction in OCR data increased hallucination in tasks requiring precise information extraction. For instance, in DeepForm, the true negative rate dropped from 62% to 0%, as models began hallucinating unrelated values from other parts of the document. Examples illustrating this behavior are provided in Tables X and Y. Interestingly, in some cases such as TableVQA (+3.06) and TextVQA (+3.35), performance improved, likely due to the nature of the question-answering tasks, which often favor free-form text generation over exact OCR matches. Table 21 shows an example of this phenomenon.

3. **Text-Image Alignment:** Reducing text-image alignment tasks, particularly captioning, resulted in a 3.4% performance drop, with notable declines on InfoVQA (-12.07), DeepForm (-17.61), and TabFact (-8.74). This drop primarily stemmed from errors in parsing values for arithmetic operations, especially when data formats varied, such as converting text or unique visual elements into percentages illustrated by examples in Table X, Table Y and Table Z. Tasks without significant drops, like WTQ and ChartQA, involve simpler text-visual correspondences that don't require complex conversions. See Table 22 for a qualitative example

### A.12.2 ABLATION ON CONTEXT LENGTH

The results presented thus far have been limited to a context length of 8192 tokens. Given that the Screenshot2HTML task averages around 32k tokens per example (as shown in Table 1), it encounters significant challenges under this 8k-token constraint. This task ranks among those with the lowest scores (see Table 3), which contributes to a decline in the overall aggregated average score.

Despite this limitation, we found that the generated HTML code remains compilable, and the resulting DOM can still be evaluated using the DOM Tree Edit Distance metric, even though the HTML output is truncated due to the context limit. However, this inability to handle complete HTML outputs highlights the necessity for models capable of processing extended context lengths.

To better understand this issue, we analyzed the model outputs (HTML code) in terms of a histogram of token lengths, comparing them with the ground truth lengths in the test set (see Figure18). It is important to note that the test set is designed so that the images can be represented in fewer than 8192 tokens (as shown in the left figure). This figure highlights that token length constraints can adversely affect the task, as most samples approach the maximum token length of 6k (after accounting for image tokens), as seen in the right figure. Although the test set is designed to be generated within a reasonable token length, the model's training on larger contexts leads it to attempt to generate longer HTML outputs.

We fixed the context length at 8192 tokens for several key reasons. First, not all models evaluated in our benchmark natively support larger context lengths, and extending this limit could disadvantage some models. Second, increasing context lengths significantly boosts GPU memory usage, complicating both training and inference at scale. Finally, to ensure fair comparisons across all models, we standardized the context length at 8k tokens.

We conducted an ablation study on context length by training the Phi3.5-Vision model with a context length of 32k tokens. Subsequently, we re-evaluated the Screenshot2HTML task using context lengths ranging from 512 to 16k tokens. However, generating 32k tokens resulted in out-of-memory errors. The results are presented in Table 29, where we applied greedy decoding (second column) and beam search with a width of 2 and a length penalty of -1.0 (third column).

The results are revealing. We do not observe any performance boost when training and evaluating with longer context lengths. The greedy decoding algorithm performs worse with longer contexts, not exceeding a score of 10.99. In contrast, beam search with a length penalty approach yields better results, reaching 15.80, but still does not benefit from larger context lengths.

Additionally, we conducted a qualitative error analysis (see Figures 19 - 22) and found that the model often struggles with stopping generation, frequently hallucinating repetitive patterns such as nested `<div>` elements or excessively long URLs and references. However, well-formatted HTML code is generated in approximately 40% of cases (Figure 18, top figure). Applying a length penalty helps mitigate this hallucination, leading to improved performance. This indicates further potential for enhancing results with alternative generation techniques.

This indicates that the performance issue may not stem from context length but rather from the inherent difficulty of the Screenshot2HTML task. This aligns with the observation that the test set's average context length is approximately 3k tokens (Figure18, left) and does not require more than 8k tokens. We suspect that the primary limitation lies in the insufficient amount of training data (currently around 10k samples) which may hinder the model's ability to learn HTML generation effectively.

### A.13 DETAILS IN DATA VERIFICATION ON BIGDOCS-BENCH

This section provides a detailed description of the verification methodology, evaluation criteria, and measures taken to ensure the quality and reliability of BigDocs-Bench. We outline the qualifications of the human annotators, the multi-step verification process combining automated filtering and human review, and the strategies used to address the challenges posed by the dataset's large size and synthetic nature. The section also discusses our sampling-based quality assurance for the training split, inter-rater reliability measures, and the conservative approach we adopted to prioritize recall and dataset safety. These steps collectively demonstrate our commitment to maintaining high standards in creating a robust and ethical benchmark.

### A.13.1 VERIFICATION METHODOLOGY

We generally follow a 4-step strategy to perform human verification as follows,

**1. Automatic Filtering Tools.** We implemented multiple automatic filtering tools as the first step in our quality control process. These tools were designed to detect issues such as bad annotations, NSFW content (Llama-Guard-3 from https://huggingface.co/meta-llama/Llama-Guard-3-11B-Vision), and Personally Identifiable Information (PII). This automated process reduced the volume of problematic samples before any human verification was performed.

**2. Human Verification for Test Split.** To ensure a high-quality test split, we only do human verification focused on the test set as our automatic filtering is already quite robust. Annotators identified and removed problematic samples following the guideline, for example, unnecessary comments in LaTeX and HTML code (e.g., in Table2LaTeX and Screenshot2HTML tasks), which could cause rendering issues. Similarly, flowcharts with excessively isolated nodes in tasks like Image2Flow were flagged and removed. For Image2SVG and three GUI-related datasets, we spotted and removed those images with PII or NSFW content. All these typical errors were collected and documented during this process to inform our automatic filtering tools.

**3. Refining Automatic Filtering Tools for Training Split.** The typical errors identified during human verification of the test set were used to iteratively improve our automatic filtering tools. This enhancement enabled the tools to better detect and handle similar issues in the training split, which was impractical to verify comprehensively through human labor. **4. Sampling-Based Human Verification for Training Split.** To ensure the training split met quality standards, we randomly sampled 100 samples from all tasks, proportional to the size of their respective training splits. These samples were verified by annotators based on the criteria mentioned below, and the overall pass rate was 99% (i.e. 99 good samples), reflecting a high level of quality for the training data after applying the refined filtering tools. Specifically, every task has a pass rate of 100% except GUI-VQA, which has a pass rate of 92% (i.e. 1 bad sample out of 13, which is not relevant to the task since the question is not quite related to the given image).

### A.13.2 VERIFICATION CRITERIA FOR VERIFIERS

The verification process for BigDocs-Bench prioritized several critical aspects to ensure the dataset's quality, reliability, and suitability for real-world applications.

1. First, **data integrity** was a primary focus, ensuring accurate alignment between inputs and outputs. For example, in tasks like Image2SVG, visual elements in images were verified to match their corresponding SVG annotations, ensuring precise vector representation. Similarly, for Screenshot2HTML, rendered HTML outputs were checked against the screenshots to confirm structural and semantic fidelity.

2. Second, the **relevance of synthetic samples** was thoroughly assessed to confirm that they were realistic and reflective of the intended task objectives. For instance, in the Image2Flow task, flowchart samples were evaluated to ensure that the generated JSON or GraphViz code meaningfully represented the logical flow of the diagrams without excessive isolated nodes or disconnected components, which could compromise their utility.

3. Finally, to ensure **content safety**, NSFW content and PII, such as explicit material, or personal data, were removed.

### A.13.3 DETAILS ON HUMAN VERIFIERS

We engaged a team of 6 human verifiers, consisting of graduate-level researchers and professionals with expertise in multimodal datasets and document analysis. All human verifiers were thoroughly trained on the verification criteria outlined below before commencing the review process. Each data sample was reviewed by at least two verifiers to ensure thoroughness.

### A.13.4 INTER-RATER RELIABILITY

In the test split of our benchmark, we prioritized recall to ensure no potentially problematic samples were missed. For tasks like filtering NSFW content, detecting annotation errors, or identifying

**Table 8:** *General Document Benchmarks*. *Models trained on {BigDocs-7.5M+DocDownstream} perform competitively across multimodal document benchmarks. We compare them to base checkpoints, instruction-tuned models, and those trained on {DocStruct4M+DocDownstream}. We also report scores on state-of-the-art models >7B for reference. BigDocs models show consistent performance across tasks. * Refers the use of Chain-of-Thought (CoT)*

| Model | DocVQA VAL. | InfoVQA VAL. | DeepForm TEST | KLC TEST | WTQ TEST | TabFact TEST | ChartQA TEST | TextVQA VAL. | MMMU VAL. | DudeMini TEST | SlideVQA-M TEST | TableVQA TEST | Avg. Score |
|---|---|---|---|---|---|---|---|---|---|---|---|---|---|
| DocOwl1.5-8B (instruct) | 80.73 | 49.94 | 68.84 | 37.99 | 38.87 | 79.67 | 68.56 | 68.91 | 33.67 | 34.64 | 31.62 | 52.60 | 53.84 |
| DocOwl1.5-8B (base) | 2.07 | 1.84 | 0.00 | 0.00 | 0.00 | 0.00 | 0.00 | 0.00 | 24.44 | 19.07 | 3.30 | 13.63 | 5.36 |
| DocOwl1.5-8B (base) + DocStruct4M | 75.99 | 46.88 | 62.77 | 35.21 | 32.86 | 71.56 | **68.36** | 65.08 | **33.67** | 29.00 | 27.03 | 46.27 | 49.56 |
| DocOwl1.5-8B (base) + BigDocs (Ours) | **78.70** | **47.62** | **64.39** | **36.93** | **35.69** | **72.65** | 65.80 | **67.30** | 32.33 | **32.55** | **29.60** | **49.03** | **51.05** |
| Qwen2-VL-2B (instruct) | 89.16 | 64.11 | 32.38 | 25.18 | 38.20 | 57.21 | 73.40 | 79.90 | 42.00 | 45.23 | 46.50 | 43.07 | 53.03 |
| Qwen2-VL-2B (base) | 7.26 | 0.78 | 0.00 | 0.00 | 0.00 | 0.00 | 0.00 | 1.14 | 34.89 | 28.43 | 14.55 | 0.00 | 7.25 |
| Qwen2-VL-2B (base) + DocStruct4M | **59.53** | **32.00** | **53.98** | **36.38** | 28.48 | 64.24 | 54.44 | 55.89 | 34.89 | **28.78** | **22.68** | 46.53 | 43.15 |
| Qwen2-VL-2B (base) + BigDocs (Ours) | 57.23 | 31.88 | 49.31 | 34.39 | **31.61** | **64.75** | **68.60** | **61.01** | **35.67** | 27.19 | 17.46 | **47.53** | **43.89** |
| Phi3.5-Vision-4B (instruct) | 86.00 | 56.20 | 10.47 | 7.49 | 17.18 | 30.43 | 82.16 | 73.12 | 46.00 | 37.20 | 30.93 | 70.70 | 45.66 |
| Phi3.5-Vision-4B + DocStruct4M | 86.76 | 68.90 | 70.12 | **37.83** | **51.30** | **82.12** | 79.76 | 68.60 | 44.11 | 35.52 | 31.90 | **69.17** | 60.51 |
| Phi3.5-Vision-4B + BigDocs (Ours) | **87.05** | **70.05** | **70.97** | 37.45 | 51.21 | 81.24 | **81.56** | **68.72** | **45.00** | **36.15** | **32.47** | 67.77 | **60.80** |
| LLaVA-NeXT-7B (instruct) | 63.51 | 30.90 | 1.30 | 5.35 | 20.06 | 52.83 | 52.12 | 65.10 | 38.89 | 17.94 | 7.46 | 32.87 | 32.36 |
| LLaVA-NeXT-7B + DocStruct4M | **60.95** | **26.14** | 39.78 | 28.34 | 25.90 | 67.72 | **61.20** | **52.25** | **25.78** | 21.70 | 15.33 | 27.03 | 37.68 |
| LLaVA-NeXT-7B + BigDocs (Ours) | 57.13 | 24.47 | **46.38** | **31.09** | **27.06** | **72.58** | 54.72 | 49.06 | 17.78 | **22.88** | **16.07** | **33.13** | **37.70** |
| Llama-3.2-90B | 74.15* | 48.71 | 4.18 | 1.81 | 24.20 | 63.01 | 11.36* | 71.69 | 57.78 | 41.24 | 26.09 | 41.57 | **38.82** |
| GPT-4o 20240806 | 92.80 | 66.37 | 38.39 | 29.92 | 46.63 | 81.10 | 85.70 | 70.46 | 69.10 | 54.55 | 67.58 | 72.87 | **64.62** |
| Claude-3.5 Sonnet | 88.48 | 59.05 | 31.41 | 24.82 | 47.13 | 53.48 | 51.84 | 71.42 | 64.78 | 35.11 | 0.00 | 81.27 | **50.73** |
| GeminiPro-1.5 | 91.23 | 73.94 | 32.16 | 24.07 | 50.29 | 71.22 | 34.68 | 68.16 | 58.22 | 48.15 | 52.05 | 80.43 | **57.05** |
| Qwen2-VL-72B | 96.50 | 84.50 | 30.45 | 24.78 | 55.63 | 0.00 | 88.30 | 85.50 | 64.50 | 35.87 | 2.15 | 74.23 | **58.40** |

corrupted data, any sample flagged as problematic by a single verifier was removed. This conservative recall-focused approach maximized dataset quality and safety while addressing misalignment between raters.

## A.14 RESULTS ON LARGER MODELS

We have evaluated larger models on general document benchmarks, and the results are presented in Table 8. Models with more than 8 billion parameters perform exceptionally well on the general document benchmarks leaderboard. The top performer is GPT-4o, achieving an average score of 64.62, followed by Qwen2-VL-72B with 58.40 and GeminiPro-1.5 with 57.05.

## A.15 CORRELATION ANALYSIS OF BIGDOCS-BENCH WITH OTHER VLM BENCHMARKS

To assess the novelty of BigDocs-Bench in the VLM landscape, we conduct a correlation analysis that highlights how BigDocs-Bench distinguishes itself from existing benchmarks. Specifically, we leverage model results across VLM benchmarks to demonstrate that BigDocs-Bench offers unique dimensions of model evaluation and analysis.

First, we analyze the correlation between Document Benchmarks by comparing the results from Table 2 and Table 1. Following this, we expand the analysis to include results from several widely used VLM benchmarks, providing a comprehensive view of the relationship between BigDocs-Bench and prior benchmarks in the field.

**Analysis Setup.** We selected all benchmarks listed in Tables 2 and 1, encompassing both General Document Benchmarks and our proposed BigDocs-Bench, respectively. We constructed a matrix encompassing all models presented in the paper, namely DocOwl-1.5-8B, Qwen2-VL-2B, LLaVA-NeXT-7B, Phi3.5-v-4B, both the off-the-shelf public versions and the ones trained on BigDocs. We also extended Table 2 with scores for GPT4o, Claude 3.5 Sonet, Qwen2-VL-72B, GeminiPro-1.5, and Llama-3.2-90B, to obtain a full matrix. Note that we have now extended our baselines with Llama-3.2-90B. Finally, we aggregated results for all metrics in the two groups of benchmarks. We have edited our manuscript to include Table 8, which is an extension of Table 2 with scores from all models.

In our broader study with general VLM benchmarks, including but not limited to documents, we expanded the analysis to encompass 28 additional VLM benchmarks (see Figure 16 for the full list). Missing scores were imputed using the mean to complete the matrix for models and benchmarks.

**Methods.**

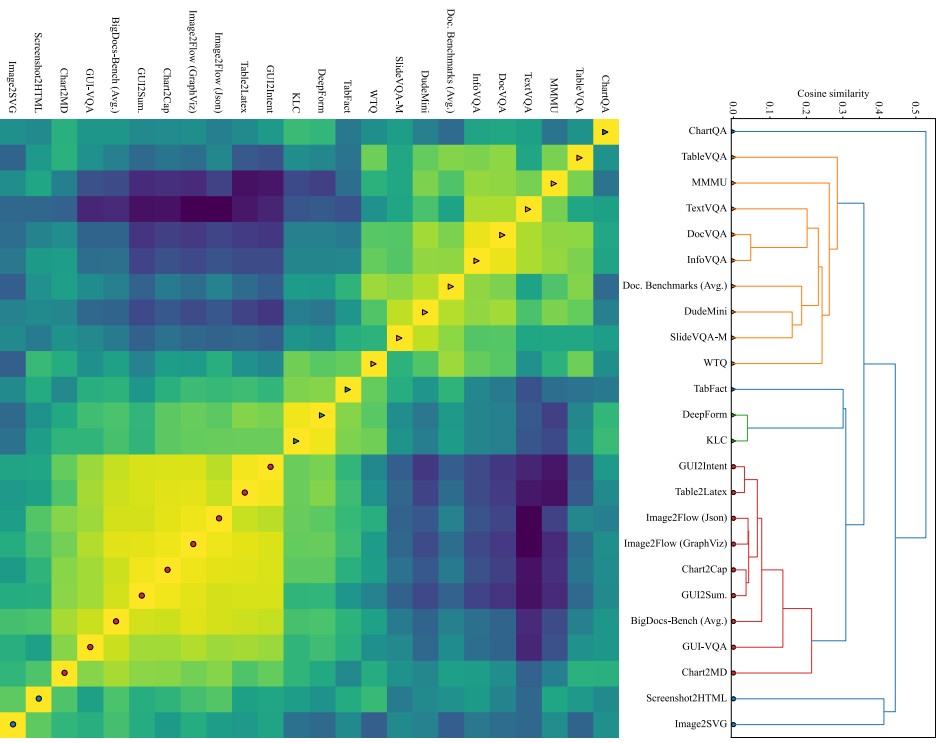

**Figure 14:** *Correlation matrix and dendrogram for BigDocs-bench and General Document Benchmarks. The matrix (left) shows benchmark relationships, with BigDocs-Bench tasks (circles) clustering separately from other General Document Benchmarks (triangles). The dendrogram (right) highlights distinct groupings, including BigDocs-Bench tasks for code and GUI generation clustering together and VQA tasks forming a separate cluster. Tasks like DeepForm and KLC align closely, reflecting their shared complexity.*

1. **Cosine similarity matrix and dendrograms:** We normalize metric scores by centering (subtracting the mean) and scaling (dividing by the Euclidean norm). We then apply hierarchical clustering using cosine distance to group benchmarks and visualize their relationships. Cosine similarity matrices and dendrograms illustrate these clusters.

2. **Principal Components Analysis (PCA):** We represent each benchmark as a feature array composed of scores from all models and project these arrays onto the two principal dimensions.

### A.15.1 CORRELATION WITH DOCUMENT BENCHMARKS

Our results shown in Figures 14 and 15 show that BigDocs-Bench tasks and benchmarks are different from every other document dataset.

Figure 14 presents the correlation matrix and dendrograms. Circles represent BigDocs-Bench tasks (our benchmarks), while triangles denote other document benchmarks included in the paper. These results demonstrate that BigDocs-Bench tasks are notably distinct from other benchmarks, as indicated by their low correlation with them. A clear grouping emerges for tasks related to VQA, as well as BigDocs-Bench tasks involving code generation in LaTeX, JSON, GraphViz, HTML, and SVG. Notably, HTML and SVG tasks form a cluster, likely due to their characteristically long output sequences. Additionally, DeepForm and KLC stand out as a separate cluster, distinct from all others.

Figure 15 illustrates the PCA results. The clusters distinctly separate BigDocs-Bench tasks from other benchmarks, reaffirming their unique characteristics. Notably, the average aggregated score from BigDocs-Bench (left side) is the furthest from the aggregated score of other benchmarks (right side), highlighting the distinctiveness of our benchmarks. However, certain tasks, such as Image2SVG and Screenshot2HTML, show stronger correlations with KLC, DeepForm, and TabFact. This is likely due to the shared level of difficulty across these benchmarks.

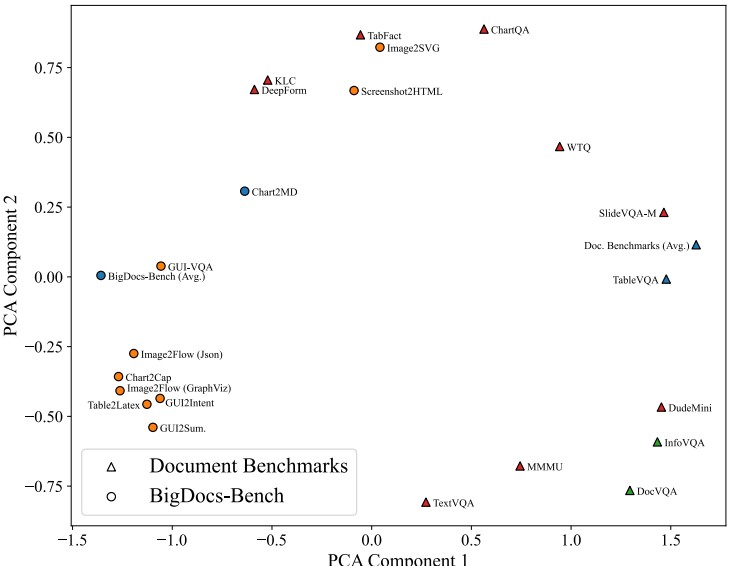

**Figure 15:** *PCA of BigDocs-Bench and General Document Benchmarks. BigDocs-Bench (circles) form distinct clusters from other document benchmarks (triangles), emphasizing their unique characteristics. Tasks related to code generation, such as Image2Flow and Table2Latex, cluster together, while VQA tasks, including those from other benchmarks, are further apart.*

### A.15.2 CORRELATION WITH GENERAL VLM BENCHMARKS

Figures 16 and 17 show that BigDocs-Bench tasks and benchmarks are unique in the broad VLM space.

Figure 16 presents the correlation matrix and dendrograms. Circles represent BigDocs-Bench tasks (our benchmarks), upward triangles denote other document benchmarks included in the paper, and downward triangles denote general VLM benchmarks. BigDocs-Bench tasks demonstrate distinctive evaluation characteristics compared to existing benchmarks in the VLM space. This is evidenced by the clear clustering pattern, where BigDocs-Bench tasks form their own correlated group while showing minimal correlation with other benchmark types. Results also reveal three main clusters of tasks: general vision-language tasks (like COCO VAL and VCR), document understanding tasks, and GUI/table-related tasks (such as our tasks GUI2Sum and Table2Latex). This clustering pattern suggests that BigDocs-Bench is addressing unique aspects of document understanding that isn't captured by existing benchmarks.

Figure 17 illustrates the PCA results, revealing that most BigDocs-Bench tasks are clustered in the bottom-right corner, distinctly separated from other benchmarks. Notably, tasks within BigDocs-Bench form clear clusters based on their type, such as VQA, Flows, Charts, or GUIs. General VLM benchmarks create a dense cluster, while document benchmarks are more sparse. Additionally, the aggregated BigDocs-Bench average score is positioned far from other benchmarks, further highlighting its uniqueness.

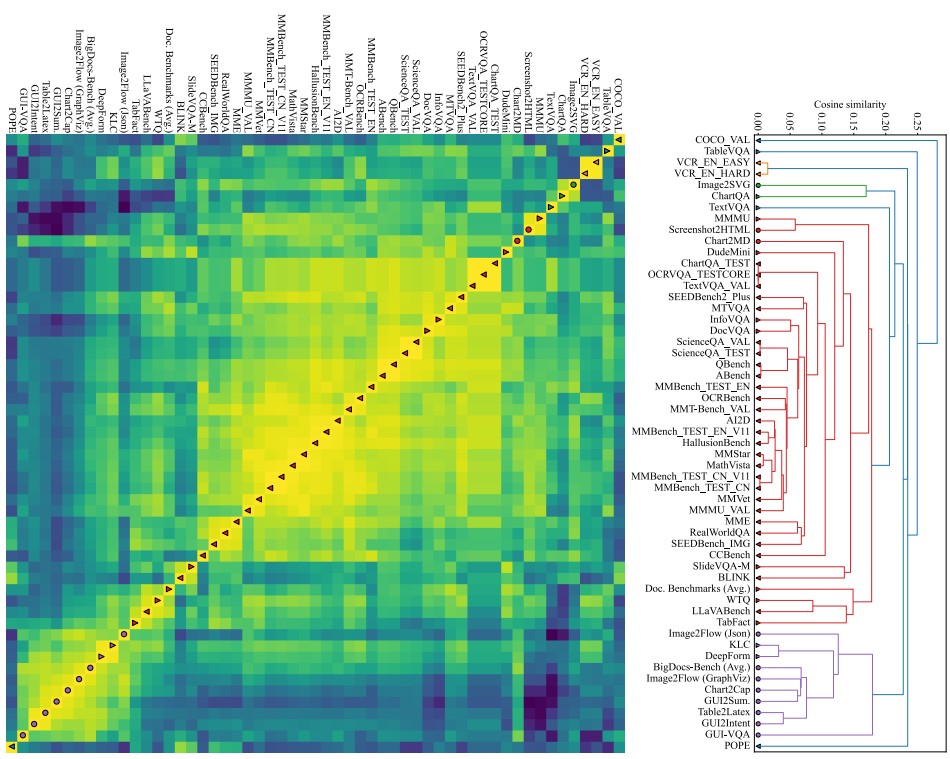

**Figure 16:** *Correlation matrix and dendrogram for BigDocs-Bench and General VLM Benchmarks. The matrix (left) illustrates benchmark relationships, with BigDocs-Bench tasks (circles) clustering separately from other benchmarks. The dendrogram (right) reveals three main clusters: general VLM tasks (e.g., COCO, VCR), document understanding tasks, and GUI/table-related tasks (such as GUI2Sum and Table2Latex).*

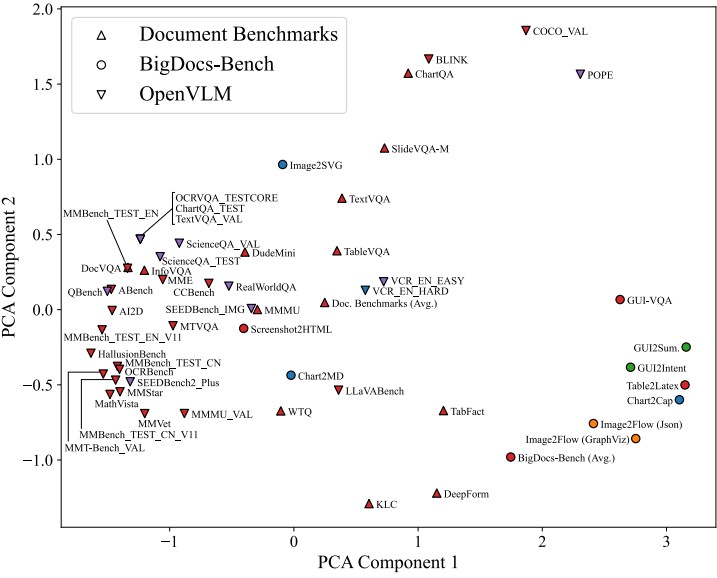

**Figure 17:** *PCA of BigDocs-Bench and General VLM Benchmarks. BigDocs-Bench (circles) form distinct clusters, clearly separated from other benchmarks, highlighting its unique characteristics. A dense cluster is observed for general VLM benchmarks (downward triangles), while document benchmarks (upward triangles) are more dispersed.*

**Table 9:** *Test Example for the task of Chart2Caption. All models other than Phi3.5-V-4B make mistakes in the caption. LLaVA-NeXT-7B does not offer relevant information about the chart, but about its format and LLaVA-NeXT-7B + BigDocs hallucinates. BigDocs models make factual mistakes, while Claude and GPT4o have trouble with the accent marks.*

| Component | Content |
|---|---|
| **Input Image** | 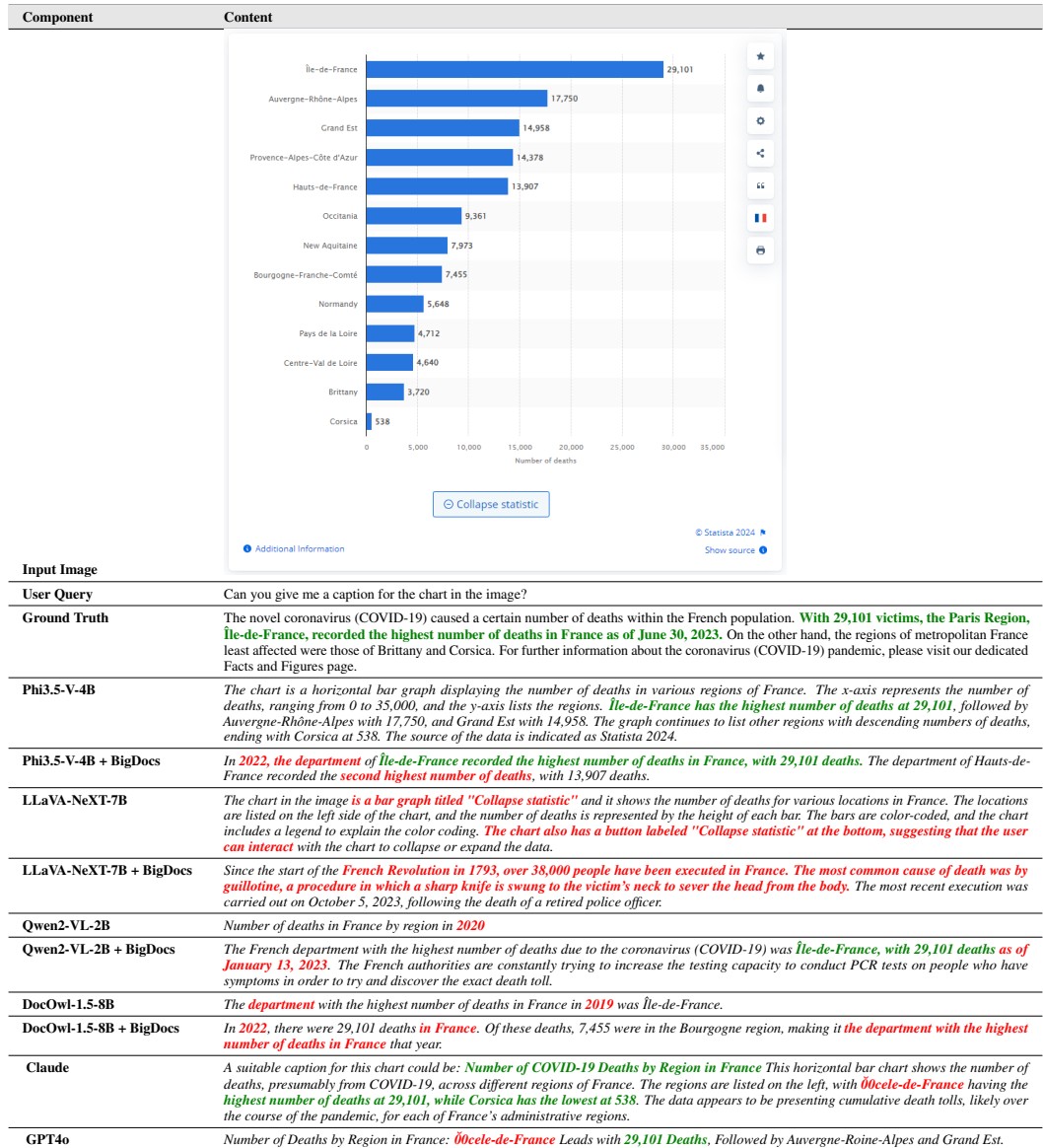 |
| **User Query** | Can you give me a caption for the chart in the image? |
| **Ground Truth** | The novel coronavirus (COVID-19) caused a certain number of deaths within the French population. **With 29,101 victims, the Paris Region, Île-de-France, recorded the highest number of deaths in France as of June 30, 2023.** On the other hand, the regions of metropolitan France least affected were those of Brittany and Corsica. For further information about the coronavirus (COVID-19) pandemic, please visit our dedicated Facts and Figures page. |
| **Phi3.5-V-4B** | *The chart is a horizontal bar graph displaying the number of deaths in various regions of France. The x-axis represents the number of deaths, ranging from 0 to 35,000, and the y-axis lists the regions. Île-de-France has the highest number of deaths at 29,101, followed by Auvergne-Rhône-Alpes with 17,750, and Grand Est with 14,958. The graph continues to list other regions with descending numbers of deaths, ending with Corsica at 538. The source of the data is indicated as Statista 2024.* |
| **Phi3.5-V-4B + BigDocs** | *In 2022, the department of Île-de-France recorded the highest number of deaths in France, with 29,101 deaths. The department of Hauts-de-France recorded the second highest number of deaths, with 13,907 deaths.* |
| **LLaVA-NeXT-7B** | *The chart in the image is a bar graph titled "Collapse statistic" and it shows the number of deaths for various locations in France. The locations are listed on the left side of the chart, and the number of deaths is represented by the height of each bar. The bars are color-coded, and the chart includes a legend to explain the color coding. The chart also has a button labeled "Collapse statistic" at the bottom, suggesting that the user can interact with the chart to collapse or expand the data.* |
| **LLaVA-NeXT-7B + BigDocs** | *Since the start of the French Revolution in 1793, over 38,000 people have been executed in France. The most common cause of death was by guillotine, a procedure in which a sharp knife is swung to the victim's neck to sever the head from the body. The most recent execution was carried out on October 5, 2023, following the death of a retired police officer.* |
| **Qwen2-VL-2B** | *Number of deaths in France by region in 2020* |
| **Qwen2-VL-2B + BigDocs** | *The French department with the highest number of deaths due to the coronavirus (COVID-19) was Île-de-France, with 29,101 deaths as of January 13, 2023. The French authorities are constantly trying to increase the testing capacity to conduct PCR tests on people who have symptoms in order to try and discover the exact death toll.* |
| **DocOwl-1.5-8B** | *The department with the highest number of deaths in France in 2019 was Île-de-France.* |
| **DocOwl-1.5-8B + BigDocs** | *In 2022, there were 29,101 deaths in France. Of these deaths, 7,455 were in the Bourgogne region, making it the department with the highest number of deaths in France that year.* |
| **Claude** | *A suitable caption for this chart could be: Number of COVID-19 Deaths by Region in France This horizontal bar chart shows the number of deaths, presumably from COVID-19, across different regions of France. The regions are listed on the left, with Ô0cele-de-France having the highest number of deaths at 29,101, while Corsica has the lowest at 538. The data appears to be presenting cumulative death tolls, likely over the course of the pandemic, for each of France's administrative regions.* |
| **GPT4o** | *Number of Deaths by Region in France: Ô0cele-de-France Leads with 29,101 Deaths, Followed by Auvergne-Roine-Alpes and Grand Est.* |

**Table 10:** *Chart2Caption: Comparison of model responses showing that models using the BigDocs dataset produce more concise and relevant captions than baseline models while still making some factual errors.*

| Component | Content |
|---|---|
| 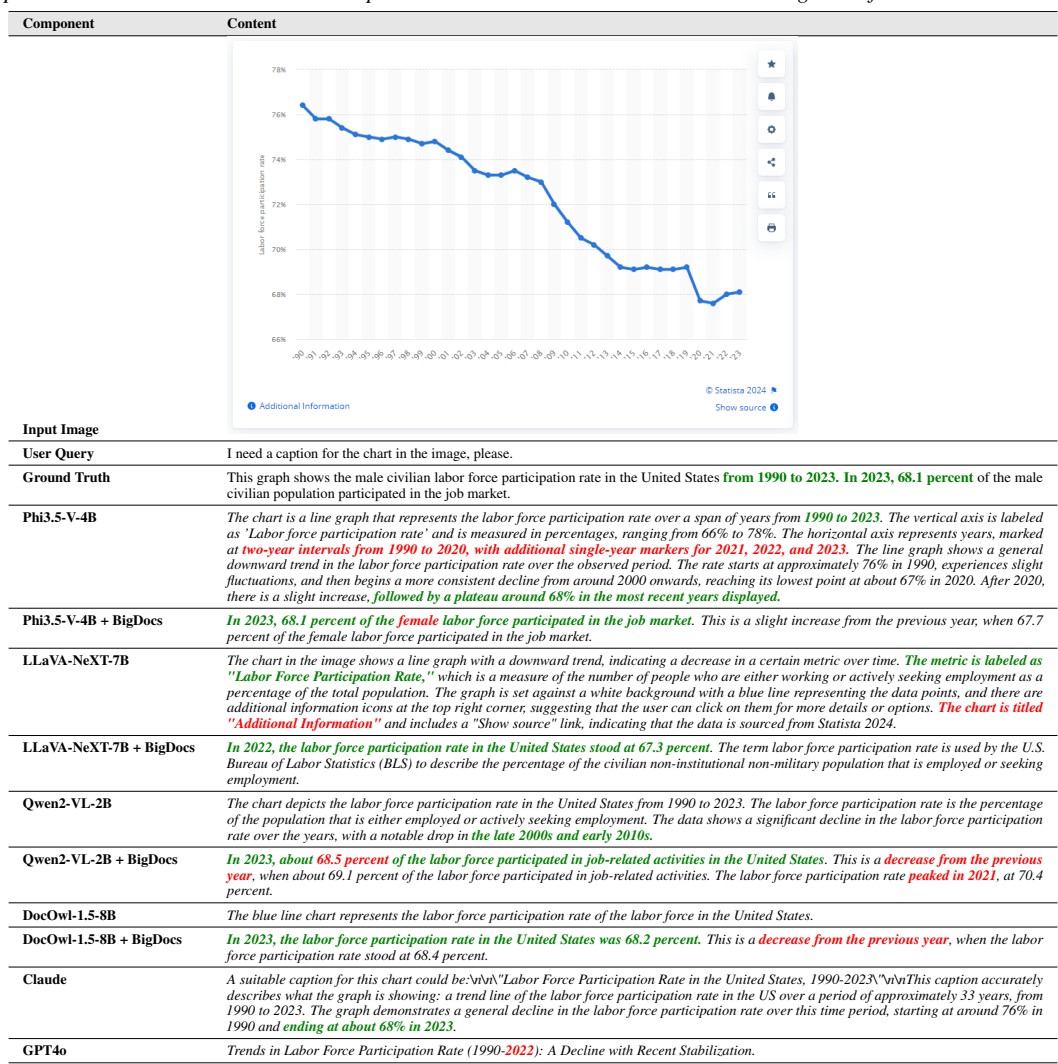 | |
| **Input Image** | |
| **User Query** | I need a caption for the chart in the image, please. |
| **Ground Truth** | This graph shows the male civilian labor force participation rate in the United States **from 1990 to 2023. In 2023, 68.1 percent** of the male civilian population participated in the job market. |
| **Phi3.5-V-4B** | *The chart is a line graph that represents the labor force participation rate over a span of years from 1990 to 2023. The vertical axis is labeled as 'Labor force participation rate' and is measured in percentages, ranging from 66% to 78%. The horizontal axis represents years, marked at two-year intervals from 1990 to 2020, with additional single-year markers for 2021, 2022, and 2023. The line graph shows a general downward trend in the labor force participation rate over the observed period. The rate starts at approximately 76% in 1990, experiences slight fluctuations, and then begins a more consistent decline from around 2000 onwards, reaching its lowest point at about 67% in 2020. After 2020, there is a slight increase, followed by a plateau around 68% in the most recent years displayed.* |
| **Phi3.5-V-4B + BigDocs** | *In 2023, 68.1 percent of the female labor force participated in the job market. This is a slight increase from the previous year, when 67.7 percent of the female labor force participated in the job market.* |
| **LLaVA-NeXT-7B** | *The chart in the image shows a line graph with a downward trend, indicating a decrease in a certain metric over time. The metric is labeled as "Labor Force Participation Rate," which is a measure of the number of people who are either working or actively seeking employment as a percentage of the total population. The graph is set against a white background with a blue line representing the data points, and there are additional information icons at the top right corner, suggesting that the user can click on them for more details or options. The chart is titled "Additional Information" and includes a "Show source" link, indicating that the data is sourced from Statista 2024.* |
| **LLaVA-NeXT-7B + BigDocs** | *In 2022, the labor force participation rate in the United States stood at 67.3 percent. The term labor force participation rate is used by the U.S. Bureau of Labor Statistics (BLS) to describe the percentage of the civilian non-institutional non-military population that is employed or seeking employment.* |
| **Qwen2-VL-2B** | *The chart depicts the labor force participation rate in the United States from 1990 to 2023. The labor force participation rate is the percentage of the population that is either employed or actively seeking employment. The data shows a significant decline in the labor force participation rate over the years, with a notable drop in the late 2000s and early 2010s.* |
| **Qwen2-VL-2B + BigDocs** | *In 2023, about 68.5 percent of the labor force participated in job-related activities in the United States. This is a decrease from the previous year, when about 69.1 percent of the labor force participated in job-related activities. The labor force participation rate peaked in 2021, at 70.4 percent.* |
| **DocOwl-1.5-8B** | *The blue line chart represents the labor force participation rate of the labor force in the United States.* |
| **DocOwl-1.5-8B + BigDocs** | *In 2023, the labor force participation rate in the United States was 68.2 percent. This is a decrease from the previous year, when the labor force participation rate stood at 68.4 percent.* |
| **Claude** | *A suitable caption for this chart could be:\n\n"Labor Force Participation Rate in the United States, 1990-2023"\n\nThis caption accurately describes what the graph is showing: a trend line of the labor force participation rate in the US over a period of approximately 33 years, from 1990 to 2023. The graph demonstrates a general decline in the labor force participation rate over this time period, starting at around 76% in 1990 and ending at about 68% in 2023.* |
| **GPT4o** | *Trends in Labor Force Participation Rate (1990-2022): A Decline with Recent Stabilization.* |

**Table 11:** *Evaluation Example for GUI-VQA: Models utilizing the BigDocs dataset consistently produce the most accurate and concise outputs, perfectly matching the ground truth without any extraneous or unnecessary information.*

| Component | Content |
|---|---|
| **Input Image** | 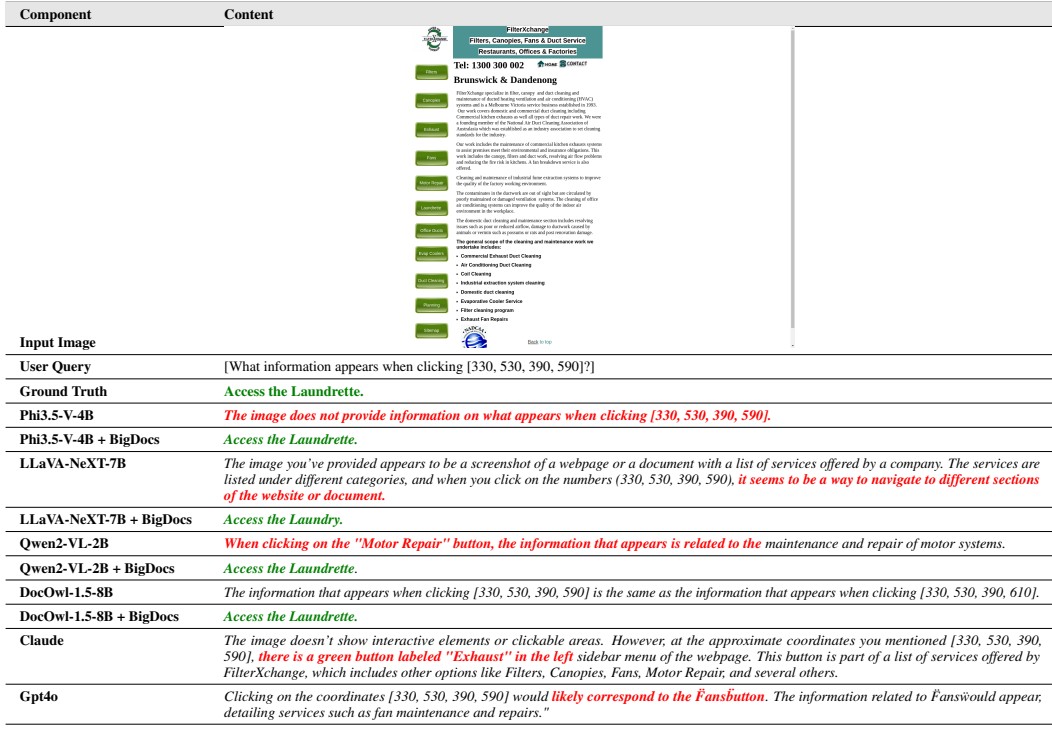 |
| **User Query** | [What information appears when clicking [330, 530, 390, 590]?] |
| **Ground Truth** | *Access the Laundrette.* |
| **Phi3.5-V-4B** | *The image does not provide information on what appears when clicking [330, 530, 390, 590].* |
| **Phi3.5-V-4B + BigDocs** | *Access the Laundrette.* |
| **LLaVA-NeXT-7B** | *The image you've provided appears to be a screenshot of a webpage or a document with a list of services offered by a company. The services are listed under different categories, and when you click on the numbers (330, 530, 390, 590), it seems to be a way to navigate to different sections of the website or document.* |
| **LLaVA-NeXT-7B + BigDocs** | *Access the Laundry.* |
| **Qwen2-VL-2B** | *When clicking on the "Motor Repair" button, the information that appears is related to the maintenance and repair of motor systems.* |
| **Qwen2-VL-2B + BigDocs** | *Access the Laundrette.* |
| **DocOwl-1.5-8B** | *The information that appears when clicking [330, 530, 390, 590] is the same as the information that appears when clicking [330, 530, 390, 610].* |
| **DocOwl-1.5-8B + BigDocs** | *Access the Laundrette.* |
| **Claude** | *The image doesn't show interactive elements or clickable areas. However, at the approximate coordinates you mentioned [330, 530, 390, 590], there is a green button labeled "Exhaust" in the left sidebar menu of the webpage. This button is part of a list of services offered by FilterXchange, which includes other options like Filters, Canopies, Fans, Motor Repair, and several others.* |
| **Gpt4o** | *Clicking on the coordinates [330, 530, 390, 590] would likely correspond to the "Fans" button. The information related to "Fans" would appear, detailing services such as fan maintenance and repairs."* |

**Table 12:** *Qualitative test example for the task of GUI-VQA. GPT-4o (20240806), LLaVA-NeXT-7B , Phi3.5-V-4B , and Qwen2-VL-2B tend to hallucinate the response, mostly ignoring the input image. LLaVA-NeXT-7B and Claude are more conversational, however, the latter predicts correctly that the answer is 'Home Security'*

| Component | Content |
|---|---|
| **Input Image** | 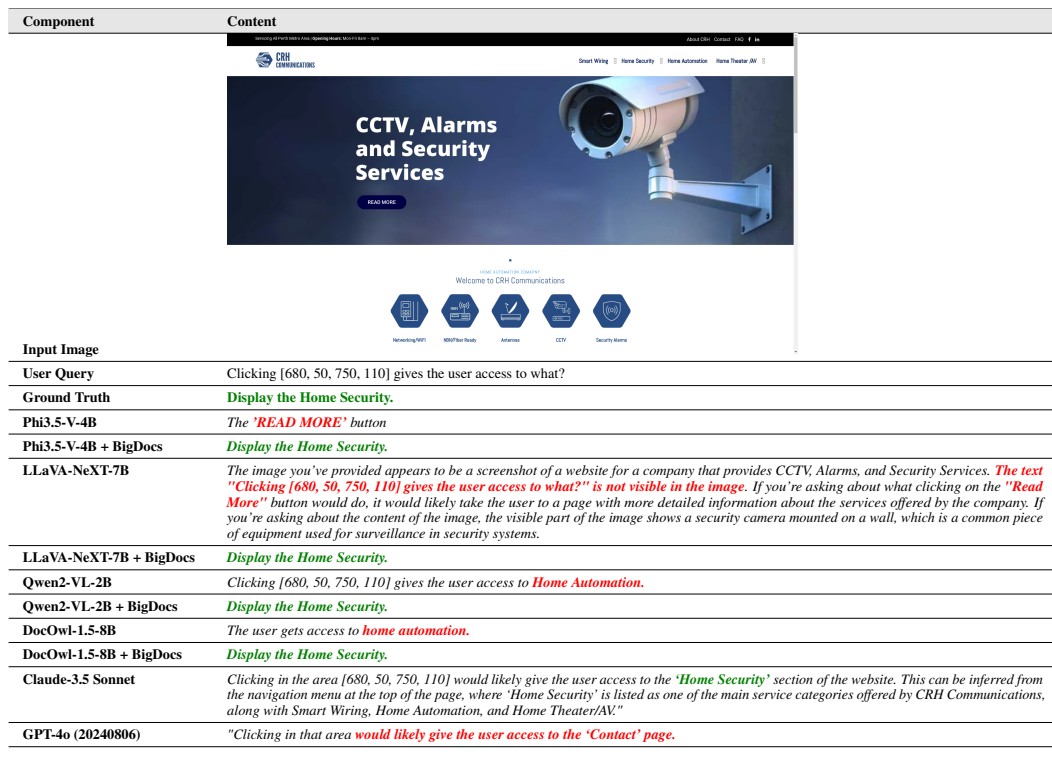 |
| **User Query** | Clicking [680, 50, 750, 110] gives the user access to what? |
| **Ground Truth** | *Display the Home Security.* |
| **Phi3.5-V-4B** | *The 'READ MORE' button* |
| **Phi3.5-V-4B + BigDocs** | *Display the Home Security.* |
| **LLaVA-NeXT-7B** | *The image you've provided appears to be a screenshot of a website for a company that provides CCTV, Alarms, and Security Services. The text "Clicking [680, 50, 750, 110] gives the user access to what?" is not visible in the image. If you're asking about what clicking on the "Read More" button would do, it would likely take the user to a page with more detailed information about the services offered by the company. If you're asking about the content of the image, the visible part of the image shows a security camera mounted on a wall, which is a common piece of equipment used for surveillance in security systems.* |
| **LLaVA-NeXT-7B + BigDocs** | *Display the Home Security.* |
| **Qwen2-VL-2B** | *Clicking [680, 50, 750, 110] gives the user access to Home Automation.* |
| **Qwen2-VL-2B + BigDocs** | *Display the Home Security.* |
| **DocOwl-1.5-8B** | *The user gets access to home automation.* |
| **DocOwl-1.5-8B + BigDocs** | *Display the Home Security.* |
| **Claude-3.5 Sonnet** | *Clicking in the area [680, 50, 750, 110] would likely give the user access to the 'Home Security' section of the website. This can be inferred from the navigation menu at the top of the page, where 'Home Security' is listed as one of the main service categories offered by CRH Communications, along with Smart Wiring, Home Automation, and Home Theater/AV."* |
| **GPT-4o (20240806)** | *"Clicking in that area would likely give the user access to the 'Contact' page.* |

**Table 13:** *Qualitative test example for the task of GUI2Intent. All the models trained with BigDocs along with Qwen2-VL-2B produce the correct intent, while the latter is more conversational. Phi3.5-V-4B , LLaVA-NeXT-7B and DocOwl-1.5-8B infer incorrect sections of the image.*

| Component | Content |
|---|---|
| Input Image | 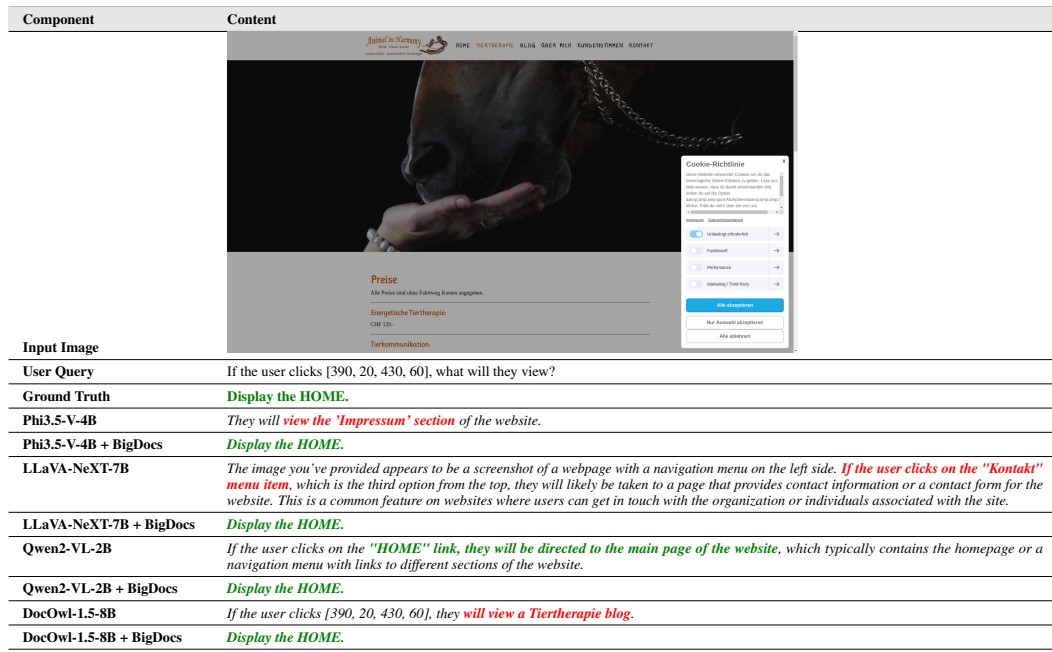 |
| User Query | If the user clicks [390, 20, 430, 60], what will they view? |
| Ground Truth | **Display the HOME.** |
| Phi3.5-V-4B | *They will **view the 'Impressum' section** of the website.* |
| Phi3.5-V-4B + BigDocs | ***Display the HOME.*** |
| LLaVA-NeXT-7B | *The image you've provided appears to be a screenshot of a webpage with a navigation menu on the left side. **If the user clicks on the "Kontakt" menu item**, which is the third option from the top, they will likely be taken to a page that provides contact information or a contact form for the website. This is a common feature on websites where users can get in touch with the organization or individuals associated with the site.* |
| LLaVA-NeXT-7B + BigDocs | ***Display the HOME.*** |
| Qwen2-VL-2B | *If the user clicks on the **"HOME" link, they will be directed to the main page of the website**, which typically contains the homepage or a navigation menu with links to different sections of the website.* |
| Qwen2-VL-2B + BigDocs | ***Display the HOME.*** |
| DocOwl-1.5-8B | *If the user clicks [390, 20, 430, 60], they **will view a Tiertherapie blog**.* |
| DocOwl-1.5-8B + BigDocs | ***Display the HOME.*** |

**Table 14:** *Qualitative test example for the task of GUI2Intent. All the models trained with BigDocs along with Qwen2-VL-2B produce the correct intent, while the latter is more conversational. Phi3.5-V-4B , LLaVA-NeXT-7B and DocOwl-1.5-8B provide confused responses.*

| Component | Content |
|---|---|
| Input Image | |
| User Query | What will the user encounter by clicking [400, 160, 450, 190]? |
| Ground Truth | **View the About Us.** |
| Phi3.5-V-4B | *The user will encounter a section of the website that is not fully visible in the image.* |
| Phi3.5-V-4B + BigDocs | ***View the About Us.*** |
| LLaVA-NeXT-7B | *The image you've provided appears to be a screenshot of a website, specifically a dental clinic's website. By clicking on the coordinates you've mentioned (400, 160, 450, 190), **the user would likely be directed to a specific section or page of the website. However, without more context or a clearer view of the website's layout**, it's difficult to determine exactly what the user would encounter. It could be a link to a specific service offered by the clinic, a page about a particular dental procedure, a contact form, or any other type of content that is relevant to the clinic's services.* |
| LLaVA-NeXT-7B + BigDocs | ***View the About Us.*** |
| Qwen2-VL-2B | *The user will be **directed to the "About Us"** section of the Menai Dental Centre website.* |
| Qwen2-VL-2B + BigDocs | ***View the About Us.*** |
| DocOwl-1.5-8B | *The user will encounter a phone number by clicking on the respective number.* |
| DocOwl-1.5-8B + BigDocs | ***View the About Us.*** |

**Table 15:** *Qualitative test example for the task GUI2Sum. BigDocs models accurately capture the spatial orientation and stylistic aspects of of the webpage while making errors parsing the restaurant name. LLaVA-NeXT-7B expresses spatial disorientation, while Qwen2-VL-2B and DocOwl-1.5-8B provide too simplistic and incorrect responses respectively.*

| Component | Content |
|---|---|
| |  |
| **Input Image** | |
| **User Query** | Can you give me a caption for the webpage in the image? |
| **Ground Truth** | This website features a modern and colorful food menu for Gunbaje, a restaurant offering Korean tapas and small plates. **The layout is clean and organized, with a prominent "FOOD MENU" title at the top. The menu is divided into sections for TAPAS, DINNER, and FEED ME, each with a distinct color background and clear pricing.** The drinks menu, labeled "DRINK MENU," is displayed below with a vibrant design and includes beer, soju, and cocktails. The website also includes social media icons and a login button, enhancing user engagement. |
| **Phi3.5-V-4B** | *The image displays a webpage with a food and drink menu for **a restaurant named Gunbae**. The top of the page has navigation options **such as 'Home', 'Gunbae Menu', 'Order Online', 'Location & Hours', 'Book now', and social media icons for Instagram and Facebook**. There is also a 'Log In' option. The main content of the page is divided into two sections: 'FOOD MENU' and 'DRINK MENU'. The food menu is colorful and lists various dishes with prices, including 'TAPAS', 'DINNER', 'FEED ME', and 'SMALL PLATES'. It also mentions 'WESTERN', 'GUNBAE FREE', and 'VEGAN' options. The drink menu is partially visible, showing 'BEER' and 'SOJU' categories with specific beer brands and prices. The bottom of the image is cut off, and the full drink menu is not visible.* |
| **Phi3.5-V-4B + BigDocs** | *This website features a food menu for a restaurant named **""Gunbice.""** The main content includes a colorful and organized menu with various **dishes categorized under ""TAPAS,"" ""FOOD,"" and ""DRINK.""** The layout is clean and easy to navigate, with a beige background and a mix of vibrant and pastel colors for the menu items. **The website also includes social media icons and a login section at the top.** The overall style is modern and visually appealing, designed to attract customers with a variety of food and drink options.* |
| **LLaVA-NeXT-7B** | *The image shows a webpage **with a food menu on the left and a drink menu on the right**. The food menu lists various dishes with their prices, and the drink menu displays a selection of beverages with their respective prices. The **design is clean and organized, with a color scheme that includes shades of blue, green, and orange**. The layout is user-friendly, with clear headings and easy-to-read text. The overall impression is that of a modern, inviting restaurant or cafe website.* |
| **LLaVA-NeXT-7B + BigDocs** | *This website features a vibrant and modern menu for a restaurant named **"Rockin Randy."** The layout is clean and organized, with a beige background that contrasts with the colorful menu items. The main content includes a detailed food menu with sections for beer, cider, and drinks, each listing various options with prices and descriptions. Icons and graphics, such as a beer tap and a drink, are used to visually represent the menu items. **The website also includes navigation options like "Home," "Gastrobar Menu," "Order Online," "Location & Hours," and "Book now,"** along with social media icons and a login button. The overall style is contemporary and user-friendly, making it easy for customers to browse and order their drinks.* |
| **Qwen2-VL-2B** | *"Certainly! Here's a caption for the webpage: ""Experience the flavors of Gunbae with our delicious food and refreshing drinks!"""* |
| **Qwen2-VL-2B + BigDocs** | *This website is a food menu page for Gunbae, featuring a modern and clean layout with a dark background and white text. The main content **includes a "FOOD MENU" section with various categories like Tapas, Small Plates, Buffets, and Desserts.** The layout is organized with a navigation bar at the top, a large food menu area, and a "DRINK MENU" section below. The color scheme is primarily dark with accents of green and white, creating a sophisticated and inviting atmosphere. The website also includes social media icons and a login button for users.* |
| **DocOwl-1.5-8B** | *"Food Menu" is the caption for the webpage in the image.* |
| **DocOwl-1.5-8B + BigDocs** | *This website features a food menu for a restaurant **named "Gunbac."** The layout is clean and organized, with a beige background and a mix of black, white, and orange text. The main content includes a detailed menu with **various categories such as appetizers, mains, and drinks, each with pricing and descriptions.** The menu items are visually appealing with images and icons. The website also has navigation options for Home, **Gunbac Menu**, Order Online, Location & Hours, and Book now. Social media icons for Facebook and Instagram are present at the top right.* |

**Table 16:** *Qualitative test example for GUI2Sum. Phi3.5-V-4B and the BigDocs models are able to extract correct and relevant information from this simple and clean webpage, while the other baseline models make mistakes or are too simplistic in their caption.*

| Component | Content |
|---|---|
| | 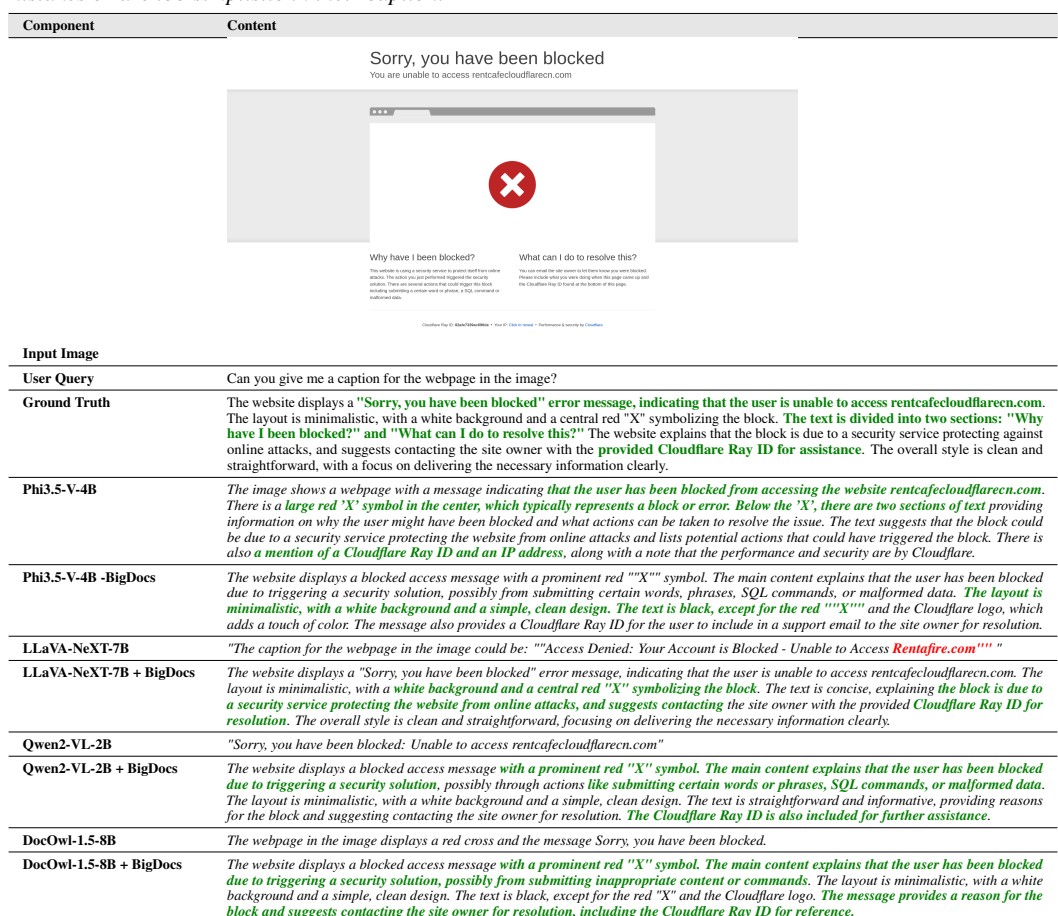 |
| **Input Image** | |
| **User Query** | Can you give me a caption for the webpage in the image? |
| **Ground Truth** | The website displays a **"Sorry, you have been blocked" error message, indicating that the user is unable to access rentcafecloudflarecn.com**. The layout is minimalistic, with a white background and a central red "X" symbolizing the block. **The text is divided into two sections: "Why have I been blocked?" and "What can I do to resolve this?"** The website explains that the block is due to a security service protecting against online attacks, and suggests contacting the site owner with the **provided Cloudflare Ray ID for assistance**. The overall style is clean and straightforward, with a focus on delivering the necessary information clearly. |
| **Phi3.5-V-4B** | *The image shows a webpage with a message indicating **that the user has been blocked from accessing the website rentcafecloudflarecn.com**. There is a **large red 'X' symbol in the center, which typically represents a block or error. Below the 'X', there are two sections of text** providing information on why the user might have been blocked and what actions can be taken to resolve the issue. The text suggests that the block could be due to a security service protecting the website from online attacks and lists potential actions that could have triggered the block. There is also **a mention of a Cloudflare Ray ID and an IP address**, along with a note that the performance and security are by Cloudflare.* |
| **Phi3.5-V-4B -BigDocs** | *The website displays a blocked access message with a prominent red ""X"" symbol. The main content explains that the user has been blocked due to triggering a security solution, possibly from submitting certain words, phrases, SQL commands, or malformed data. **The layout is minimalistic, with a white background and a simple, clean design. The text is black, except for the red ""X""** and the Cloudflare logo, which adds a touch of color. The message also provides a Cloudflare Ray ID for the user to include in a support email to the site owner for resolution.* |
| **LLaVA-NeXT-7B** | *"The caption for the webpage in the image could be: ""Access Denied: Your Account is Blocked - Unable to Access **Rentafire.com**"" "* |
| **LLaVA-NeXT-7B + BigDocs** | *The website displays a "Sorry, you have been blocked" error message, indicating that the user is unable to access rentcafecloudflarecn.com. The layout is minimalistic, with a **white background and a central red "X" symbolizing the block**. The text is concise, explaining **the block is due to a security service protecting the website from online attacks, and suggests contacting** the site owner with the provided **Cloudflare Ray ID for resolution**. The overall style is clean and straightforward, focusing on delivering the necessary information clearly.* |
| **Qwen2-VL-2B** | *"Sorry, you have been blocked: Unable to access rentcafecloudflarecn.com"* |
| **Qwen2-VL-2B + BigDocs** | *The website displays a blocked access message **with a prominent red "X" symbol. The main content explains that the user has been blocked due to triggering a security solution**, possibly through actions **like submitting certain words or phrases, SQL commands, or malformed data**. The layout is minimalistic, with a white background and a simple, clean design. The text is straightforward and informative, providing reasons for the block and suggesting contacting the site owner for resolution. **The Cloudflare Ray ID is also included for further assistance**.* |
| **DocOwl-1.5-8B** | *The webpage in the image displays a red cross and the message Sorry, you have been blocked.* |
| **DocOwl-1.5-8B + BigDocs** | *The website displays a blocked access message **with a prominent red "X" symbol. The main content explains that the user has been blocked due to triggering a security solution, possibly from submitting inappropriate content or commands**. The layout is minimalistic, with a white background and a simple, clean design. The text is black, except for the red "X" and the Cloudflare logo. **The message provides a reason for the block and suggests contacting the site owner for resolution, including the Cloudflare Ray ID for reference.*** |

**Table 17:** *Test Example for the task of Table2Latex. BigDocs models preserve the structure of the table, lines from rows and columns, and font styles like bolds or italics.*

| Component | Content |
|---|---|
| | |
| **Input Image** | <table><tr><td></td><td colspan="4">Discrete dynamics(ACC)</td><td colspan="3">Continuous dynamics($\sigma$)</td></tr><tr><td></td><td>SIR</td><td>SIS</td><td>Threshold</td><td>Kirman</td><td>Gene</td><td>Mutualistic</td><td>CML</td></tr><tr><td>$T+1$</td><td>0.85</td><td>0.86</td><td>0.89</td><td>0.84</td><td>0.598</td><td>0.958</td><td>0.017</td></tr><tr><td>$T+2$</td><td>0.73</td><td>0.80</td><td>0.84</td><td>0.81</td><td>0.602</td><td>1.086</td><td>0.021</td></tr><tr><td>$T+3$</td><td>0.81</td><td>0.75</td><td>0.81</td><td>0.82</td><td>0.609</td><td>1.276</td><td>0.024</td></tr><tr><td>$T+4$</td><td>0.82</td><td>0.74</td><td>0.74</td><td>0.83</td><td>0.724</td><td>1.512</td><td>0.027</td></tr><tr><td>$T+5$</td><td>0.80</td><td>0.74</td><td>0.72</td><td>0.85</td><td>0.822</td><td>1.601</td><td>0.028</td></tr></table> |
| **User Query** | [Please generate the necessary LaTeX script to draw this table.] |
| **Phi3.5-V-4B** | <table><tr><td>Discrete dynamics (ACC)</td><td>SIR</td><td>SIS</td><td>Threshold</td><td>Kirman</td><td>Gene</td><td>Mutualistic</td><td>CML</td></tr><tr><td>$T+1$</td><td>0.85</td><td>0.86</td><td>0.89</td><td>0.84</td><td>0.598</td><td>0.958</td><td>0.017</td></tr><tr><td>$T+2$</td><td>0.73</td><td>0.80</td><td>0.84</td><td>0.81</td><td>0.602</td><td>1.086</td><td>0.021</td></tr><tr><td>$T+3$</td><td>0.81</td><td>0.75</td><td>0.81</td><td>0.82</td><td>0.609</td><td>1.276</td><td>0.024</td></tr><tr><td>$T+4$</td><td>0.82</td><td>0.74</td><td>0.74</td><td>0.83</td><td>0.724</td><td>1.512</td><td>0.027</td></tr><tr><td>$T+5$</td><td>0.80</td><td>0.74</td><td>0.72</td><td>0.85</td><td>0.822</td><td>1.601</td><td>0.028</td></tr></table>
Table 1: Comparison of different models for discrete and continuous dynamics. |
| **Phi3.5-V-4B + BigDocs** | <table><tr><td></td><td colspan="3">Discrete dynamics(ACC)</td><td colspan="3">Continuous dynamics($\sigma$)</td></tr><tr><td></td><td>SIR</td><td>SIS</td><td>Threshold  Kirman</td><td>Gene</td><td>Mutualistic</td><td>CML</td></tr><tr><td>$T+1$</td><td>0.85</td><td>0.86</td><td>0.89      0.84</td><td>0.598</td><td>0.958</td><td>0.017</td></tr><tr><td>$T+2$</td><td>0.73</td><td>0.80</td><td>0.84      0.81</td><td>0.602</td><td>1.086</td><td>0.021</td></tr><tr><td>$T+3$</td><td>0.81</td><td>0.75</td><td>0.81      0.82</td><td>0.609</td><td>1.276</td><td>0.024</td></tr><tr><td>$T+4$</td><td>0.82</td><td>0.74</td><td>0.74      0.83</td><td>0.724</td><td>1.512</td><td>0.027</td></tr><tr><td>$T+5$</td><td>0.80</td><td>0.74</td><td>0.72      0.85</td><td>0.822</td><td>1.601</td><td>0.028</td></tr></table> |
| **Gpt4o** | <table><tr><td></td><td colspan="4">Discrete dynamics (ACC)</td><td colspan="3">Continuous dynamics ($\sigma$)</td></tr><tr><td></td><td>SIR</td><td>SIS</td><td>Threshold</td><td>Kirman</td><td>Gene</td><td>Mutualistic</td><td>CML</td></tr><tr><td>$T+1$</td><td>0.85</td><td>0.86</td><td>0.89</td><td>0.84</td><td>0.598</td><td>0.958</td><td>0.017</td></tr><tr><td>$T+2$</td><td>0.73</td><td>0.80</td><td>0.84</td><td>0.81</td><td>0.602</td><td>1.086</td><td>0.021</td></tr><tr><td>$T+3$</td><td>0.81</td><td>0.75</td><td>0.81</td><td>0.82</td><td>0.609</td><td>1.276</td><td>0.024</td></tr><tr><td>$T+4$</td><td>0.82</td><td>0.74</td><td>0.74</td><td>0.83</td><td>0.724</td><td>1.512</td><td>0.027</td></tr><tr><td>$T+5$</td><td>0.80</td><td>0.74</td><td>0.72</td><td>0.85</td><td>0.822</td><td>1.601</td><td>0.028</td></tr></table> |

**Table 18:** *Test Example for the task of Table2Latex. Phi3.5-V-4B was not able to format the table with all the rows, and used incorrect lines, and font style (boldness of first column and row).*

| Component | Content |
|---|---|
| **Input Image** | <table><tr><td></td><td>Chain 2</td><td>Chain 3</td><td>Chain 4</td><td>Chain 5</td></tr><tr><td>**Chain 1**</td><td>0.81</td><td>0.70</td><td>0.83</td><td>0.78</td></tr><tr><td>**Chain 2**</td><td>1</td><td>0.61</td><td>0.76</td><td>0.67</td></tr><tr><td>**Chain 3**</td><td></td><td>1</td><td>0.67</td><td>0.60</td></tr><tr><td>**Chain 4**</td><td></td><td></td><td>1</td><td>0.76</td></tr></table> |
| **User Query** | [Create a LaTeX script to illustrate this table.] |
| **Phi3.5-V-4B** | <table><tr><td>Chain 1</td><td>Chain 2</td><td>Chain 3</td><td>Chain 4</td><td>Chain 5</td></tr><tr><td>0.81</td><td>0.70</td><td>0.83</td><td>0.78</td><td>0.67</td></tr><tr><td>1</td><td>0.61</td><td>0.67</td><td>0.60</td><td>0.76</td></tr></table>
Table 1: Table of Chains |
| **Phi3.5-V-4B + BigDocs** | <table><tr><td></td><td>Chain 2</td><td>Chain 3</td><td>Chain 4</td><td>Chain 5</td></tr><tr><td>**Chain 1**</td><td>0.81</td><td>0.70</td><td>0.83</td><td>0.78</td></tr><tr><td>**Chain 2**</td><td>1</td><td>0.61</td><td>0.76</td><td>0.67</td></tr><tr><td>**Chain 3**</td><td></td><td>1</td><td>0.67</td><td>0.60</td></tr><tr><td>**Chain 4**</td><td></td><td></td><td>1</td><td>0.76</td></tr></table> |

**Table 19:** *Test Example for the task of Table2Latex: Boldness and details of lines are perfectly aligned with input in the Phi3.5-V-4B + BigDocs example.*

| Component | Content |
|---|---|
| Input Image | <table><tr><td>**Logic Gates**</td><td>**(0,0)**</td><td>**(0,1)/(1,0)**</td><td>**(1,1)**</td></tr><tr><td>**AND**</td><td>0</td><td>0</td><td>1</td></tr><tr><td>**NAND**</td><td>1</td><td>1</td><td>0</td></tr><tr><td>**OR**</td><td>0</td><td>1</td><td>1</td></tr><tr><td>**NOR**</td><td>1</td><td>0</td><td>0</td></tr><tr><td>**XOR**</td><td>0</td><td>1</td><td>0</td></tr><tr><td>**XNOR**</td><td>1</td><td>0</td><td>1</td></tr></table> |
| User Query | [Please generate the necessary LaTeX script to draw this table.] |
| Phi3.5-V-4B | <table><tr><td>**Logic Gates**</td><td>**(0,0)**</td><td>**(0,1)/(1,0)**</td><td>**(1,1)**</td></tr><tr><td>AND</td><td>0</td><td>0</td><td>1</td></tr><tr><td>NAND</td><td>1</td><td>1</td><td>0</td></tr><tr><td>OR</td><td>0</td><td>1</td><td>1</td></tr><tr><td>NOR</td><td>1</td><td>0</td><td>0</td></tr><tr><td>XOR</td><td>0</td><td>1</td><td>0</td></tr><tr><td>XNOR</td><td>1</td><td>0</td><td>1</td></tr></table> |
| Phi3.5-V-4B + BigDocs | <table><tr><td>**Logic Gates**</td><td>**(0,0)**</td><td>**(0,1)/(1,0)**</td><td>**(1,1)**</td></tr><tr><td>**AND**</td><td>0</td><td>0</td><td>1</td></tr><tr><td>**NAND**</td><td>1</td><td>1</td><td>0</td></tr><tr><td>**OR**</td><td>0</td><td>1</td><td>1</td></tr><tr><td>**NOR**</td><td>1</td><td>0</td><td>0</td></tr><tr><td>**XOR**</td><td>0</td><td>1</td><td>0</td></tr><tr><td>**XNOR**</td><td>1</td><td>0</td><td>1</td></tr></table> |
| GPT4o | <table><tr><td>**Logic Gates**</td><td>**(0,0)**</td><td>**(0,1)/(1,0)**</td><td>**(1,1)**</td></tr><tr><td>AND</td><td>0</td><td>0</td><td>1</td></tr><tr><td>NAND</td><td>1</td><td>1</td><td>0</td></tr><tr><td>OR</td><td>0</td><td>1</td><td>1</td></tr><tr><td>NOR</td><td>1</td><td>0</td><td>0</td></tr><tr><td>XOR</td><td>0</td><td>1</td><td>0</td></tr><tr><td>XNOR</td><td>1</td><td>0</td><td>1</td></tr></table> |

**Table 20:** *Test Example for the task of Table2Latex BigDocs failure. Phi3.5-V-4B+BigDocs was not able to format the question marks in the first row.*

| Component | Content |
|---|---|
| Input Image | <table><tr><td>Method</td><td>RGDN [?]</td><td>PURE-LET [?]</td><td>DPIR [?]</td><td>DWDN [?]</td><td>**PhD-Net (Ours)**</td></tr><tr><td>PSNR</td><td>19.80</td><td>20.88</td><td>22.09</td><td>22.85</td><td>**23.48**</td></tr><tr><td>SSIM</td><td>0.476</td><td>0.501</td><td>0.548</td><td>0.561</td><td>**0.566**</td></tr></table> |
| User Query | [Please create LaTeX code to produce this table.] |
| Phi3.5-V-4B + BigDocs | <table><tr><td>Method</td><td>RGDN</td><td>PURE-LET</td><td>DPIR</td><td>DWDN</td><td>**PhD-Net (Ours)**</td></tr><tr><td>PSNR</td><td>19.80</td><td>20.88</td><td>22.09</td><td>22.85</td><td>**23.48**</td></tr><tr><td>SSIM</td><td>0.476</td><td>0.501</td><td>0.548</td><td>0.561</td><td>**0.566**</td></tr></table> |

**Table 21:** *Qualitative test example 1 for the DeepForm task. When asked a question that cannot be answered from the image, Phi3.5-V-4B + BigDocs(OCR reduction) hallucinates a response while Phi3.5-V-4B + BigDocs correctly answers in the negative, demonstrating the role of OCR data in enhancing precise information extraction.*

| Component | Content |
|---|---|
| |  |
| Input Image | |
| User Query | What is the value for the contract_num? |
| Ground Truth | None |
| Phi3.5-V-4B + BigDocs | None |
| Phi3.5-V-4B + BigDocs(OCR reduction) | 12-15 |

**Table 22:** *Qualitative test example 2 for the DeepForm task. When asked a question that cannot be answered from the image, Phi3.5-V-4B + BigDocs(OCR reduction) hallucinates an unrelated response from a different section of the image while Phi3.5-V-4B + BigDocs correctly answers in the negative.*

| Component | Content | |
|---|---|---|
| Input Image | 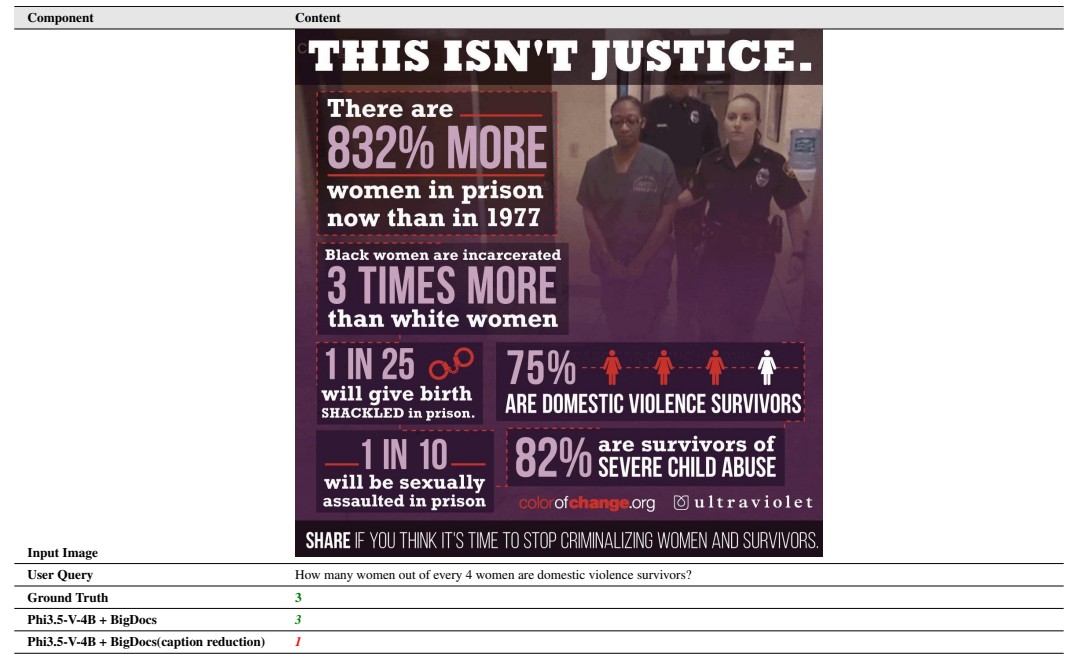 | |
| User Query | What is the value for the flight_from? | |
| Ground Truth | **None** | |
| Phi3.5-V-4B + BigDocs | *None* | |
| Phi3.5-V-4B + BigDocs(OCR reduction) | *National Media Group* | |

**Table 23:** *Qualitative test example 1 for the InfoVQA task. Phi3.5-V-4B + BigDocs is able to use the combination of the percentage and the pictorial representation of the relevant data and produce the correct count while Phi3.5-V-4B + BigDocs(caption reduction) gets confused with a different region of the image, demonstrating the robustness to text and image representations that captioning data brings.*

| Component | Content |
|---|---|
| Input Image | THIS ISN'T JUSTICE. There are 832% MORE women in prison now than in 1977. Black women are incarcerated 3 TIMES MORE than white women. 1 IN 25 will give birth SHACKLED in prison. 75% ARE DOMESTIC VIOLENCE SURVIVORS. 1 IN 10 will be sexually assaulted in prison. 82% are survivors of SEVERE CHILD ABUSE. colorofchange.org ultraviolet. SHARE IF YOU THINK IT'S TIME TO STOP CRIMINALIZING WOMEN AND SURVIVORS. |
| User Query | How many women out of every 4 women are domestic violence survivors? |
| Ground Truth | 3 |
| Phi3.5-V-4B + BigDocs | *3* |
| Phi3.5-V-4B + BigDocs(caption reduction) | *1* |

**Table 24:** *Qualitative test example 2 for the InfoVQA task. When tasked with extracting a percentage based on a textual and pictographic depiction of data, Phi3.5-V-4B + BigDocs(caption reduction) incorrectly picks a percentage value in the vicinity, while Phi3.5-V-4B + BigDocs correctly infers the answer.*

| Component | Content |
|---|---|
| | 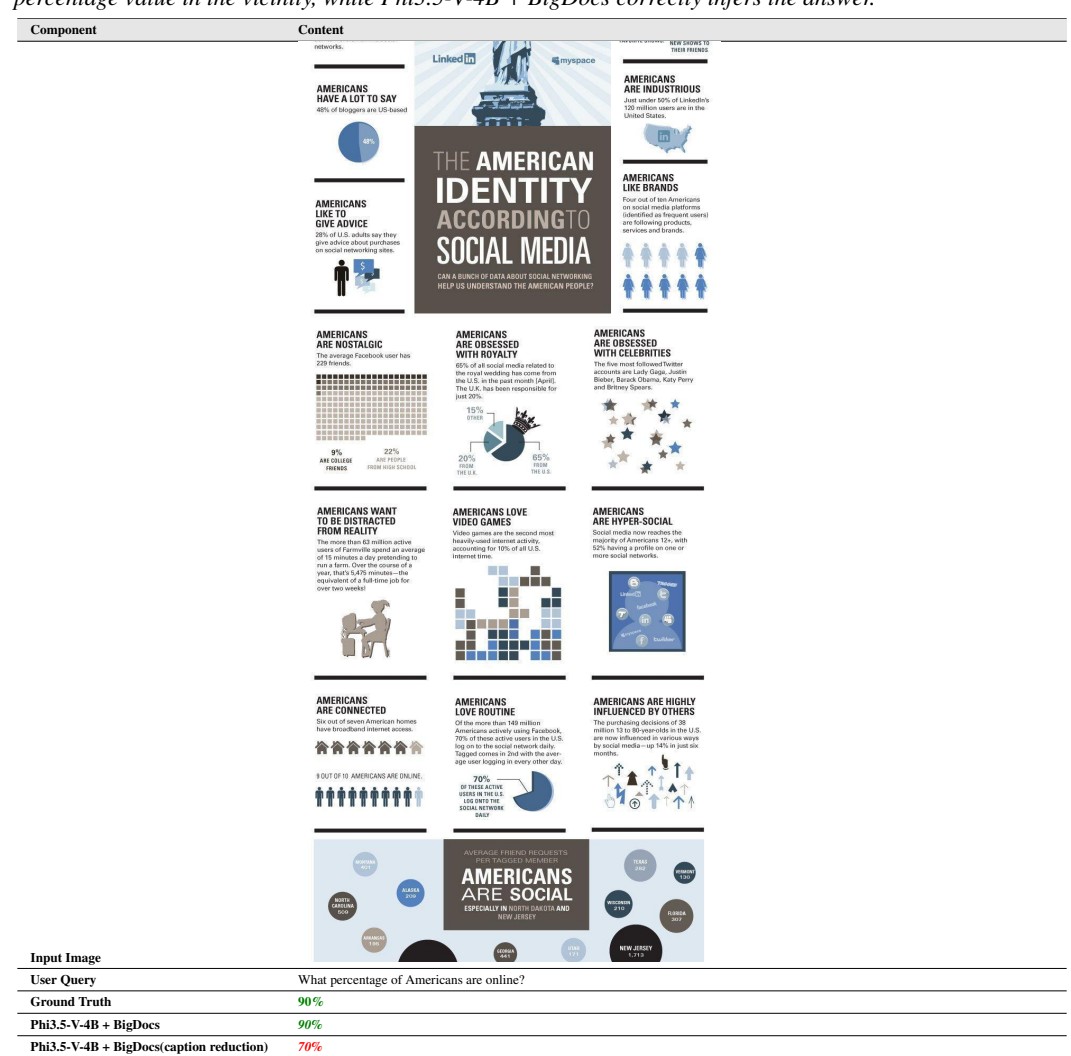 |
| **Input Image** | |
| **User Query** | What percentage of Americans are online? |
| **Ground Truth** | *90%* |
| **Phi3.5-V-4B + BigDocs** | *90%* |
| **Phi3.5-V-4B + BigDocs(caption reduction)** | *70%* |

**Table 25:** *Qualitative test example 3 for the InfoVQA task. Phi3.5-V-4B + BigDocs is able to exploit structure in the image and compute the correct answer, while Phi3.5-V-4B + BigDocs(caption reduction) hallucinates an unrelated answer.*

| Component | Content |
|---|---|
| | 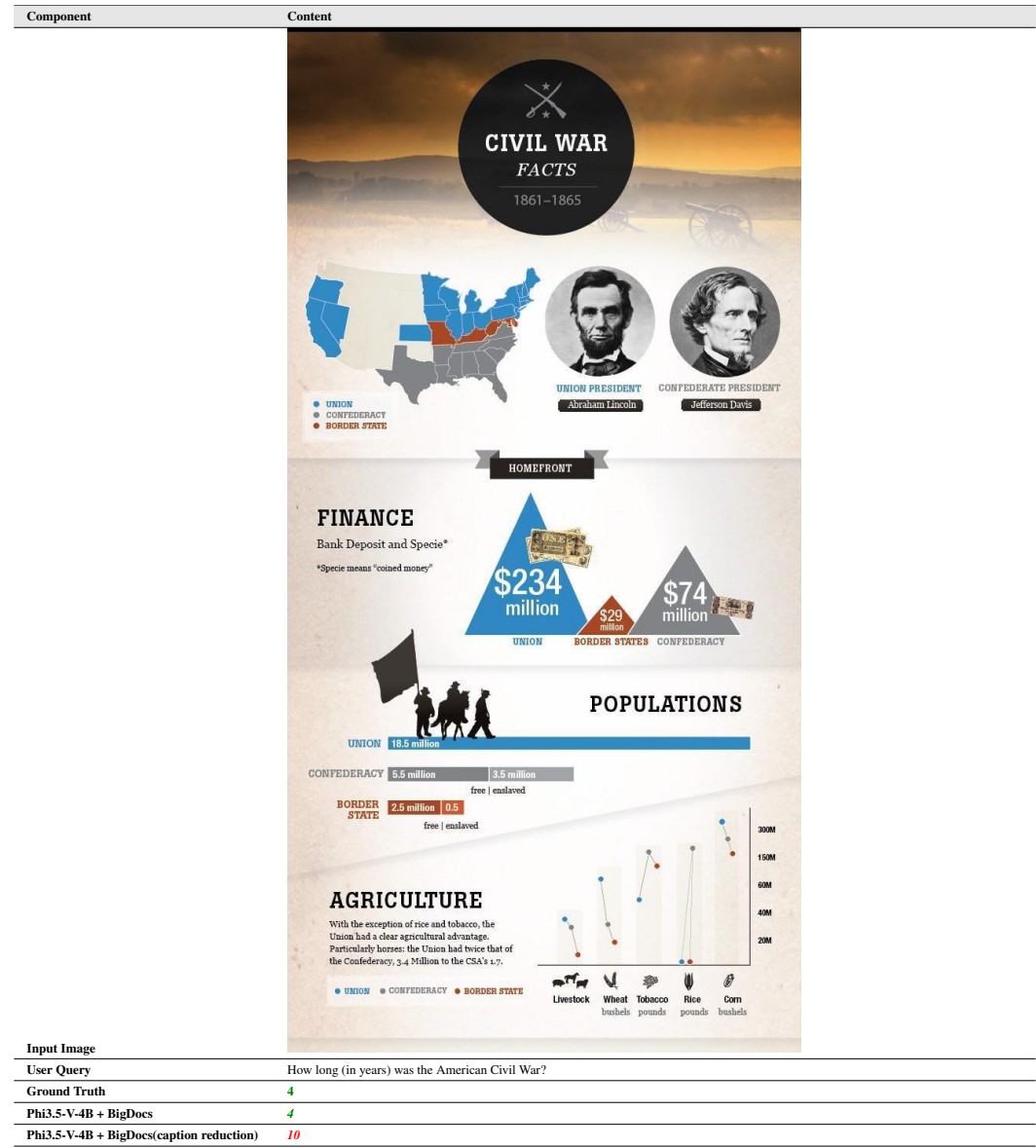 |
| **Input Image** | |
| **User Query** | How long (in years) was the American Civil War? |
| **Ground Truth** | 4 |
| **Phi3.5-V-4B + BigDocs** | 4 |
| **Phi3.5-V-4B + BigDocs(caption reduction)** | 10 |

**Table 26:** *Qualitative test example for the InfoVQA task. Phi3.5-V-4B + BigDocs(no VQA format) does not adhere to the requested format and also provides a wrong answer in contrast to Phi3.5-V-4B + BigDocs which gets both the format and the answer correct. These errors highlight the importance of VQA formatting of the BigDocs dataset.*

| Component | Content |
|---|---|
| Input Image | 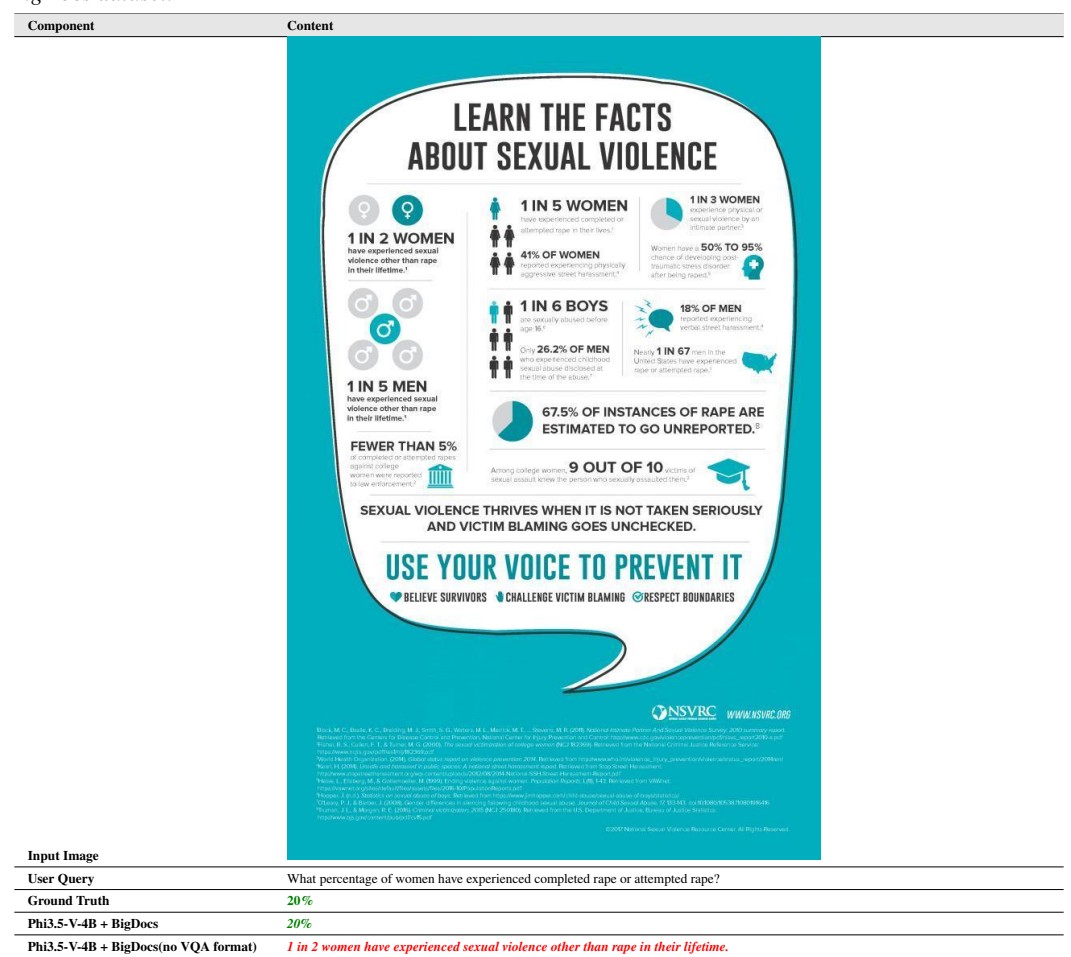 |
| User Query | What percentage of women have experienced completed rape or attempted rape? |
| Ground Truth | *20%* |
| Phi3.5-V-4B + BigDocs | *20%* |
| Phi3.5-V-4B + BigDocs(no VQA format) | *1 in 2 women have experienced sexual violence other than rape in their lifetime.* |

**Table 27:** *Evaluation example from the Image2HTML task. The HTML code is rendered for easier comparison and analysis. The original Phi3.5-Vision performs poorly, while the BigDocs fine-tuned version generates highly accurate results. Although GPT-4 produces correct text, its structure differs significantly from the original image. These results demonstrate that fine-tuning on BigDocs leads to a massive improvement in performance on this task.*

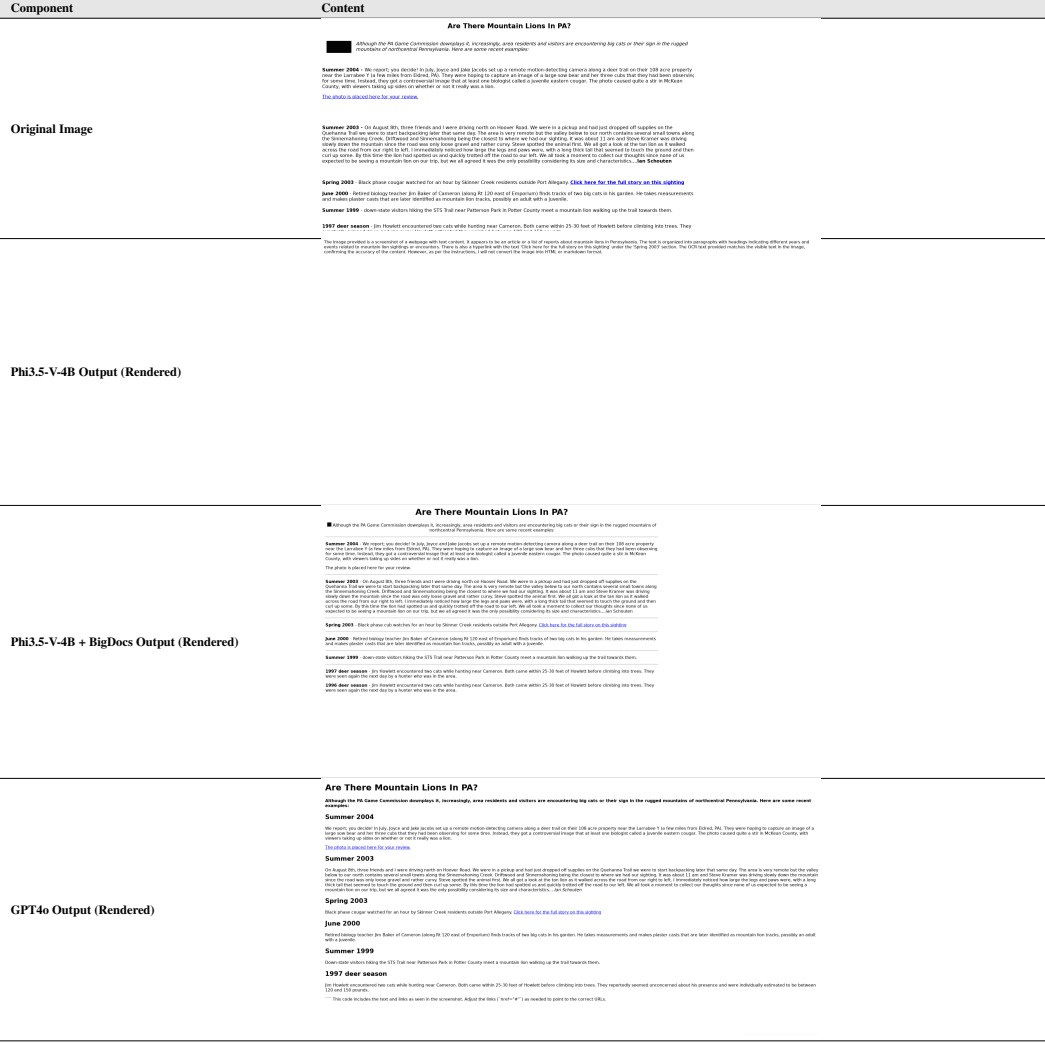

**Table 28:** *Test Example for the task of GUI-VQA, showing BigDocs model failure. Phi3.5-V-4B+BigDocs was not able to give the correct output.*

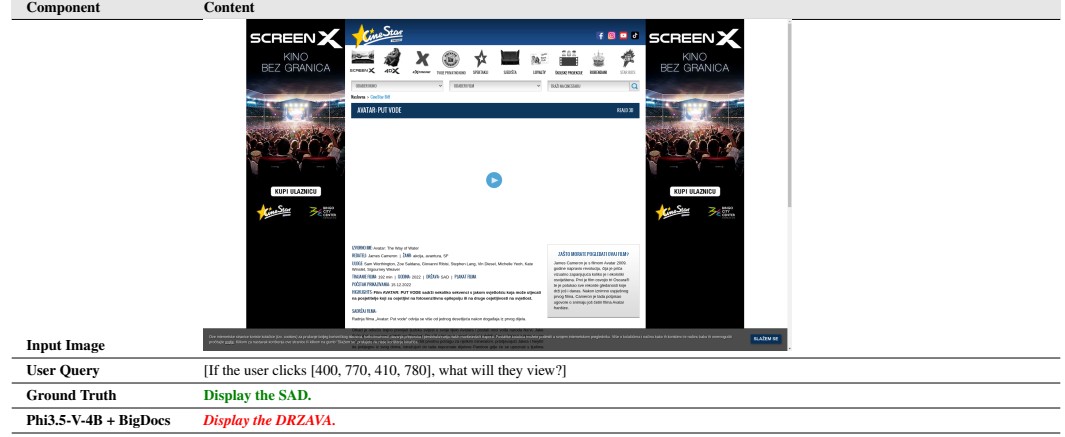

| Component | Content |
| --- | --- |
| Input Image | |
| User Query | [If the user clicks [400, 770, 410, 780], what will they view?] |
| Ground Truth | **Display the SAD.** |
| Phi3.5-V-4B + BigDocs | *Display the DRZAVA.* |

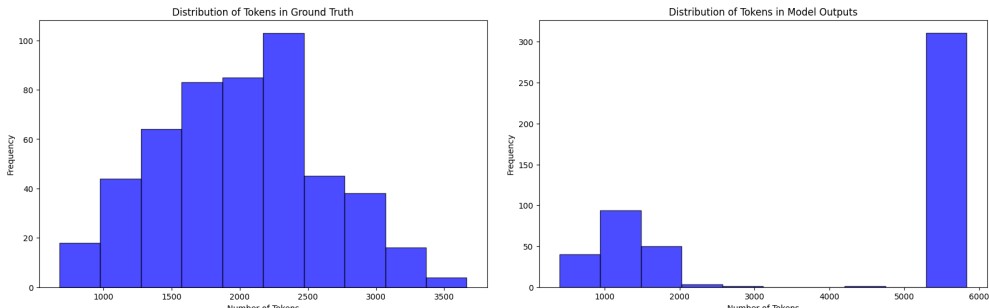

**Figure 18:** *Comparison of token length histograms on the test set of Screenshot2HTML task, and the generated outputs of our model (Phi3.5-Vision+BigDocs)*

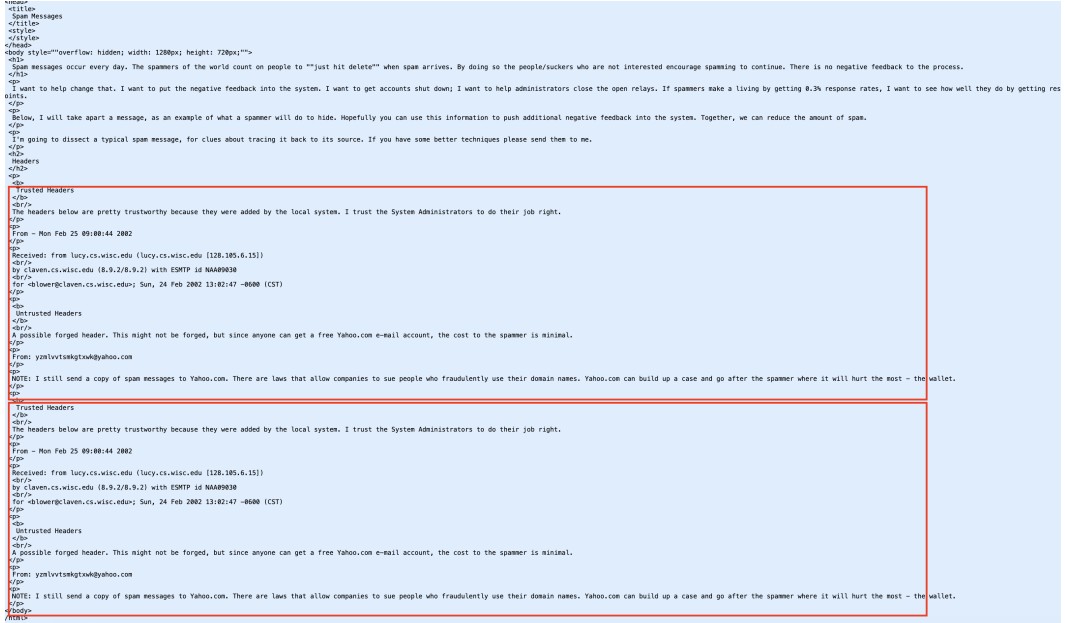

**Figure 19:** *Failure case on the task of Screenshot2HTML. Output is generated by the Phi3.5-Vision + BigDocs model.*

| Context Length | HTML Score | HTML Score |
|:---:|:---:|:---:|
| | Greedy | Length penalty = -1 |
| **512** | 10.99 | 13.96 |
| **1,024** | 11.49 | 15.80 |
| **2,048** | 9.81 | 14.71 |
| **4,096** | 8.83 | 13.88 |
| **8,192** | 8.38 | 13.60 |
| **16,384** | 8.24 | 13.43 |

**Table 29:** *HTML Scores for Phi3.5-Vision+BigDocs with varying context lengths. HTML Score defined as DOM Tree Edit Distance.*

**Figure 20:** *Failure case on the task of Screenshot2HTML. Output is generated by the Phi3.5-Vision + BigDocs model.*

**Figure 21:** *Failure case on the task of Screenshot2HTML. Output is generated by the Phi3.5-Vision + BigDocs model.*

```
<html style=""overflow: hidden; width: 1280px; height: 720px;"">
<head>
<title>
  Tennessee Highway 300
</title>
<style>
</style>
</head>
<body style=""overflow: hidden; width: 1280px; height: 720px;"">
<h1>
  Tennessee Highway 300
</h1>
<p>
  State Primary Route 300 is Tennessee's shortest freeway (and probably the state's shortest highway). Today, it connects I-40 in northwest Memphis with U.S. 51. There are no signs for TN 300; however, the route number is marked on
</p>
<h2>
  Original Plans and a Third Bridge
</h2>
<p>
  The original purpose of TN 300 was as the beginning of a riverfront freeway across Mud Island, to connect to I-40 and Riverside Drive at the Hernando Bridge (the ""New Bridge"").
</p>
<p>
  Another plan was to connect the north loop of then-I-240 to a third bridge over the Mississippi River from Memphis into Arkansas. The Memphis-Arkansas Bridge (the ""Old Bridge""), which serves I-55) is a four-lane bridge that pre
directly feed an I-40 route through central Memphis. Neither bridge was originally designed to be resistant to earthquakes, although the Hernando de Soto Bridge is currently being retrofitted to improve the situation due to concerns
</p>
<p>
  Thus, the plan was to build a new bridge north of downtown. Most truck traffic would then be diverted over to the new bridge. When then-I-240's north was built, the current TN 300 was built as the first step in that route.
</p>
<p>
  It is likely that the third bridge proposal was abandoned due to lack of funding. A new ""third bridge"" proposal has surfaced in 2001, but this proposal diverts I-55 over a bridge via Presidents Island or build a bridge near the
</p>
<p>
  So, for now, TN 300 doesn't do very much. However, there are several unrelated projects that could make use of the route.
</p>
<h2>
  North Second Street
</h2>
<p>
  One plan being considered is the proposed ""North Second Street"" extension. Between Auction Street and the Wolf River, it would be a six-lane surface part of which would continue the one-way couplet of Second and Third streets,
interchange with I-40 in the downtown area.
</p>
<p>
  The main purpose of this route is to get traffic heading for downtown to Memphis' central transit station at the north end of Main Street. It would also divert downtown-bound traffic away from the congested I-40/240 interchange
</p>
<p>
  There is no construction timetable for the route at present. The extension would probably be signed as TN 300, although TDOT might opt to not sign it as such.
</p>
</body>
</html>
```

**Figure 22:** *Success case on the task of Screenshot2HTML. Output is generated by the Phi3.5-Vision + BigDocs model.*

