# OpenReview forum: "BigDocs: An Open Dataset for Training Multimodal Models on Document and Code Tasks"
_ICLR.cc/2025/Conference — ICLR 2025 Poster_

### Official Review · Reviewer_ZUCq · 2024-10-27

**Soundness:** 3
**Presentation:** 3
**Contribution:** 3
**Rating:** 5
**Confidence:** 3

**Summary:**

This paper introduces BigDocs, an open-source large-scale multimodal dataset for document understanding and code generation tasks. The main contributions include:

1. BigDocs-7.5M: A high-quality dataset containing 7.5 million image-text pairs across 30 tasks
2. BigDocs-Bench: 10 new benchmark tasks focusing on structured output generation
3. BigDocs Toolkit: A set of tools for data processing and preparation
4. Experimental validation: Demonstrated effectiveness through comparisons with models like GPT-4

**Strengths:**

1. Addresses a significant issue: licensing restrictions and accessibility problems in existing document understanding datasets

2. Quality assurance:
        Rigorous data filtering process
        Traceable metadata

3. Comprehensive task coverage: from basic document information extraction to complex structured output generation

4. Open-source commitment: supports responsible AI development

**Weaknesses:**

1. Insufficient validation of benchmark quality and reliability

    a). While Section 4.2 mentions manual human verification for BigDocs-Bench, the paper lacks crucial details about the verification methodology and evaluation criteria. For example, the number of human verifiers involved, their qualifications, the specific criteria they used for evaluation, or how inter-rater reliability was ensured.

    b). Given the large volume of synthetic data in the benchmark, the paper fails to address the practical challenges of comprehensive human verification. The sampling strategy and quality assurance process for human verification are not described, raising questions about the robustness of the validation process

2. Limited scope of base model experiments

    a). The experimental validation is confined to models ranging from 2B to 7B parameters. The absence of experiments with larger-scale base models (>7B parameters) limits the understanding of the dataset's effectiveness across different model scales

3. Insufficient qualitative analysis of model performance

    a). The paper lacks detailed error analysis and concrete examples in the Qualitative Results section.

    b). Section 5.3 would benefit from including error distribution patterns across different models and tasks. The absence of systematic error analysis makes it challenging for readers to:

        i. Understand the specific improvements BigDocs-7.5M brings to different document processing tasks

        ii. Identify potential limitations or biases in the dataset

        iii. Comprehend the typical failure modes of models trained on this dataset

**Questions:**

In addition to the three points mentioned in the weaknesses section, I am concerned about the performance reported in Table 3. The unusually low performance of advanced models like GPT-4o and Claude-3.5-sonnet raises questions about the experiment's setup. Did you use one-shot prompting in your experiments? If zero-shot prompting was used, this might create an unfair comparison with models that haven't been fine-tuned on these specific benchmarks. Could you clarify the prompting strategy and consider providing results with one-shot or few-shot prompting for a more equitable comparison?

---

> ### Author Response · Authors · 2024-11-22
>
> We sincerely thank the reviewer for their thorough and constructive feedback that will help strengthen our work. We appreciate the recognition of our key contributions in addressing licensing restrictions, ensuring data quality, providing comprehensive task coverage, and maintaining open-source commitment.
>
> We appreciate the raised concerns and will address and clarify each of them as follows.
>
> >W1: (a, b): Human Verification of Data Creation in BigDocsBench
>
> To address the concerns about verification methodology and evaluation criteria for BigDocs-Bench, we provide the following details:
>
> ### **A. Number of Human Verifiers and Their Qualifications**
>
> We engaged a team of **6 human verifiers**, consisting of graduate-level researchers and professionals with expertise in multimodal datasets and document analysis. All human verifiers were trained on verification criteria mentioned below before beginning the review process. We have at least 2 verifiers on each data sample.
>
> ### **B. Verification Methodology**
>
> **Automatic Filtering Tools**: We implemented multiple automatic filtering tools as the first step in our quality control process. These tools were designed to detect issues such as bad annotations, NSFW content (using Llama-Guard-3 from https://huggingface.co/meta-llama/Llama-Guard-3-11B-Vision), and Personally Identifiable Information (PII). This automated process reduced the volume of problematic samples before any human verification was performed.
>
> **Human Verification for Test Split**:  To ensure a high-quality test split, we only do human verification focused on the test set as our automatic filtering is already quite robust.  Annotators identified and removed problematic samples following the guideline, for example, unnecessary comments in LaTeX and HTML code (e.g., in Table2LaTeX and Screenshot2HTML tasks), which could cause rendering issues. Similarly, flowcharts with excessive isolated nodes in tasks like Image2Flow were flagged and removed. For Image2SVG and three GUI-related datasets, we spotted and removed those images with PII or NSFW content. All these typical errors were collected and documented during this process to inform our automatic filtering tools.
>
> **Refining Automatic Filtering Tools for Training Split**: The typical errors identified during human verification of the test set were used to iteratively improve our automatic filtering tools. This enhancement enabled the tools to better detect and handle similar issues in the training split, which was impractical to verify comprehensively through human labor.
>
> **Sampling-Based Human Verification for Training Split**: To ensure the training split met quality standards, we randomly sampled 100 samples from all tasks, proportional to the size of their respective training splits. These samples were verified by annotators based on the criteria mentioned below, and the overall pass rate was 99% (i.e. 99 good samples), reflecting a high level of quality for the training data after applying the refined filtering tools. Specifically, every task has a pass rate of 100% except GUI-VQA, which has a pass rate of 92% (i.e. 1 bad sample out of 13, which is not relevant to the task since the question is not quite related to the given image).
>
> ### **C. Verification Criteria For Verifiers**
>
> The verification process for BigDocs-Bench prioritized several critical aspects to ensure the dataset's quality, reliability, and suitability for real-world applications.
>
> First, **data integrity** was a primary focus, ensuring accurate alignment between inputs and outputs. For example, in tasks like Image2SVG, visual elements in images were verified to match their corresponding SVG annotations, ensuring precise vector representation. Similarly, for Screenshot2HTML, rendered HTML outputs were checked against the screenshots to confirm structural and semantic fidelity.
>
> Second, the **relevance of synthetic samples** was thoroughly assessed to confirm that they were realistic and reflective of the intended task objectives. For instance, in the Image2Flow task, flowchart samples were evaluated to ensure that the generated JSON or GraphViz code meaningfully represented the logical flow of the diagrams without excessive isolated nodes or disconnected components, which could compromise their utility.
> Finally, to ensure **content safety**,  NSFW content and PII, such as explicit material, or personal data, were removed.
>
> ### **D. Inter-Rater Reliability**
>
> In the test split of our benchmark, we prioritized recall to ensure no potentially problematic samples were missed. For tasks like filtering NSFW content, detecting annotation errors, or identifying corrupted data, any sample flagged as problematic by a single verifier was removed. This conservative recall-focused approach maximized dataset quality and safety while addressing misalignment between raters.
>
> We have added this procedure in Appendix A.13.

---

> > ### Author Response · Authors · 2024-11-22
> >
> > >W2: "The experimental validation is limited to models with 2-7B parameters, which restricts the understanding of the dataset's effectiveness for larger-scale models.
> >
> > We appreciate the reviewer’s point about the limited scope of model experiments in terms of size. To clarify, our **training experiments** primarily focused on state-of-the-art models ranging from 2B to 7B parameters, a decision driven by resource constraints and the scope of our training efforts.
> >
> > As this is primarily a dataset contribution paper, we prioritized demonstrating the dataset's utility within reasonable computational constraints. Training larger models, while valuable, would require substantial resources - **for instance, we estimate that training a 70B Qwen2VL model would take approximately 20 days on our available infrastructure (64 H100 GPUs).**
> >
> > In terms of **validation experiments**, *the original paper did include validation experiments on large models (over 7B parameters), up to 70B*, including GPT-4, Qwen2-VL-70B, and Claude 3.5 Sonet (as shown in BigDocs-Bench results, Table 3). However, these models were not evaluated on the General Document Benchmarks (Table 2). We have now evaluated these larger models across all benchmarks in the paper and included all scores in Table 8. Additionally, we have added a new baseline, Llama3.2-Vision-90B, to explore models up to 90B. Table 8 can also be found here: https://imgur.com/a/ICy9C25 and in the edited manuscript.
> >
> > With these updated results, **we observe that GPT-4 outperforms all models on the General Document Benchmarks**, achieving an aggregated score of **64.62%**, surpassing our best model, Phi3.5-Vision + BigDocs, which reaches **60.80%**. This is impressive considering the size and training infrastructure differences, while BigDocs ensures full transparency. Notably, *the largest open model, Llama3.2-Vision-90B, underperforms smaller models* with a low aggregated score or **38.38**, particularly due to its poor performance in DeepForm and WTQ, even with the most optimized prompts.

---

> > > ### Author Response · Authors · 2024-11-22
> > >
> > > >W3 (a, b): Insufficient qualitative analysis of model performance
> > >
> > > We appreciate the reviewer’s insightful feedback on the lack of error analysis and concrete examples in our paper. We acknowledge that the current qualitative results, as shown in Figure 8 (with diverse task outputs like Chart2Caption, Image2Latex, Image2SVG, Flow2JSON) and Table 6 (comparing different models on tasks like GUI VQA, Chart Captioning, Latex Generation, and HTML Generation), do not yet fully capture error distributions or failure modes across models and tasks. This comparison, while informative, still falls short of providing a deeper understanding of error patterns and model-specific limitations.
> > >
> > > To address this, **we have conducted a more comprehensive and systematic analysis of qualitative examples, which has led to the inclusion of Tables [9-28]** (now added to the manuscript). To perform this evaluation, we randomly selected 10 test samples for each task in SVG-Bench and inspected the outputs of the 14 models present in the main paper. We selected samples that highlight improvements obtained with BigDocs, limitations, typical failures, and possible biases of models. Results and discussions are included in Appendix A.10 (Qualitative Results).
> > >
> > > **The following are notable observations indicating improvements with BigDocs.**
> > >
> > > 1. BigDocs models perform well at tasks requiring localization in GUI as evidenced by the GUI-VQA and GUI2Intent examples in Tables [11-14] (https://imgur.com/a/HPIXiIw, https://imgur.com/a/EjvLtZ8) while the other models provide incorrect answers. Qwen2-VL-2B gets the correct answer in some of the cases but tends to be very verbose.
> > >
> > > 2. BigDocs models do well at capturing spatial orientation and stylistic elements even in webpages with a lot of elements like the GUI2Sum example illustrated in Table 15 (https://imgur.com/a/8T0XgWA). Baseline models like Qwen2-VL-2B, Llava-NeXT-7B, and DocOwl-1.5-8B make mistakes with spatial orientation, understanding the layout, or providing simplistic captions.
> > >
> > > 3. Tables [17-20] (https://imgur.com/a/YXuBWld) show that BigDocs models accurately preserve the format of the table, column and row lines, and style of texts, for the task of Table2Latex, beating both baseline models and closed ones like GPT-4o.
> > >
> > > **Here are some failure cases**
> > >
> > > 1. Complex analysis and reasoning over charts is still a challenge to both BigDocs models and other open-source models like Qwen2-VL-2B, Llava-NeXT-7B, and DocOwl-1.5-8B as evidenced in Table 9 (https://imgur.com/a/T6kyt5d) and Table 10 (https://imgur.com/a/TBu1TkX). *BigDocs models provide more concise captions while still making some factual mistakes.**
> > >
> > > 2. When interacting with web pages containing a high density of elements, BigDocs models can sometimes make mistakes with unique-sounding names like the example in Table 15 (https://imgur.com/a/8T0XgWA).
> > >
> > > These updates, along with the accompanying discussion added to the manuscript, now offer a clearer picture of the strengths, limitations, and failure modes of the models trained on BigDocs-7.5M.
> > >
> > > Once again, we thank the reviewer for providing this suggestion, as it has greatly improved the quality of our paper.

---

> > > > ### Author Response · Authors · 2024-11-22
> > > >
> > > > >Q1: "The unusually low performance of advanced models like GPT-4o and Claude-3.5-sonnet raises questions about the experiment's setup"
> > > >
> > > > We appreciate the reviewer’s observation regarding the performance issues of some models in Table 3, particularly GPT-4 and Claude-3.5-Sonnet. Notably, this concern was also raised by Reviewer 1wn6.
> > > >
> > > > **Explanation of Initial Results**
> > > >
> > > > The performance discrepancies noted by the reviewer stem from differences in model training, instruction tuning datasets, and prompt templates. Each model requires specific prompt engineering to produce optimal outputs, as their responses vary widely in style and structure. For instance, some models adopt a conversational tone, while others are more structured, making uniform evaluation challenging.
> > > >
> > > > In our original submission, we applied a single, consistent prompt across all closed models for fairness. However, we acknowledge that this approach limited the ability of certain models to fully showcase their strengths, particularly on complex tasks like Chart2MD and GUI2Sum. Parsing the outputs was also challenging due to the heterogeneity in formatting, and this likely impacted the results for some tasks.
> > > >
> > > > **Improvements Made**
> > > >
> > > > We conducted a thorough review and optimized prompts for each model to better align with their strengths and output styles. We found that some models needed prompts like "answer briefly" or "answer in one sentence or phrase" to avoid long, chatty outputs, especially for tasks like Chart2MD, which involves markdown generation.
> > > >
> > > > Following the reviewer’s suggestion, we implemented one-shot prompting techniques, which proved especially effective for the Image2Flow task in both its JSON and Graphviz versions. Previously, models struggled with these formats, often scoring zero due to unfamiliarity with the exact syntax. By including a single example extracted from the training set, we significantly improved model alignment and performance.
> > > >
> > > > We have re-computed all benchmark scores using this updated prompting framework, updated Table 3 in the paper, and provided the revised table with highlighted edits here: https://imgur.com/a/yKT7VAB.
> > > >
> > > > We are pleased to report that these optimizations have led to significant performance improvements. For example, Claude3.5-Sonet achieved a remarkable **54%** boost on Chart2MD, while GUI2Intent performance for Claude2.5-Sonet increased substantially from **4.87%** to **13.12%**.
> > > >
> > > > The updated results reveal a more consistent and expected trend: state-of-the-art closed models outperform open-source models by approximately **30%**, while BigDocs models continue to lead with a strong margin of **40-50%**, achieving a peak performance of **56.32%** with Phi3.5-Vision.
> > > >
> > > > We appreciate the reviewer’s suggestion to revisit the scores in Table 3 and their recommendation to incorporate one-shot prompting, which significantly enhanced the competitiveness of our benchmark.

---

> > > > > ### Author Response · Authors · 2024-11-25
> > > > >
> > > > > Dear Reviewer,
> > > > >
> > > > > Thank you again for your insightful review. It has significantly improved our paper, particularly with **1)** the detailed clarification of our human verification process for synthetic data, **2)** the inclusion of validation experiments on larger models, **3)** the comprehensive analysis of qualitative results you proposed, and **4)** the substantial effort spent optimizing prompts and one-shot settings to improve our leaderboard performance.
> > > > >
> > > > > We would like to confirm that all your comments have been fully addressed.
> > > > >
> > > > > Thanks!

---

> ### Author Response · Authors · 2024-11-29
>
> Dear Reviewer,
>
> Thank you for dedicating your time and effort to reviewing our manuscript and for providing insightful suggestions. As the author-reviewer discussion phase nears its conclusion, we would like to ensure that our responses have adequately addressed your concerns. Please feel free to reach out if you have any additional questions or require further clarification.
>
> Best regards

---

> > ### Comment · Reviewer_ZUCq · 2024-12-01
> >
> > Thank you for your thoughtful and detailed response to my concerns. I appreciate the clarifications provided. The authors have adequately addressed my questions about the methodology and experimental setup.
> >
> > After careful consideration of the rebuttal, I believe the technical soundness of the paper should be rated as 'good'. The authors have demonstrated the reliability of their approach and provided sufficient evidence to support their claims. However, I will maintain my original overall assessment, as the broader impact and significance of the contribution remain at the same level as initially evaluated.

---

> > > ### Author Response · Authors · 2024-12-01
> > >
> > > Dear Reviewer ZUCq:
> > >
> > > Thank you for your response, and for acknowledging the clarifications we provided regarding the methodology and experimental setup. We appreciate your updated assessment of the technical soundness as 'good' and your recognition of the reliability of our approach.
> > >
> > > We noticed that you maintained your original overall assessment based on your evaluation of the broader impact and significance of the work. We would be grateful if you could provide more specific details about your concerns regarding these aspects, as they were not explicitly outlined in the original review. This would allow us to better understand your perspective and provide targeted clarifications to address any remaining uncertainties.
> > >
> > > Once again, thank you for your time and effort in reviewing our work. Your feedback has been invaluable in improving our manuscript.
> > >
> > > Best Regards

---

> > > > ### Author Response · Authors · 2024-12-02
> > > >
> > > > Dear Reviewer,
> > > >
> > > > We have fully addressed your feedback, adding 25 pages of updates. **Based on your feedback, this effort has *at least* elevated the paper from “marginally below acceptance” (5) to deserving a higher score (at least a 6, marginally above acceptance)**. We kindly request you to revise your score accordingly.
> > > >
> > > > Best regards,
> > > >
> > > > The BigDocs Team

---

### Official Review · Reviewer_1wn6 · 2024-11-04

**Soundness:** 4
**Presentation:** 4
**Contribution:** 3
**Rating:** 8
**Confidence:** 4

**Summary:**

The paper introduces a new large-scale and license-permissive dataset BigDocs for continual pertaining on multimodal document and code tasks. Based on this dataset, the paper also introduces BigDocs-Bench containing the training set, validation set and test set to evaluate tasks like code generation and reasoning from documents.

**Strengths:**

+ The paper introduces a new large-scale dataset for pertaining and fine-tuning, and a corresponding benchmark, with permissive license, which is the first one in the target domain. This is a concrete contribution to the research community and industry in the subarea.
+ The paper presents comprehensive experiments with leading open and closed models on both general and proposed benchmarks. The results are promising. Also, human evaluations provide further evidence on the effectiveness of training on BigDocs.

**Weaknesses:**

+ The results on the proposed BigDocs-Bench are a bit strange. The performances of the same model on different tasks are not stable, e.g., Claude-3.5-sonnet behaves very bad on Chart2MD and gets a high score on GUI2Sum. Further, small models can gets higher scores than a much larger model, e.g., Qwen2-VL-2B gets a higher avg. score(20.00) than Claude-3.5-Sonnet(18.31) and GeminiPro-1.5(19.23). This could indicate the benchmark is too specific for a general model not trained with BigDocs.
+ In human evaluation, gpt-4 gets a higher win rate than Phi3.5-BigDocs, while in BigDocs-Bench it's the opposite, which again indicates the benchmark may not differentiate a stronger model.

**Questions:**

+ 'BigDocs will be open-sourced (upon acceptance)'. If you want to open-source this dataset, what's the point of waiting upon acceptance?
+ Personally, I think it's not necessary to keep a *hidden test set* for the benchmark.

---

> ### Author Response · Authors · 2024-11-22
>
> We appreciate the reviewer’s positive assessment of our dataset and benchmark contributions, the comprehensive experiments, and the inclusion of human evaluations. We are glad that the permissive licensing, breadth of experiments, and the overall utility of BigDocs-7.5M and BigDocs-Bench were seen as valuable contributions.
>
> We respond below to address the concerns raised in the review.
>
> >W1: GPT-4o and Claude-3.5-sonnet underperform on BigDocs-Bench (Table 3)
>
> We appreciate the reviewer’s observation about the variability in model performance across tasks and acknowledge that this highlighted an area where our original methodology could be refined.
>
> **Explanation of Initial Results**
>
> The performance discrepancies noted by the reviewer stem from differences in model training, instruction tuning datasets, and prompt templates. Each model requires specific prompt engineering to produce optimal outputs, as their responses vary widely in style and structure. For instance, some models adopt a conversational tone, while others are more structured, making uniform evaluation challenging.
>
> In our original submission, we applied a single, consistent prompt across all closed models for fairness. However, we acknowledge that this approach limited the ability of certain models to fully showcase their strengths, particularly on complex tasks like Chart2MD and GUI2Sum. Parsing the outputs was also challenging due to the heterogeneity in formatting, and this likely impacted the results for some tasks.
>
> **Improvements Made**
>
> We conducted a thorough review and optimized prompts for each model to better align with their strengths and output styles. We found that some models needed prompts like "answer briefly" or "answer in one sentence or phrase" to avoid long, chatty outputs, especially for tasks like Chart2MD, which involves markdown generation. For tasks like Image2Flow, we found it crucial to use 1-shot prompting by providing the format of a training example. This helped models understand formats like JSON or Graphviz, which were specified in the prompt but not always generated correctly by most models.
>
> We have re-computed all benchmark scores with the new prompting framework, and have updated the corresponding Table 3 in the paper. **We provide here an image of the new table, with the edits highlighlted https://imgur.com/a/yKT7VAB**.
>
> We are pleased to report that **this optimization has led to a significant performance boost**. For Chart2MD, Claude3.5-Sonet saw an improvement of **54%**, while GUI2Intent improved considerably, from **4.87% to 13.12%** with Claude3.5-Sonet.
>
> *The updated results now show a more consistent and expected pattern*, with state-of-the-art closed models outperforming open-source models by about **30%**. *BigDocs models continue to lead with a strong advantage of 40-50%, reaching 56.32% in the best case with Phi3.5-Vision.*
>
> We appreciate the reviewer’s feedback, which prompted us to revisit and improve our evaluation methodology. We have updated the paper's tables and discussion to reflect the revised benchmarks, ensuring a more accurate representation of model performance.

---

> > ### Author Response · Authors · 2024-11-22
> >
> > >W2: "In human evaluation, GPT-4 has a higher win rate than Phi3.5-BigDocs, but Phi3.5-BigDocs outperforms GPT-4 on BigDocs-Bench, which is inconsistent."
> >
> > **Looking at the graph (https://imgur.com/a/02T3NA3)**, it’s clear that while Phi3.5-BigDocs slightly outperforms GPT-4 in quantitative scores on Screenshot2HTML **(12.05 vs. 10.33)**, the human evaluation reveals a tight competition between the two models, with GPT-4 edging out as the winner. The notable "neither" and "draw" segments suggest that human evaluators often struggled to decisively favor one model, which reflects the complexity of HTML generation—tasked with handling extensive context lengths (as noted in Table 1) and varied structures.
> >
> > While we acknowledge the reviewer’s observation, we believe that the competition is close both quantitative and qualitative, and the subjective nature of human evaluation means this should not be considered a significant issue.

---

> > > ### Author Response · Authors · 2024-11-22
> > >
> > > >Q1: "If you want to open-source, what's the point of waiting upon acceptance?"
> > >
> > > The statement about our plans to open-source BigDocs was included in the anonymous version of the paper to maintain the double-blind review process. The dataset and other artifacts will be released through common channels once all necessary mechanisms, such as an opt-out option, are in place. We will remove this statement from the paper.
> > >
> > > >Q2: "It is not necessary to keep a hidden test"
> > >
> > > We respectfully disagree with this point. Maintaining a hidden test set is widely recognized as a best practice in benchmark design and has been a longstanding tradition in the multimodal research community [1, 2, 3, 4, 5]. This approach safeguards the integrity of evaluations by preventing overfitting to the test data, ensuring that reported results reflect genuine generalization to unseen examples. Additionally, it minimizes the risk of data contamination and fosters fair comparisons across different models.
> > >
> > > While some recent benchmarks have overlooked this practice, we believe it is a valuable addition to our BigDocs-Bench. We will provide a community leaderboard, and allow researchers to submit their models for evaluation and compute results on unseen data. This approach will provide better insights into the models' generalization capabilities on truly unseen data.
> > >
> > > [1] Antol, S., Agrawal, A., Lu, J., Mitchell, M., Batra, D., Zitnick, C. L., & Parikh, D. (2015). Vqa: Visual question answering. In Proceedings of the IEEE international conference on computer vision (pp. 2425-2433).
> > >
> > > [2] Lin, T. Y., Maire, M., Belongie, S., Hays, J., Perona, P., Ramanan, D., ... & Zitnick, C. L. (2014). Microsoft coco: Common objects in context. In Computer Vision–ECCV 2014: 13th European Conference, Zurich, Switzerland, September 6-12, 2014, Proceedings, Part V 13 (pp. 740-755). Springer International Publishing.
> > >
> > > [3] Dinan, E., Logacheva, V., Malykh, V., Miller, A., Shuster, K., Urbanek, J., ... & Weston, J. (2020). The second conversational intelligence challenge (convai2). In The NeurIPS'18 Competition: From Machine Learning to Intelligent Conversations (pp. 187-208). Springer International Publishing.

---

> > > > ### Comment · Reviewer_1wn6 · 2024-11-23
> > > > **Thanks for your rebuttal**
> > > >
> > > > Thanks for the reply. I believe this is good work considering the contribution. I will raise my score accordingly.
> > > >
> > > > I have a few suggestions:
> > > >
> > > > 1) Based on the updated results and my own experience, complex tasks involving structured data should have different optimized prompts for different series of models. This is important to make the benchmark reliable. As such, it would be good to let the community try and share their own prompts.
> > > >
> > > > 2) While a hidden test set effectively prevents data contamination, it might slow down updates to the leaderboard. Balancing these trade-offs is challenging, and I hope the authors can address this effectively.

---

> > > > > ### Author Response · Authors · 2024-11-25
> > > > >
> > > > > Thank you for your comments and thoughtful suggestions. We acknowledge the importance of allowing the community to optimize prompts for better performance on structured data tasks and will prioritize this feature. We also appreciate your feedback on maintaining an up-to-date leaderboard and will work to address this effectively.

---

### Official Review · Reviewer_UxFn · 2024-11-10

**Soundness:** 3
**Presentation:** 4
**Contribution:** 3
**Rating:** 5
**Confidence:** 3

**Summary:**

The paper introduces BigDocs-7.5M, a large-scale, open-access dataset (CC-BY-4.0) designed to enhance multimodal model training on document and code-related tasks. Addressing limitations in existing datasets such as restrictive licensing and data access, BigDocs-7.5M provides a rich collection of 7.5 million multimodal documents suitable for a variety of tasks including document reasoning, structured output generation, and graphical interface interpretation. The authors alrso release the BigDocs-Bench to evaluate LLMs ability to analyze code and docs over 10 categories of tasks.

**Strengths:**

1. Comprehensive and Novel Dataset: The introduction of BigDocs-7.5M offers a permissively licensed, open-source dataset that includes a wide variety of document types and structured outputs

2.  Improvement in Model Performance: The paper convincingly demonstrates that models trained on BigDocs-7.5M outperform those trained on existing closed-source datasets, particularly in multimodal document understanding and code generation tasks​ with Phi-3.5 Finetuned at ~50% vis-a-vis gpt-4o ~25%.

3. Thorough evaluation suite: The introduction of BigDocs-Bench for benchmarking model performance across 10 novel tasks is a valuable contribution, providing detailed insights into the models' capabilities.

**Weaknesses:**

1.With regards to human-evaluation in section 5.3 of the paper, could the authors shed more light on the evaluators' qualifications, selection process, and any potential conflicts of interest?

2. While I do appreciate the value of BigDocs-Bench, and the evaluation on a collection of open-source and closed-source models, I would encourage the authors to consider including a correlation analysis between BigDocs-Bench and specific benchmarks like human-eval and RULER. This would provide a clearer picture of how BigDocs-Bench relates to existing evaluation metrics in the field.

**Questions:**

While I appreciate the authors pro-actively acknowledging that the limited context length of the models trained (8192 tokens) might impact the performance of the models, can the authors provide some insights on the artifacts of this decision. For example, we can see both the 7b and the phi-3.5 models saturating around 50%. Do the authors suppose this might be an artifact of the context length? On the same lines, or perhaps, is there another reason for the plateau? Or is it merely the dataset getting harder beyond half the samples?

**Details Of Ethics Concerns:**

In the appendix of the paper, the authors mention "Note that
datasets with * are those fully or partially curated by us " and then go on to cite the work as such "CDIP-1M* (Soboro, 2022): [Task included: (1).." which makes me wonder if this is in violation of the double-blind policy? In the spirit of erring on the side of caution, I'm flagging this.

---

> ### Author Response · Authors · 2024-11-22
>
> We appreciate the thoughtful review and the recognition of the contributions made by BigDocs-7.5M and BigDocs-Bench. We are pleased that the reviewer acknowledged BigDocs-7.5M's novelty, openness, and performance benefits, as well as the robustness of BigDocs-Bench for diverse multimodal evaluations.
>
> We also appreciate the reviewers' concerns and are glad to address them.
>
> >W1: "More clarity on human evaluators"
>
> To ensure fairness and minimize bias, we invited evaluators from diverse backgrounds, including PhD researchers, ML practitioners, multimodal AI experts, and individuals from both technical and non-technical domains. Key authors were excluded from participating to avoid conflicts of interest. For the evaluation, we used our own web interface, to make sure that all participants were anonymous and model outputs were randomized to prevent pattern recognition. We are open to releasing the detailed human evaluation results, as a supplementary document.

---

> ### Author Response · Authors · 2024-11-22
>
> >W2: Correlation analysis between BigDocs-Bench and other benchmarks
>
> We appreciate the suggestion of a correlation analysis to highlight BigDocs-Bench's uniqueness in the VLM space and thank the reviewer for recommending benchmarks such as RULER [1] and "human-eval" (likely referring to HumanEval [2]). These datasets, however, might not be suitable for correlation analysis in our vision-language model (VLM) setup, mainly because they are text-only benchmarks. RULER [1] is an LLM benchmark designed to measure long-context performance, while HumanEval [2] is a benchmark to evaluate models on code generation tasks. Both are text-only benchmarks.
>
> While the proposed benchmarks may not directly apply to our multimodal setting, we acknowledge the importance of such correlation analysis. We thus performed a study on the correlation of BigDocs-Bench and other popular general VLM benchmarks. We present the results in the form of a correlation matrix, a dendrogram plot, and a PCA analysis. We facilitate the figures here (https://imgur.com/a/SQHV3lm), and we also added them to the manuscript (Figures 16 and 17)
>
> Please see our response to reviewer gUaq (W2), who also requested a correlation analysis focused on the document benchmarks presented in the original paper. This might be of interest to the reviewer for a more detailed view of document-specific benchmarks.
>
> **Analysis Setup.** We selected all benchmarks listed in Tables 2 and 3, covering both General Document Benchmarks and our proposed BigDocs-Bench, and expanded the analysis to include 28 additional VLM benchmarks (see figure for the complete list). The matrix was constructed using all models presented in the paper, including instruction-tuned versions of DocOwl-1.5-8B, Qwen2-VL-2B, LLaVA-NeXT-7B, Phi3.5-v-4B, Qwen2-VL-72B, Llama-3.2-90B, GPT-4o, Claude 3.5 Sonet, Qwen2-VL-72B, and GeminiPro-1.5. Notably, we added Llama-3.2-90B to our baselines during the rebuttal period.
>
> We evaluated as many models and benchmarks as resources allowed to construct a complete matrix of benchmarks and datasets. Missing scores were imputed using the mean to ensure a fully populated matrix for the analysis.
>
> **Methods.**
> 1. **Cosine similarity matrix and dendrograms:** We normalize metric scores by centering (subtracting the mean) and scaling (dividing by the Euclidean norm). We then apply hierarchical clustering using cosine distance to group benchmarks and visualize their relationships. Cosine similarity matrices and dendrograms illustrate these clusters.
> 2. **Principal Components Analysis (PCA):** We represent each benchmark as a feature array composed of scores from all models and project these arrays onto the two principal dimensions.
>
> **Results**
>
> Figure 16  (https://imgur.com/a/SQHV3lm, top) presents the correlation matrix and dendrograms. Circles represent BigDocs-Bench tasks (our benchmarks), upward triangles denote other document benchmarks included in the paper, and downward triangles denote general VLM benchmarks. BigDocs-Bench tasks demonstrate distinctive evaluation characteristics compared to existing benchmarks in the VLM space. This is evidenced by the clear clustering pattern, where BigDocs-Bench tasks form their own correlated group while showing minimal correlation with other benchmark types. Results also reveal three main clusters of tasks: general vision-language tasks (like COCO_VAL and VCR), document understanding tasks, and GUI/table-related tasks (such as our tasks GUI2Sum and Table2Latex). This clustering pattern suggests that BigDocs-Bench is addressing unique aspects of document understanding that is not captured by existing benchmarks.
>
> Figure 17 (bottom) illustrates the PCA results, revealing that most BigDocs-Bench tasks are clustered in the bottom-right corner, distinctly separated from other benchmarks. Notably, tasks within BigDocs-Bench form clear clusters based on their type, such as VQA, Flows, Charts, or GUIs. General VLM benchmarks create a dense cluster, while document benchmarks are more sparse.
>
> **Unique Aspects of BigDocs-Bench.** Our correlation results show a clear separation between BigDocs-Bench and the larger VLM benchmark landscape. In summary, it introduces the following novel elements to the landscapes of VLM evaluation and document understanding:
>
> **1. Document and Chart Reasoning:** Tasks focused on understanding, question answering, and summarization of charts, graphs, and workflows from images.
>
> **2. Web and GUI Understanding:** Tasks targeting web and graphical user interface comprehension, a less explored area in current benchmarks.
>
> **3. Multimodal Code Generation:** Tasks involving the conversion of images into code representations, such as inverse rendering and code generation (e.g., HTML, SVG, JSON).
>
> **4. Long-Context Structured Output Generation:** Tasks requiring structured output over extended contexts, such as generating long JSON, SVG, HTML or Markdown from images

---

> ### Author Response · Authors · 2024-11-22
>
> References:
>
> [1] Hsieh, Cheng-Ping, et al. "RULER: What's the Real Context Size of Your Long-Context Language Models?." arXiv preprint arXiv:2404.06654 (2024).
>
> [2] Chen, Mark, et al. "Evaluating large language models trained on code." arXiv preprint arXiv:2107.03374 (2021).

---

> ### Author Response · Authors · 2024-11-22
>
> >Q1: "The context length of the models (8192 tokens) might impact performance"
>
> This is a valuable observation, particularly given that the *Screenshot2HTML task, which averages around 32k tokens per example*(as shown in Table 1), faces significant challenges under an 8k-token limit. This task ranks among those with the lowest scores (as seen in Table 3), contributing to a decrease in the overall aggregated average score. As such, we believe that **the Screenshot2SVG task is could be affected by saturation due to the limited context length*. Therefore, we focus our answer on this task.
>
> Despite this limitation, we observed that the generated HTML code remains compilable, and the resulting DOM can still be evaluated using the DOM Tree Edit Distance metric, even though the HTML output is truncated due to the context limit. However, the inability to handle complete HTML outputs underscores the need for models capable of processing extended context lengths.
>
> To better understand this issue, we analyzed the model outputs (HTML code) in terms of a histogram of token lengths, comparing them with the ground truth lengths in the test set (both images are available here: https://imgur.com/a/9oLDAlA). It’s important to note that the test set is designed so that the images can be represented in fewer than 8192 tokens (top figure).
> **This analysis confirms that the task is indeed suffering from token length constraints,** as most samples reach the maximum token length of 6k (after subtracting image tokens), as seen in the bottom figure (https://imgur.com/a/9oLDAlA). Although the test set is intended to be generated within a reasonable token length, the model attempts to generate longer HTML outputs due to its training on larger contexts.
>
> **We are conducting an ablation experiment on the context length** by fine-tuning the Phi-3.5-Vision model with an extended context length of 16k. This training is still ongoing, and we will share the results once available.
>
> There are several key reasons for our decision to fix the context length at 8192 tokens:
>
> **1. Model Compatibility:** Not all models evaluated in our benchmark natively support larger context lengths, and extending this limit could disadvantage some models.
>
> **2. Compute Efficiency:** Increasing context lengths substantially boosts GPU memory usage, making both training and inference at scale more challenging.
>
> **3. Comparability:** To ensure fair comparisons across all models, we standardized the context length at 8k tokens.
>
> In conclusion, the reviewer’s observation effectively identified the issue with the token length constraint caused by the 8k token limit in the Screenshot2HTML task. Training a model with a larger context and re-evaluating may yield improvements for this task. Nevertheless, this highlights the importance of long-context tasks, such as HTML generation, as challenging benchmarks that drive innovation.

---

> > ### Author Response · Authors · 2024-11-22
> >
> > >Ethics concern: Problem with CDIP-1M, potentially breaking the double-blind policy
> >
> > We appreciate the reviewer’s diligence in flagging potential concerns about the double-blind policy. However, this is not a violation of the policy. The datasets marked with an asterisk (*) refer to cases where we performed substantial filtering, curation, and reannotation to adapt the original datasets for our work. Specifically, in the case of "CDIP-1M," we cite the original IIT CDIP [3] dumps, as they are the original source. However, the dataset included in BigDocs is derivative work created through our extensive contributions, as outlined in the appendix.
> >
> > [3] Ian Soboro. Complex document information processing (cdip) dataset. https://doi.org/10.18434/mds2-2531, 2022. Accessed: 2024-06-20.

---

> > > ### Author Response · Authors · 2024-11-25
> > >
> > > Dear Reviewer,
> > >
> > > Thank you again for your insightful feedback. It has significantly improved our paper, particularly with the proposed correlation analysis of our benchmark with other VLM benchmarks, the context length saturation issue, and the clarifications on human evaluation.
> > >
> > > We would like to confirm that all your comments have been fully addressed.
> > >
> > > Thanks!

---

> > > > ### Comment · Reviewer_UxFn · 2024-11-27
> > > >
> > > > 1. Thank you for the response. I appreciate the correlation analysis, and that has fully addressed my concern there.
> > > > 2. Thank you for clarifying how the upstream dataset was used. This has also fully addressed my concern here, however it might be valuable to change the wording in the paper to alleviate the confusion.
> > > >
> > > > 3. Regarding human annotators, the authors mention "Key authors were excluded from participating to avoid conflicts of interest". Am I fair to conclude then, that only "key" authors of the paper did not participate, but "non key" authors of the paper did participate?
> > > >
> > > > 3. Long context - thank you for acknowledging, and I await the results from the 16K run (if it were to finish in time hopefully).

---

> > > > > ### Author Response · Authors · 2024-11-28
> > > > >
> > > > > Thanks for your response! We are glad that we were able to successfully address your concerns regarding **1)** the correlation analysis and **2)** the upstream datasets.
> > > > >
> > > > > **We have incorporated your feedback into the manuscript** and **revised the wording** in relevant sections to alleviate any confusion (see Appendix A.2, Section 5.3, and Appendix A.8 for details).
> > > > >
> > > > > Regarding 3) human annotators, **you are correct—some “non-key” authors participated in the human evaluation**. These individuals were either a) research advisors or b) contributors to the dataset creation who had no prior knowledge of the model’s performance on the evaluated tasks. We hope this clarification resolves this point.
> > > > >
> > > > > ## **The Long Context Experiment**
> > > > >
> > > > > **The training run has successfully concluded**. We trained it the Phi3.5-Vision model with a context length of **32k tokens**. Following this, we re-evaluated the Screenshot2HTML task using context lengths ranging from 512 to 16k. Using 32k tokens during generation gave out-of-memory errors. The results are presented in the table below (Table 29 added to the paper). We applied greedy decoding (second column) and beam search with a width of 2 and a length penalty of -1.0 (third column).
> > > > >
> > > > > | **Context Length** | **HTML Score**   | **HTML Score**         |
> > > > > |-------------------|------------------|-------------------------|
> > > > > |                   | _Greedy_         | _Length penalty = -1_   |
> > > > > | **512**           | 10.99            | 13.96                   |
> > > > > | **1,024**         | 11.49            | 15.80                   |
> > > > > | **2,048**         | 9.81             | 14.71                   |
> > > > > | **4,096**         | 8.83             | 13.88                   |
> > > > > | **8,192**         | 8.38             | 13.60                   |
> > > > > | **16,384**        | 8.24             | 13.43                   |
> > > > >
> > > > > **The results are revealing. We do not observe any performance boost when training and evaluating with longer context lengths.** The greedy decoding algorithm performs worse with longer contexts, not exceeding a score of 10.99. In contrast, beam search with a length penalty approach yields better results, reaching 15.80, but still does not benefit from larger context lengths.
> > > > >
> > > > > Additionally, **we conducted a qualitative error analysis** (see examples: https://imgur.com/a/v9bkO56) and found that *the model often struggles with stopping generation, frequently hallucinating repetitive patterns* such as nested <div> elements or excessively long URLs and references. However, well-formatted HTML code is generated in approximately 40% of cases (https://imgur.com/a/9oLDAlA, top figure). **We find that applying a length penalty helps mitigate this hallucination**, leading to improved performance. This suggests that there is further potential to enhance performance with alternative generation techniques.
> > > > >
> > > > > This indicates that the performance issue may not be due to context length but rather the inherent difficulty of the Screenshot2HTML task. This is consistent with the observation that the test set's average context length is approximately 3k tokens (see https://imgur.com/a/9oLDAlA) and does not necessitate more than 8k tokens. **We suspect that the primary limitation is the insufficient amount of training data**—currently around 10k samples—which may impede the model's ability to learn HTML generation effectively.
> > > > >
> > > > > ---
> > > > >
> > > > > ### **Returning to the initial question, “Does BigDocs-Bench results saturate at 50% because of the context length limit?”**
> > > > >
> > > > > Not necessarily. Given that the test set's context length is around 3k tokens, models with an 8k context length should be able to handle the tasks effectively. However, in our ablation experiments with larger context lengths, while some samples performed well, the majority exhibited hallucinated repetitive patterns. This indicates that the primary issue lies in the limited training data. Nonetheless, the inherently long nature of HTML (averaging 32k tokens in the training dataset, see Table 1) suggests that models require both extended context lengths and substantially more training data to excel in this task.
> > > > > We view these results positively, as they indicate a challenging task that is worth exploring further in the VLM field. This content has been included in the new Section A.12.1 of our appendix, "Ablation on Context Length."
> > > > >
> > > > > We included this content in the new Section A.12.1 of our appendix, Ablation on Context Length.
> > > > >
> > > > > ---
> > > > >
> > > > > **Final note:** Considering that we have effectively addressed all your concerns through substantial experiments and discussions, significantly improving the quality and clarity of our updated manuscript, we kindly ask you to consider increasing your scores to support our work.

---

> > > > > > ### Author Response · Authors · 2024-11-29
> > > > > >
> > > > > > Dear Reviewer,
> > > > > >
> > > > > > As the author-reviewer period comes to a close, we want to ensure that all your concerns have been addressed.
> > > > > >
> > > > > > To summarize your last concern on long context (as the full discussion might be lengthy): **TL;DR: Our experiments provided positive insights, showing that context-length tasks are both challenging and valuable for the VLM landscape. Notably, our benchmark does not saturate even with an 8k context window.**
> > > > > >
> > > > > > We kindly invite you to review these updates, share any additional feedback, and consider adjusting your scores if you feel your concerns have been resolved.
> > > > > >
> > > > > > Thanks!

---

> > > > > > > ### Author Response · Authors · 2024-12-02
> > > > > > >
> > > > > > > Dear Reviewer UxFn,
> > > > > > >
> > > > > > > Today is the last day of the discussion period and we are eager to hear your feedback on the rebuttal. **Given the lack of discussion in general, we kindly ask you to be bold in your decision and raise your score if you would like to support our work.**
> > > > > > >
> > > > > > > Best regards,
> > > > > > > BigDocs team

---

### Official Review · Reviewer_gUaq · 2024-11-10

**Soundness:** 3
**Presentation:** 3
**Contribution:** 3
**Rating:** 5
**Confidence:** 2

**Summary:**

The paper introduces BigDocs-7.5M, a large-scale, license-permissive dataset for training multimodal models on document and code-related tasks. Along with a comprehensive suite of tools and data analysis, the authors present BigDocs-Bench, featuring 10 downstream tasks that assess a model’s ability to generate long-format code outputs from images. These tasks serve as practical benchmarks for real-world applications. In paper the experiments show that models trained on BigDocs outperform those trained on existing datasets. All the artifacts in this paper is open source and permissive.

**Strengths:**

1. All the stuff in the paper is under permissive license, which is a big plus for an artifact and benchmark focused research paper.
2. The dataset curation process and filtering make sense for better data quality
3. Bonus point on keeping multimodal in mind when creating a document-understanding dataset.
4. Data contamination analysis is nice to have when the dataset proposes training set.

**Weaknesses:**

1. Many aspects of the data curation process, including the effect of OCR, VQA format and text-image alignment, etc. are sort of still unanswered in the experiment section, making readers wonder, why and if these curation processes are needed, or are they actually contributing to better performance resulted in the final model.
2. As a aggregation benchmark with a lot of different downstream tasks, how are the aggregated score on this BigDocs-Bench correlates with other existing benchmarks also presented in the paper? A comparison of what new aspects this new Benchmark is adding to existing research landscape is crucial to justify.

**Questions:**

See above

---

> ### Author Response · Authors · 2024-11-22
>
> We sincerely thank the reviewer for their thoughtful comments and constructive feedback. We appreciate the recognition of the dataset’s license permissiveness, data curation, multimodal considerations, and data contamination analysis. These were deliberate design decisions to ensure that BigDocs-7.5M and BigDocs-Bench contribute meaningfully to the research community.
>
> Below, we address each concern and clarify the aspects mentioned, incorporating additional analyses to the manuscript.
>
> >W1: The effect of OCR, VQA format, and text-image alignment is not demonstrated in the experiments.
>
> We appreciate the reviewer’s observation regarding the need for clarity on the impact of various data curation processes. To address this, **we conducted an ablation study to isolate the effects of each processing step**.
>
> **Experiment Setup**. Using the Phi3.5-Vision-Instruct model, we created three modified versions of BigDocs-7.5M: one without the VQA format, one reducing OCR tasks to 30%, and one reducing text-alignment tasks to 30%. Each dataset variant was used to train the model, followed by fine-tuning the model on downstream tasks. We evaluated these models on the benchmarks listed in Table 2 (General Document Benchmarks), with results shown below, including the differences from the original results in parentheses.
>
>
> | Model        | DocVQA | InfoVQA | DeepForm | KLC| WTQ | TabFact| ChartQA | TextVQA | MMMU | Dude-M | SlideVQA-M | TableVQA| Avg. Score |
> |--------------|---------------|----------------|------------------|-------------|-------------|-----------------|-----------------|----------------|-------------|--------------------|---------------------|------------------|------------|
> | Original     | 87.05         | 70.05          | 70.97            | 37.45       | 51.21       | 81.24           | 81.56           | 68.72          | 45.00       | 36.15              | 32.47               | 67.77            | 60.80      |
> | No VQA       | 82.65 (-4.4)  | 54.92 (-15.13) | 40.64 (-30.33)   | 28.8 (-8.65)| 36.93 (-14.28)| 65.3 (-15.94)   | 78.4 (-3.16)    | 65.24 (-3.48) | 42.33 (-2.67) | 34.71 (-1.44)     | 28.37 (-4.1)        | 61.13 (-6.64)    | **51.62** (-9.19) |
> | 30% OCR      | 85.79 (-1.26) | **60.03** (-10.02)| **54.19** (-16.78)| 34.9 (-2.55)| 52.68 (1.47) | 73.76 (-7.48)   | 81.56 (0)      | 72.07 (3.35)  | 42.56 (-2.44) | 36.72 (0.57)     | 32.07 (-0.4)        | 70.83 (3.06)     | **58.1** (-2.71)  |
> | 30% caption  | 85.36 (-1.69) | **57.98** (-12.07)| **53.36** (-17.61)| 35.09 (-2.36)| 50.91 (-0.3) | 72.5 (-8.74)    | 82.4 (0.84)    | 71.92 (3.2)   | 40.89 (-4.11) | 36.76 (0.61)     | 31.97 (-0.5)        | 69.37 (1.6)      | **57.38** (-3.43) |
>
>
> **Main Findings**
>
> 1. **VQA formatting was critical**, with its removal leading to a 9.19% average performance drop, especially on InfoVQA (-15.13), DeepForm (-30.33), WTQ (-14.28), and TabFact (-15.94). A common issue observed is the failure to follow the requested output format, as shown in Table 26 (see https://imgur.com/a/08H5tDH). We hypothesize that the VQA format used during continual pre-training (CPT) improves instruction tuning and supports multitask generalization.
>
> 2. **Removing OCR** tasks caused a 2.71% average performance drop, with significant declines on InfoVQA (-10.02), DeepForm (-16.78), and TabFact (-7.48), while DUDE, SlideVQA, and ChartQA remained largely unaffected. Reducing OCR data increased hallucination in tasks requiring precise information extraction. In DeepForm, for example, the true negative rate dropped from 62% to 0%, with models hallucinating unrelated values from other document sections. Examples of this behavior are shown in Tables 21 and 22 (https://imgur.com/a/bPomLGE). In some cases, such as TableVQA (+3.06) and TextVQA (+3.35), performance improved, likely due to the advantage of free-form text generation over exact OCR matches. (See example: https://imgur.com/a/jN8V84z)
>
> 3. **Reducing text-image alignment tasks**, especially captioning, caused a 3.43% performance drop, with significant declines on InfoVQA (-12.07), DeepForm (-17.61), and TabFact (-8.74). This drop was mainly due to errors in parsing values for arithmetic operations, especially when converting text or visual elements into percentages or performing calculations on extracted visual data. This is shown in Tables 23, 24, and 25 (see example here: https://imgur.com/a/6Oqy6RC). Tasks like WTQ and ChartQA, which involve simpler text-visual correspondences, do not require complex conversions and thus show no significant drops.
>
> These results highlight the importance of VQA format, OCR, and text-alignment tasks, and suggest the potential for optimal data mixtures across tasks, datasets, and sample sizes. We are experimenting with different weight mixtures of BigDocs-15M and will share the results if they provide valuable insights. We appreciate the reviewer’s suggestion, which led to this analysis, now included in Table 7 (Appendix A.12) of the updated manuscript.

---

> ### Author Response · Authors · 2024-11-22
>
> >W2: “How does the aggregated score on BigDocs-Bench correlate with existing benchmarks?”
>
> We thank the reviewer for raising this important question. To better contextualize BigDocs-Bench within the broader VLM and Document Understanding research landscape, **we conducted a correlation analysis, comparing BigDocs-Bench scores with those of other document benchmarks presented in our paper**. This evaluation includes both aggregated scores and task-level performance. We present the results in the form of a correlation matrix, a dendrogram plot, and a PCA analysis. **We facilitate the figures here: https://imgur.com/a/jYR57Jz**
>
> We also recommend the reviewer refer to our response to Reviewer UxFn (Response W2), which addresses the same issue and provides additional insights on broader VLM benchmarks.
>
> **Analysis Setup.** We selected all benchmarks listed in Tables 2 and 3, encompassing both General Document Benchmarks and our proposed BigDocs-Bench, respectively. We constructed a matrix encompassing all models presented in the paper, namely DocOwl-1.5-8B, Qwen2-VL-2B, LLaVA-NeXT-7B, Phi3.5-v-4B, both the off-the-shelf public versions and the ones trained on BigDocs. We also extended Table 2 with scores for GPT4o, Claude 3.5 Sonet, Qwen2-VL-72B, GeminiPro-1.5, and Llama-3.2-90B, to obtain a full matrix. Note that we have now extended our baselines with Llama-3.2-90B. Finally, we aggregated results for all metrics in the two groups of benchmarks. We have edited our manuscript to include Table 8, which is an extension of Table 2 with scores from all models.
>
> **Methods.**
> 1) **Cosine similarity matrix and dendrograms:** We normalize metric scores by centering (subtracting the mean) and scaling (dividing by the Euclidean norm). We then apply hierarchical clustering using cosine distance to group benchmarks and visualize their relationships. Cosine similarity matrices and dendrograms illustrate these clusters.
>
> 2) **Principal Components Analysis (PCA):** We represent each benchmark as a feature array composed of scores from all models and project these arrays onto the two principal dimensions.
>
> Figure 14 in the edited manuscript (Appendix A.15) presents the correlation matrix and dendrograms. Circles represent BigDocs-Bench tasks (our benchmarks), while triangles denote other document benchmarks included in the paper. These results demonstrate that *BigDocs-Bench tasks are notably distinct from other benchmarks*, as indicated by their low correlation with them. *A clear grouping emerges for tasks related to VQA, as well as BigDocs-Bench tasks involving code generation in LaTeX, JSON, GraphViz, HTML, and SVG*. Notably, HTML and SVG tasks form a cluster, likely due to their characteristically long output sequences. Additionally, DeepForm and KLC stand out as a separate cluster, distinct from all others.
>
> Figure 15 in the manuscript illustrates the PCA results. The clusters distinctly separate BigDocs-Bench tasks from other benchmarks, reaffirming their unique characteristics. *Notably, the average aggregated score from BigDocs-Bench (left side) is the furthest from the aggregated score of other benchmarks* (right side), highlighting the distinctiveness of our benchmarks. However, certain tasks, such as Image2SVG and Screenshot2HTML, show stronger correlations with KLC, DeepForm, and TabFact. This is likely due to the shared level of difficulty across these benchmarks, which require nuanced understanding and complex reasoning.
>
> **Unique Aspects of BigDocs-Bench.** Our correlation results show a clear separation between BigDocs-Bench and other document benchmarks. In summary, it introduces the following novel elements to the landscapes of VLM evaluation and document understanding:
>
> 1. **Document and Chart Reasoning:** Tasks focused on understanding, question answering, and summarization of charts, graphs, and workflows from images.
> 2. **Web and GUI Understanding:** Tasks targeting web and graphical user interface comprehension, a less explored area in current benchmarks.
> 3. **Multimodal Code Generation:** Tasks involving the conversion of images into code representations, such as inverse rendering and code generation (e.g., HTML, SVG, JSON).
> 4. **Long-Context Structured Output Generation:** Tasks requiring structured output over extended contexts, such as generating long JSON, SVG, HTML or Markdown from images

---

> > ### Author Response · Authors · 2024-11-25
> >
> > Dear Reviewer,
> >
> > Thank you again for your insightful review. It has greatly improved our paper, particularly with the ablation of our data curation pipelines and the correlation study you proposed.
> >
> > **We want to ensure that all your comments and concerns have been addressed.**
> >
> > Thanks!

---

> > > ### Author Response · Authors · 2024-11-29
> > >
> > > Dear Reviewer,
> > >
> > > Thank you for dedicating your time and effort to reviewing our manuscript and for providing insightful suggestions. As the author-reviewer discussion phase nears its conclusion, we would like to ensure that our responses have adequately addressed your concerns. Please feel free to reach out if you have any additional questions or require further clarification.
> > >
> > > Best regards

---

> > > > ### Author Response · Authors · 2024-12-02
> > > >
> > > > Dear Reviewer gUaq,
> > > >
> > > > Today is the last day of the discussion period and we are eager to hear your feedback on the rebuttal. **Given the lack of discussion in general, we kindly ask you to be bold in your decision and raise your score if you would like to support our work.**
> > > >
> > > > Best regards,
> > > > BigDocs team

---

### Author Response · Authors · 2024-11-29
**General Comments - Rebuttal Improvements**

Dear Reviewers,

We thank all reviewers for their constructive feedback, which has greatly helped us improve the manuscript. Below is a summary of the key changes, experiments, and additions made during the rebuttal phase:

#### **Experiments done**

1) **Correlation analysis** of BigDocs-Bench and other benchmarks
2) **Ablations on the data curation pipeline** (VQA, OCR, text-image alignment)
3) **Re-compute BigDocs-Bench leaderboard** with improved prompting
4) Ablations on the **context length**
5) Experiments with **larger models**
6) Enhanced **qualitative evaluations** with deeper error analysis and insights

All experiments yielded positive results, providing valuable insights and discussion.

#### **Other concerns that required clarity:**

7) Expanded **details on human data verification**, ensuring more thorough explanations.
8) Provided additional **detail on the human evaluation**

The reviewers who responded have **well-received these changes**. We are currently awaiting feedback from two reviewers regarding a few remaining concerns.

#### **Changes in the manuscript**

We have incorporated all reviewer feedback to enhance the writing and clarity of the main paper. The manuscript has expanded from **33 pages to 58 pages**, with **25 new pages in the appendix**, providing detailed qualitative and error analyses, as well as the clarifications and experiments conducted during the rebuttal phase.

To summarize: During the rebuttal, we updated Section 4.2, Tables 2 and 3 (main paper), as well as Section 5.3, Appendices A.1, A.8, A.10, A.12, A.13, A.14, A.15, alongside Tables 7–29 and Figures 14–22 (Appendix). These updates significantly improved the quality and clarity of our paper.

---

As the discussion period is close to an end, we kindly invite reviewers to review these updates, share feedback, and consider adjusting their scores if concerns have been addressed.

Thank you again for your thoughtful comments and for your time reviewing our work.

---

### Author Response · Authors · 2024-12-04
**Final General Comment (please read)**

We summarize the rebuttal as follows:

During the rebuttal period, we made significant efforts to address the reviewers' concerns comprehensively. **We added over 25 pages of detailed content to the appendix, including additional experiments, analyses, and clarifications.** These additions directly addressed all the reviewers' concerns.

---


These were the main concerns:
1) Datasets correlation analysis
2) More ablations and experiments with larger models
3) Boost baseline results
4) Clarifications on human data verification processes

---

- *3/4 reviewers provided at least one reply, validating that our response successfully addressed all their concerns**.

- **Reviewer 1wn6, with the highest confidence of 4, raised their overall evaluation from 6 to 8**, firmly supporting acceptance.


- **Reviewer UxFn (score of 5) was satisfied with our clarifications and was awaiting our final experiment.** Although we submitted it as requested, **they became silent** and the score was not updated.


- **Reviewer gUaq (score of 5) did not reply to our response**, which is disappointing, as their questions (ablations and correlation) led to insightful results and significantly improved our work!


- **Reviewer ZUCq (score of 5)** was satisfied with our clarifications but **decided to keep 5 as they were not convinced about the broader impact**

---
**Our broader impact:** The current state of the AI industry is disheartening. Despite data being the most critical driver of model performance—arguably even more than architectural innovations—companies often hoard datasets and practices behind closed doors. This lack of openness stifles progress and collaboration, creating barriers that limit the equitable growth of AI.

In our work, we aim to show how large, reliable datasets can be built in a responsible and transparent manner. By emphasizing openness and accountability, we hope to encourage a more collaborative and equitable approach to AI development, ensuring that data—the foundation of lasting progress—remains a shared resource for innovation.

---

**Finally, our submission received excellent scores across critical aspects, such as soundness (3, 3, 4, 3), presentation (3, 4, 4, 3), and contribution (3, 3, 3, 3).** However, we noticed discrepancies in the **scores of 5 provided, which do not align with the positive comments and judgments they shared**. We understand the reviewers may have faced time constraints during this busy period and respectfully request their support in revisiting their overall scores to better reflect the positive attitude highlighted in their feedback.

Best Regards,

BigDocs Team

---

### Meta-Review · Area_Chair_DUoh · 2024-12-16

**Metareview:**

The paper introduces BigDocs-7.5M, a large-scale, open-access dataset (licensed under CC-BY-4.0) designed to advance multimodal model training for document and code-related tasks. Addressing the limitations of existing datasets, BigDocs-7.5M encompasses 7.5 million diverse documents suitable for tasks such as document reasoning and structured output generation. Additionally, the work presents BigDocs-Bench, an evaluation suite comprising 10 novel tasks, providing comprehensive insights into model performance. Models trained on BigDocs-7.5M outperform those relying on closed-source datasets, with fine-tuned Phi-3.5 achieving approximately 50% accuracy, significantly surpassing GPT-4’s ~25%.

The availability of a large-scale open-source dataset with permissive licensing is a critical milestone for the open-source community. Combined with rigorous evaluation, this work represents a significant and impactful contribution to the field.

**Additional Comments On Reviewer Discussion:**

The author’s response during the rebuttal period was highly informative.
(1) It provides detailed explanations about the annotators and annotation process involved in creating BigDocs-Bench, which validates the benchmark’s quality.
(2) It includes several new experiments to further validate the design of the work. For example, there are new ablations on the effect of OCR, VQA format, and text-image alignment and demonstrating the necessity of them.
(3) It provides further clarifications on various questions raised by the reviewers, mostly on the missing details.

From my perspective, the submission did not exhibit any significant deficiencies, even in the initial round of reviews. The rebuttal effectively addressed the concerns raised by the reviewers. However, it is regrettable that the reviewers appeared to lack sufficient engagement during the period.

---

### Decision · Program_Chairs · 2025-01-22

Accept (Poster)